# ❤ HEARTS: Benchmarking LLM Reasoning on Health Time Series

Sirui Li [* 1]  Shuhan Xiao [* 1]  Mihir Joshi [1]  Ahmed Metwally [2]  Daniel McDuff [2]  Wei Wang [1]  Yuzhe Yang [1]

## Abstract

The rise of large language models (LLMs) has shifted time series analysis from narrow analytics to general-purpose reasoning. Yet, existing benchmarks cover a limited set of *health time series* modalities and tasks, failing to capture the diverse domains and temporal dependencies of real-world physiological modeling. To bridge this gap, we introduce **HEARTS** (**Hea**lth **R**easoning over **T**ime **S**eries), a unified benchmark for evaluating hierarchical LLM reasoning over general health time series. HEARTS integrates 16 datasets across 12 domains and 20 signal modalities, defining a taxonomy of 110 tasks grouped into four capabilities: *Perception*, *Inference*, *Generation*, and *Deduction*. Evaluating 16 state-of-the-art LLMs on over 20K test samples reveals intriguing findings. First, LLMs substantially underperform specialized models, with performance only weakly related to general reasoning scores. Moreover, LLMs often rely on simple heuristics and struggle with multi-step temporal reasoning. Finally, performance declines with increasing temporal complexity, with similar failure modes within model families, indicating scaling alone is insufficient. By making these gaps measurable, HEARTS provides a standardized testbed and living benchmark for developing next-generation LLM agents capable of reasoning over diverse health signals. Code and data are available at https://yang-ai-lab.github.io/HEARTS/

## 1. Introduction

Time series data serve as the backbone of decision-making in various domains, including energy (Alvarez et al., 2010), finance (Lu & Xu, 2024), transportation (Guan et al., 2023), climate (Mudelsee, 2019), political science (Beck & Katz,

---

*Equal contribution [1]UCLA [2]Google Research. Correspondence to: Yuzhe Yang <yuzhey@ucla.edu>.

*Proceedings of the 43rd International Conference on Machine Learning*, Seoul, South Korea. PMLR 306, 2026. Copyright 2026 by the author(s).

2011), and healthcare (Xu et al., 2026; Zhang et al., 2025). Among these, healthcare is uniquely high-stakes: physiological time series encode multi-scale dynamics across time and frequency, where a brief anomaly, a slow drift, or a cross-channel interaction can change diagnosis and treatment (Shuai et al., 2026; Yang et al., 2022). Consequently, the ability to *reason* over these signals by linking observations to mechanisms and context is a key requirement for the next generation of healthcare artificial intelligence.

The rise of large language models (LLMs) has sparked a shift in this challenge. Because LLMs show strong reasoning over text, code, and mathematics (Shao et al., 2024; Wei et al., 2022; Yang et al., 2025b), recent work has begun adapting them to time series, aiming to transfer general problem-solving skills to temporal data (Gruver et al., 2023; Jin et al., 2023). However, a critical question remains:

*Do LLMs perform **genuine reasoning** over health time series, or only exploit surface patterns and domain priors?*

While LLMs are effective at semantic reasoning in language, their ability to carry out multi-step deduction, cross-channel correlation, and long-horizon synthesis on numerical physiological data remains unclear and is often limited in practice (Feli et al., 2025; Jin et al., 2024; Zhou & Yu, 2024).

Despite the pressing need, evaluation has not kept pace with model development. On the one hand, existing time series reasoning benchmarks (Cai et al., 2024; Chen et al., 2025; Sen et al., 2025) underrepresent health time series, failing to capture realistic temporal structures such as long-range dependencies and extreme variances in scale and frequency. On the other hand, benchmarks that include health data either rely on synthetic signals that miss real clinical context and noise (Guan et al., 2025; Merrill et al., 2024), or are narrowly focused on a single domain and a small set of modalities (e.g., ECG) (Kong et al., 2025; Oh et al., 2023b; Ye et al., 2025), limiting their ability to test diverse, human-level reasoning over heterogeneous health signals.

To bridge this gap, we present HEARTS (**Hea**lth **R**easoning over **T**ime **S**eries), a unified living benchmark designed to evaluate hierarchical reasoning over realistic health time-series. HEARTS integrates **16** real-world datasets across **12** health domains, covering **20** signal modalities with diverse temporal resolutions and frequency ranges (Fig. 1). Cru-

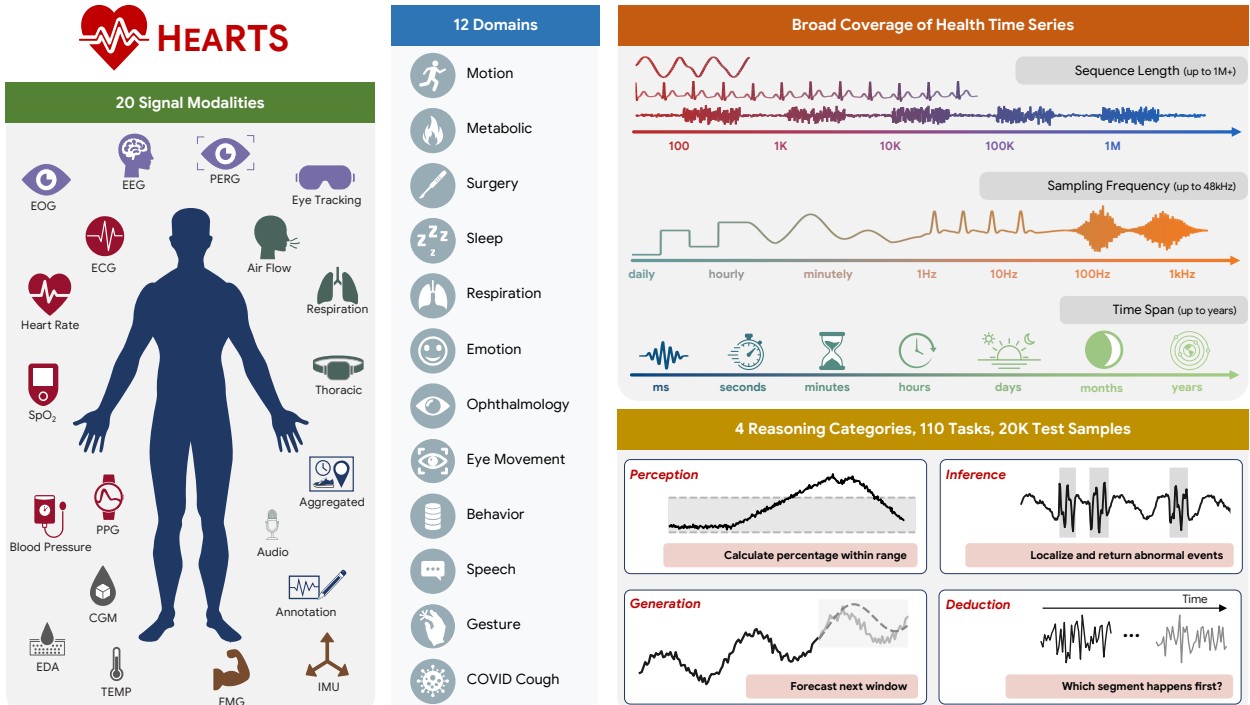

*Figure 1.* **Overview of HEARTS.** We present the first diverse benchmark for health time-series reasoning, encompassing 20 signal modalities spanning 16 datasets and 12 health domains, with to date the broadest coverage of sequence length, frequency, and time span. It comprises over 20K test samples across 110 tasks, organized into four reasoning categories. More details are in Appendix A.2 and A.3.

cially, we define a comprehensive taxonomy of **110** tasks organized into four capability families: *Perception*, *Inference*, *Generation*, and *Deduction*. This design supports controlled evaluation from lower-level signal understanding to higher-level reasoning that requires long-context integration, cross-channel consistency, and temporal directionality.

HEARTS is designed to be easily extensible, supporting new models, datasets, and tasks as the field evolves. It operates as a *living ecosystem*, with standardized community submissions and rigorous quality checks to support continual expansion. Using the initial release, we evaluate **16** state-of-the-art (SOTA) LLMs on more than **20K** test samples, and reveal findings that highlight key gaps and directions for future research. Our contributions are as follows:

- We introduce HEARTS, the first diverse benchmark for health time-series reasoning, spanning 12 health domains, 20 signal modalities, and 110 tasks, with to date the broadest coverage of sequence length, frequency, and time span.
- We propose a unified, hierarchical evaluation setting that goes beyond standard multiple-choice questions, with four cognitive levels that range from basic calculation to long-horizon synthesis and counterfactual deduction.
- Experiments on 16 LLMs over more than 20K test samples show that SOTA models struggle with health time-series reasoning across tasks, and that performance is only weakly related to general reasoning scores.
- We identify consistent scaling challenges with longer inputs, higher sampling frequencies, and longer time spans,

revealing model-agnostic difficulty patterns and highlighting open directions for future research.

## 2. Related Work

**Language Models for Time Series Analysis.** Early work on adapting LLMs to time series focused on forecasting, either by bridging modality gaps through reprogramming and feature alignment (Jin et al., 2023; Li et al., 2024) or by leveraging long-context prompting for zero-shot prediction (Liu et al., 2024b; Lu et al., 2024). Recent studies shift toward agentic and tool-augmented settings, using multi-agent designs for data understanding (Lee et al., 2025; Zhou et al., 2025) and combining analytical tools with hierarchical chain-of-thought for more complex reasoning workflows (Guan et al., 2025; Liu et al., 2026). Despite this progress, it remains unclear how well these methods support *physiological* reasoning, including cross-signal correlation, long-horizon synthesis, and clinically meaningful inference. In this work, we propose a dedicated benchmark for evaluating LLM reasoning over health time series.

**Time Series Reasoning Benchmarks.** Healthcare remains underrepresented in existing time series reasoning benchmarks. Most frameworks target general-domain data, with health modalities absent or only lightly covered (Cai et al., 2024; Chen et al., 2025; Sen et al., 2025). Benchmarks that include health settings often sacrifice realism for breadth, or remain limited in domain and modality coverage. For

*Table 1.* **Comparisons between HEARTS and other time series reasoning benchmarks.**

| Benchmark | Real World? | Sequence Length Range | Frequency Range | Time Range | # Domain | # TS Modality[†] | # Test Sample[†] | # Task | Task Type | | | |
|---|---|---|---|---|---|---|---|---|---|---|---|---|
| | | | | | | | | | Per. | Inf. | Gen. | Ded. |
| *TimeSeriesExam* (Cai et al., 2024) | ✗ | 1K | - | - | 1 | 0 | 0 | 100 | ✓ | ✓ | ✗ | ✗ |
| *BEDTime* (Sen et al., 2025) | ✗ | 36–25K | - | - | 2 | 0 | 0 | 12 | ✓ | ✓ | ✗ | ✗ |
| *TSR-SUITE* (Guan et al., 2025) | ✗ | 48–0.8K | daily–hourly | hours–days | 10 | 15 | - | 36 | ✗ | ✓ | ✓ | ✗ |
| *TS Reasoning* (Merrill et al., 2024) | ✗ | 10–1.5K | annual–hourly | hours–years | 10 | 15 | 14K | 50 | ✗ | ✓ | ✓ | ✗ |
| *MTBench* (Chen et al., 2025) | ✓ | 168–1.3K | hourly–minutely | days–months | 2 | 0 | 0 | 12 | ✓ | ✓ | ✗ | ✗ |
| *TSAIA* (Ye et al., 2025) | ✓ | 10–0.9K | daily–128Hz | secs–years | 4 | 1 | 0.07K | 12 | ✗ | ✓ | ✓ | ✗ |
| *ECG-QA* (Oh et al., 2023b) | ✓ | 5K | 500Hz | seconds | 1 | 1 | 8K | 70 | ✗ | ✓ | ✗ | ✗ |
| *Time-MQA* (Kong et al., 2025) | ✓ | 8–0.2K | daily–256Hz | secs–days | 12 | 3 | 56K | 60 | ✗ | ✓ | ✓ | ✗ |
| **HEARTS (ours)** | ✓ | **60–1M** | **daily–48kHz** | **secs–years** | **12** | **20** | **20K** | **110** | ✓ | ✓ | ✓ | ✓ |

Per.: Perception, Inf.: Inference, Gen.: Generation, Ded.: Deduction. ■ Sequence Length Coverage ■ Frequency Coverage ■ Time Coverage [†]Only health time series considered.

example, recent benchmarks rely on synthetic health signals (Guan et al., 2025; Merrill et al., 2024), which can miss real physiological semantics, noise, and clinical constraints. Conversely, benchmarks built on real-world medical data are typically confined to a few domains and a small set of modalities (Kong et al., 2025; Oh et al., 2023b; Ye et al., 2025). As summarized in Table 1, current benchmarks therefore lack the diversity and fidelity required to assess human-level reasoning in real-world health contexts. In contrast, HEARTS addresses these gaps by providing a unified, health-focused evaluation framework with diverse real-world datasets and a hierarchical task taxonomy that tests reasoning across disparate temporal scales and healthcare contexts.

## 3. HEARTS

To address the evaluation gap, we introduce HEARTS, the first unified benchmark for health time-series reasoning. Departing from the conventional paradigm that reduces diverse objectives to a single QA format, HEARTS establishes a taxonomy anchored in four categories: Perception, Inference, Generation, and Deduction. This allows for evaluation in intrinsic task formats rather than relying solely on MCQs. HEARTS integrates 16 datasets across 12 domains and 20 signal modalities, encompassing 110 tasks and 20,226 test cases to ensure robust and comprehensive evaluation.

### 3.1. Data Overview

**Dataset Diversity and Domain Coverage.** HEARTS curates 16 datasets across 12 domains to facilitate rigorous benchmarking. This encompasses a broad spectrum of health contexts (complete statistics in Appendix A.1):

- **Motion:** Human activity recognition (HAR) and intensity estimation via wearable IMU data from Capture24 (Chan et al., 2024) and PAMAP2 (Bleser et al., 2015).
- **Metabolic:** Longitudinal glucose variability and dietary response tracking via Shanghai Diabetes (Zhao et al., 2023) and CGMacros (Das et al., 2025).
- **Surgery:** Intra-operative vital sign monitoring and anesthesia management using VitalDB (Lee et al., 2022).
- **Sleep:** Polysomnography (PSG) based sleep physiology analysis from SHHS (Quan et al., 1997).
- **Respiration:** Respiratory mechanics and breathing pattern

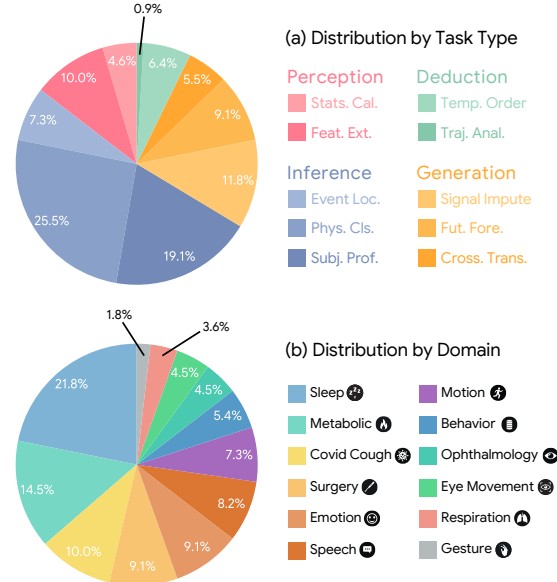

*Figure 2.* **Task and domain distributions in HEARTS.**

analysis utilizing Harespod (Zhang et al., 2024).
- **Emotion:** Affective state recognition derived from multi-modal biosignals in PhyMER (Pant et al., 2023).
- **Ophthalmology:** Retinal function assessment via pattern electroretinography (PERG) signals in PERG-IOBA (Fernández et al., 2024).
- **Eye Movement:** Oculomotor trajectory analysis using GazeBase (Griffith et al., 2021).
- **Behavior:** Longitudinal behavior and mental health monitoring using GLOBEM (Xu et al., 2023).
- **Speech:** Voice biomarker identification and pathological speech analysis using Bridge2AI-voice (Bensoussan et al., 2025) and VCTK (Yamagishi et al., 2019).
- **Gesture:** Electromyography (EMG) based hand gesture classification leveraging GrabMyo (Pradhan et al., 2022).
- **COVID Cough:** Respiratory symptom screening via cough audio recordings in CoughVID (Orlandic et al., 2021) and Coswara (Sharma et al., 2020).

**Multi-Modal Signal Complexity.** HEARTS highlights broad modality coverage: it includes 20 distinct signal channels, spanning standard bioelectrical recordings to high-dimensional acoustic streams and daily aggregated measurements (Fig. 1). This heterogeneity goes beyond prior

*Table 2.* **Breakdown of LLM performance across each reasoning category.** The best results across all models are shown in **bold**, and the second-best results are underlined. Overall scores are computed as the macro-average of the four category scores. SOTA ML performance is reported only on the 32-task subset, as detailed in Sec. 4.1.

| Model | Perception | | Inference | | | Generation | | | Deduction | | Overall Score |
|---|---|---|---|---|---|---|---|---|---|---|---|
| | Stat. Calc. | Feat. Ext. | Event Loc. | Phys. Cls. | Subj. Prof. | Sig. Imp. | Fut. Fore. | Cross. Trans. | Temp. Ord. | Traj. Anal. | |
| Naive Baseline | 1.00 | 1.00 | 0.21 | 0.33 | 0.45 | 0.68 | 0.59 | 0.59 | 0.45 | 0.50 | 0.61 |
| SOTA ML* | - | - | 0.78 | 0.85 | 0.86 | - | - | - | - | 0.74 | - |
| *Non-Reasoning Models:* | | | | | | | | | | | |
| Nemotron Nano 12B V2 | 0.58 | 0.49 | 0.22 | 0.34 | 0.50 | 0.61 | 0.49 | 0.44 | 0.45 | 0.54 | 0.47 |
| Llama 4 Maverick | 0.75 | 0.71 | 0.30 | 0.40 | 0.56 | 0.75 | 0.59 | 0.54 | 0.50 | 0.49 | 0.58 |
| GPT 4.1 Mini | 0.89 | 0.77 | 0.29 | 0.43 | 0.56 | 0.79 | 0.63 | 0.57 | 0.56 | **0.55** | 0.63 |
| Claude 4.5 Haiku | 0.93 | 0.79 | 0.32 | 0.43 | 0.52 | 0.80 | 0.65 | 0.60 | 0.55 | 0.50 | 0.63 |
| Qwen3 235B | 0.88 | 0.76 | 0.29 | 0.40 | 0.56 | 0.78 | 0.64 | 0.59 | 0.62 | 0.49 | 0.63 |
| Qwen3 Coder 480B | 0.92 | 0.80 | 0.32 | 0.43 | 0.55 | 0.82 | 0.66 | 0.66 | 0.51 | 0.47 | 0.63 |
| DeepSeek V3.1 | 0.91 | 0.79 | 0.29 | 0.43 | 0.55 | 0.77 | 0.63 | 0.60 | 0.62 | 0.52 | 0.64 |
| *Reasoning Models:* | | | | | | | | | | | |
| MiniMax M2 | 0.88 | 0.68 | 0.31 | **0.48** | 0.52 | 0.81 | 0.64 | 0.64 | 0.50 | **0.55** | 0.61 |
| Gemini 2.5 Pro | 0.82 | 0.74 | 0.32 | 0.40 | 0.51 | 0.79 | 0.65 | 0.65 | 0.59 | 0.45 | 0.62 |
| Gemini 2.5 Flash | 0.88 | 0.74 | 0.27 | 0.41 | 0.54 | 0.77 | 0.60 | 0.61 | 0.60 | 0.46 | 0.62 |
| GPT 5 Mini | 0.88 | 0.74 | 0.23 | 0.42 | 0.55 | 0.79 | 0.63 | 0.64 | 0.57 | 0.48 | 0.62 |
| Grok 4.1 Fast | 0.93 | 0.78 | 0.34 | 0.44 | 0.55 | 0.83 | 0.66 | 0.63 | 0.60 | 0.45 | 0.65 |
| Kimi K2 Thinking | 0.91 | 0.80 | 0.36 | 0.44 | 0.56 | 0.82 | 0.67 | 0.63 | 0.60 | 0.43 | 0.65 |
| GLM 4.7 | 0.92 | 0.80 | 0.38 | 0.45 | 0.55 | 0.82 | 0.68 | 0.64 | 0.60 | 0.47 | 0.66 |
| GLM 5 | **0.94** | 0.81 | 0.40 | 0.44 | 0.56 | 0.84 | 0.67 | 0.65 | 0.68 | 0.52 | 0.68 |
| Gemini 3.1 Pro | **0.94** | **0.82** | **0.50** | 0.43 | **0.57** | **0.85** | **0.70** | **0.69** | **0.70** | 0.39 | **0.69** |

limited-modality benchmarks and challenges models to generalize across signals with fundamentally different physical origins, sampling regimes, and physiological semantics.

**Temporal Scale and Resolution.** HEARTS captures wide variation in temporal scale: *input lengths* range from 60 to over 1M steps, and *sampling frequencies* span from daily metrics to 48kHz. Consequently, observation windows vary from seconds to years. This setting requires models to handle both high-frequency structure and long-range dependencies across highly different resolutions.

### 3.2. Task Overview

**Design Principles and Formulation.** Task construction in HEARTS follows 3 principles that ensure practical relevance, label validity and well-defined reasoning structure: ❶ *Evidence-Grounded Significance*: each task is anchored in real world needs by aligning its definition with authoritative medical guidelines or prior peer-reviewed studies of the same (or closely related) problem. ❷ *Context-Grounded Label Validity*: labels are directly derived directly from the recorded signals and metadata, avoiding external assumptions that cannot be checked within the dataset. ❸ *Data-Grounded Multi-Step Reasoning*: solving a task must require non-trivial, multi-step reasoning over the data instead of single-shot pattern matching. We formalize the reasoning

process as a Markov Decision Process (MDP) trajectory $\tau = \{(s_i, a_i, r_i)\}_{i=0}^{n}$, where states $s_i \in S$ encode the data context, actions $a_i \in A$ denote intermediate decisions, and rewards $r_i \in R$ evaluate the validity of the reasoning path. This formulation enforces that successful solutions proceed through a sequence of data-grounded decisions rather than a single shortcut step.

**Hierarchical Reasoning Taxonomy.** Guided by these principles, we curate 110 tasks grouped into four hierarchical reasoning categories. Fig. 2 summarizes their distribution across task types and domains. A key feature of HEARTS is that it moves beyond the multiple-choice paradigm, emphasizing exact numerical answers and open-ended generation. Appendix A.2 and A.3 provide detailed design examples.

- *Perception.* This category focuses on identifying physiological markers that serve as prerequisites for downstream analysis. Tasks include:
    - *Statistical Calculation:* Computing descriptive statistics of raw signals (e.g. the percentage of time a CGM signal remains within normal range).
    - *Feature Extraction:* Derive informative representations or biomarkers from signals (e.g. isolating spectral bandpower from EEG data).
- *Inference.* This category advances to analysis at varying granularities, ranging from precise temporal grounding to holistic subject characterization. It includes:

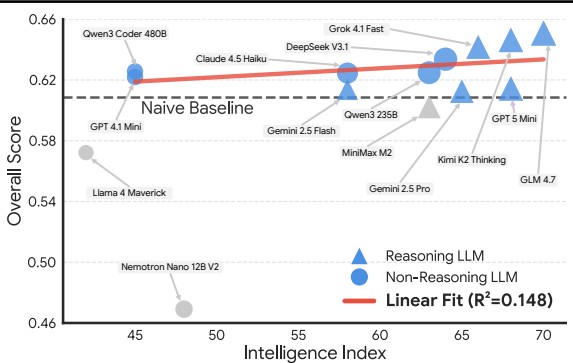

*Figure 3.* **Regression analysis of intelligence index vs. performance on HEARTS.** Models performing below the naive baseline are excluded as outliers.

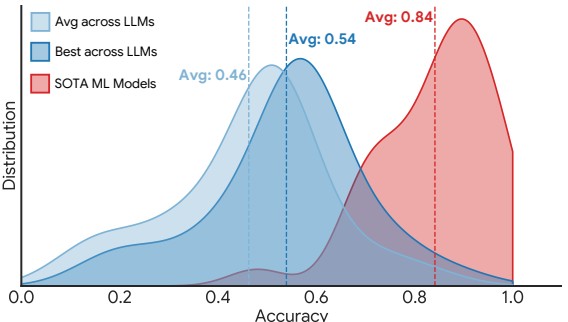

*Figure 4.* **Performance comparison of state-of-the-art ML methods and LLMs (best/average) across 32 tasks.** Density reflects the concentration of tasks at each performance level.

- *Event Localization:* Identify the timestamp or temporal boundaries of specific events within a signal.
- *Physiological Classification:* Categorizing physiological state of a signal segment (e.g., sleep staging).
- *Subject Profiling:* Inferring holistic, subject-level attributes or conditions (e.g., Parkinson's prediction).
- *Generation.* This category evaluates temporal understanding by requiring point-by-point synthesis of complete sequences. Three types of task are included:
  - *Future Forecasting:* Predict future segments of a biosignal based on observed historical context.
  - *Signal Imputation:* Reconstructing missing or corrupted portions within a temporal sequence to restore signal integrity.
  - *Cross-modal Translation:* Synthesizing a target signal modality from a source channel based on their underlying physiological correlations.
- *Deduction.* Finally, this category evaluates arrow-of-time reasoning and the ability to derive insights from longitudinal dependencies and multi-session interactions:
  - *Temporal Ordering:* Determine the relative temporal order of signals.
  - *Trajectory Analysis:* Analyze long-term health trajectories across multiple visits or sessions.

### 3.3. HEARTS as a Living Ecosystem

Unlike traditional benchmarks that become static artifacts, we position HEARTS as a dynamic, community-driven ecosystem that can evolve with rapid progress in AI for healthcare. We provide standardized contribution protocols on our project website, allowing the broader research community to submit new *datasets*, *clinical tasks* (task APIs in Appendix A.4.1), and new *agent architecture* (agent APIs in Appendix A.4.2). In addition, any LLM can be evaluated in HEARTS through a simple API endpoint interface. To maintain benchmark integrity, we use a human-in-the-loop maintenance process: a dedicated team reviews submissions for correctness and compliance with privacy requirements before merging them into the core repository. This continu-

ous integration setup keeps the leaderboard up to date (see Fig. 11), supporting a collaborative benchmark that grows in coverage and difficulty through community effort.

## 4. Experiments and Analysis

**Standards and Baselines.** We standardize evaluation by using 200 test cases per task, unless limited by data availability, yielding 20,226 test cases in total. Each task is scored with a metric that matches its output type: Accuracy, IoU, or sMAPE. For the sMAPE-based tasks, we first apply min-max normalization to predictions and ground truth, and report $1 - \frac{1}{2}$sMAPE so that scores are strictly bounded in $[0, 1]$ (higher is better). We also include a naive baseline to contextualize performance across tasks and metrics. The naive baseline details are provided in Appendix B.1.

**Model Selection.** To address context-window limits when working with long time-series inputs, we standardize evaluation with the CodeAct framework (Wang et al., 2024a). CodeAct allows an LLM to reason by writing Python code that reads and analyzes data from files, so the model does not need to ingest the full raw sequence in its prompt. To evaluate the model's intrinsic reasoning capabilities, rather than its ability to call advanced tools, we intentionally restrict the environment to minimal Python packages necessary for health time series processing (e.g., *neurokit2*). We evaluate 16 state-of-the-art models spanning 11 families, covering both lightweight open-source models and large proprietary reasoning systems. Specifically, we evaluate: GPT-4.1 and GPT-5 mini (OpenAI, 2025; Singh et al., 2025a), Claude 4.5 Haiku (Anthropic Team, 2025), Gemini 2.5 Pro and 2.5 Flash (Comanici et al., 2025), Gemini 3.1 Pro (Google DeepMind, 2026), Qwen3 235B and Qwen3 Coder 480B (Yang et al., 2025a), DeepSeek V3.1 (Liu et al., 2024a), Llama 4 Maverick (Meta AI, 2025), Grok 4.1 Fast (xAI, 2025), MiniMax M2 (MiniMax AI, 2025), GLM 4.7 (Zhipu AI, 2025) and GLM 5 (GLM-5-Team et al., 2026), Nemotron Nano 12B V2 (Basant et al., 2025), and Kimi K2 Thinking (Team et al., 2025).

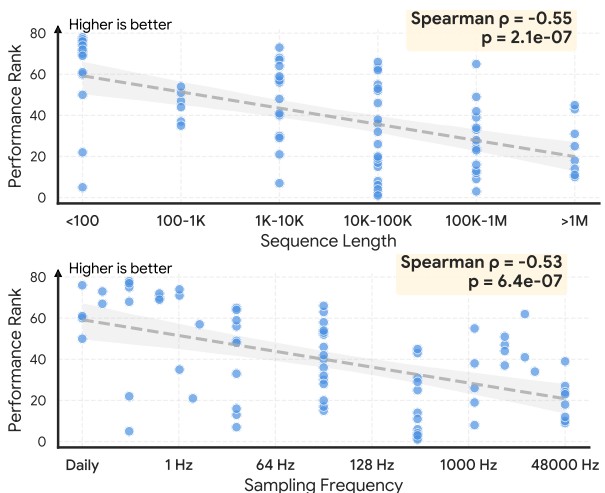

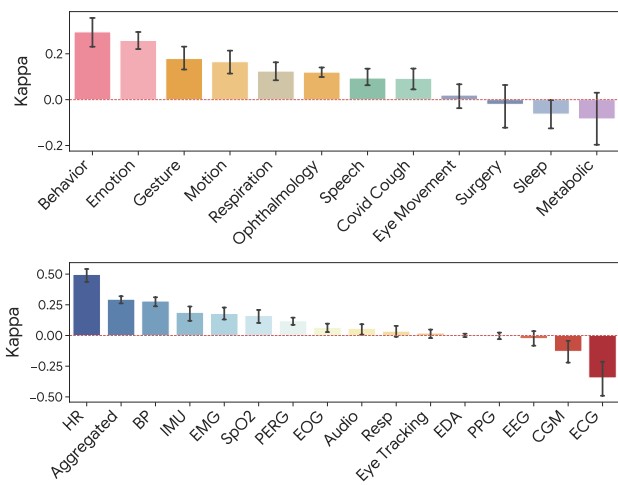

*Figure 5.* **Regression analysis of performance gain vs. data complexity.** Each point represents the 16-model average Kappa.

*Figure 6.* **Performance gains across different input modalities and domains.**

## 4.1. Are LLMs Good at Health Time-Series Reasoning?

**LLMs make only small gains over a naive baseline, and performance is weakly related to general intelligence indexes.** Table 2 shows that most models outperform the naive baseline, but by modest margins, suggesting that current LLMs still lack strong reasoning ability in the health time-series setting. We further test whether success on HEARTS tracks general reasoning capability using the intelligence index from (Kim et al., 2025) (Appendix B.2). After removing clear outliers that do not exceed the naive baseline, Fig. 3 shows no meaningful linear correlation between intelligence index and HEARTS performance. This indicates that health time-series reasoning is not well predicted by performance on broad reasoning benchmarks, and likely depends on domain- and data-specific skills that are not captured by general scores.

**LLMs lag far behind specialized time-series models.** To quantify this gap, we compare our evaluated LLMs (reporting both average and best performance across 16 models) against published task-specific SOTA baselines on a curated subset of 32 *Inference* and *Deduction* tasks (Fig. 4). We observe a clear separation: specialized models concentrate in a high-performance regime, whereas LLMs are shifted substantially lower, even under best-case selection. Despite minor experimental detail differences, the non-overlapping distributions indicate that general-purpose LLM agents still fall short of the precision achieved by domain-specific methods on challenging physiological reasoning tasks.

Beyond standard ML baselines, we also evaluate OpenTSLM (Langer et al., 2025), a family of time-series language models designed for reasoning over temporal data. Following the original experimental settings, we benchmark OpenTSLM on three representative HEARTS tasks. We find that OpenTSLM underperforms relative to CodeAct-enabled LLMs (complete results are in Appendix C.6).

> **Finding 1:** LLMs lag substantially behind specialized models, and the performance on health time series is only weakly related to their general reasoning capabilities.

## 4.2. Category-Level Behavioral Consistency in LLMs

**Perception and Inference tasks depend on explicit rules and data separability.** In the Perception category, while aggregate performance remains robust, LLMs demonstrate heightened proficiency specifically on tasks with well-defined algorithms or invokable functions (see Appendix C.2). Inference tasks reveal a polarized performance landscape, hinging largely on the explicitness of task rules and the inherent separability of the data. Models outperform random guessing only on tasks with clear rule-based thresholds or high intrinsic discriminability aligned with their pre-trained knowledge. For instance, agents succeed in scenarios with explicit quantitative thresholds (e.g., hypotension at 65) or clear common-sense patterns (e.g., reduced mobility during COVID). Conversely, performance deteriorates significantly in scenarios requiring fine-grained distinctions without reference, or specialized domain feature extraction (e.g., raw EMG interpretation). Notably, Subject Profiling degrades to near-random guessing without explicit medical thresholds (e.g., diabetes CV), highlighting the model's inability to capture transient events or derive holistic insights without priors.

**Generation and Deduction tasks are governed by low-complexity heuristics.** Performance on Generation tasks is characterized by a reliance on low-complexity heuristics rather than deep temporal reasoning. Across Signal Imputation and Future Forecasting tasks, LLM strategies are remarkably simplistic, predominantly defaulting to copy-pasting with noise injection, linear interpolation, basic statistical averaging, or regression fitting. Even in the rare instances where LLMs deploy auto-regressive models, the re-

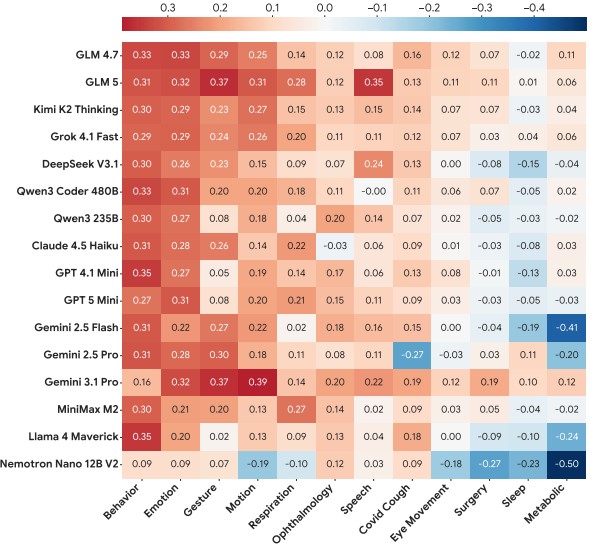

*Figure 7.* **Model performance heatmap across domains** The heatmap illustrates the Kappa scores across 12 domains. Heatmap for different input modalities is provided in Appendix C.3.

sulting generations remain no more sophisticated. A similar trend is observed in Cross-modal Translation tasks, where behavior is highly consistent yet formulaic. Instead of employing novel generative synthesis, models revert to standard deterministic signal processing routines, such as peak finding for heart rate estimation, thereby revealing a tendency to treat generation as a mechanical transformation rather than a semantic reconstruction. Performance on Deduction tasks parallels the limitations observed in inference, suggesting a deficiency in modeling temporal causality and physical signal continuity. Successful predictions appear to depend heavily on matching salient signal features to generalized priors, such as associating delta waves with early sleep stages, rather than capturing underlying dynamics. A comprehensive breakdown of experimental results across all categories is presented in Table 2.

> **Finding 2:** LLMs rely heavily on low-complexity heuristics and explicit priors, and show limited deep temporal reasoning across task categories.

### 4.3. Impact of Temporal Resolution and Data Properties

**Longer input sequences and higher sampling frequencies correlate with worse performance.** Our statistical analysis regarding input sequence length and sampling frequency validates that increased temporal resolution negatively impacts reasoning efficacy (Fig. 5). To rigorously isolate the effect of temporal resolution, we exclude six outlier tasks whose performance is confounded by intrinsic task characteristics, namely low-frequency tasks dominated by complex biological reasoning rather than data scale, and high-frequency tasks that are solvable through exploitable statistical patterns or low-variability outputs. Within the filtered set, we identify a significant inverse correlation between performance

and input sequence length, alongside a parallel negative trend regarding sampling frequency. These results quantify the "cognitive burden" of high-resolution temporal data, confirming that current LLMs struggle to maintain performance as temporal horizons expand and signal density increases.

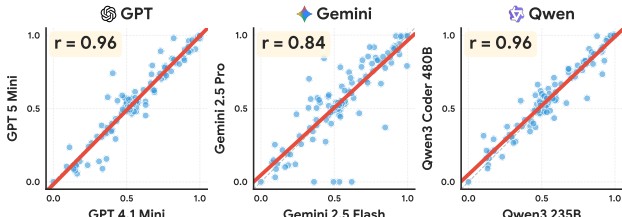

*Figure 8.* **Analysis of behavioral consistency across LLM families.** Each dot denotes a task, with coordinates indicating the performance of the two compared models.

**LLMs show large performance differences across signal modalities and domains.** To quantify these variations, we employ the Kappa coefficient, providing a normalized view of improvement over the naive baseline across all domains and input modalities. Specifically, to eliminate confounders arising from channel multiplicity, our modality-wise analysis is restricted to a subset of 81 single-channel tasks covering 16 distinct modalities. As illustrated in Fig. 6, the 95% confidence intervals reveal significant performance stratification, confirming that model proficiency varies across input modality and domain. Furthermore, the performance heatmaps across 16 models (Fig. 7) uncover a striking inter-model consistency: LLMs exhibit a synchronized competence profile, where domains accessible to one model are generally tractable for the entire cohort. This suggests a universal hierarchy of difficulty intrinsic to the health time-series domains themselves, rather than variance driven by specific model architectures.

> **Finding 3:** Performance degrades with temporal complexity, with a shared, model-agnostic difficulty ordering across domains and input modalities.

### 4.4. Performance Across LLM Families

**Models within the same family show strong consistency in task performance.** We quantify the alignment of reasoning behaviors within model families by conducting a Pearson correlation analysis on absolute task scores for the GPT, Qwen, and Gemini series. Results in Fig. 8 present a remarkable intra-family consistency: all three families demonstrate strong linear correlations. Models sharing an architectural lineage possess a synchronized competence profile, where tasks that are challenging for one variant prove consistently difficult for its counterparts. Interestingly, this behavioral alignment coexists with limited gains in absolute performance, as the regression lines closely follow $y = x$. This suggests that while architectural design governs the distribution of task difficulty, increased model

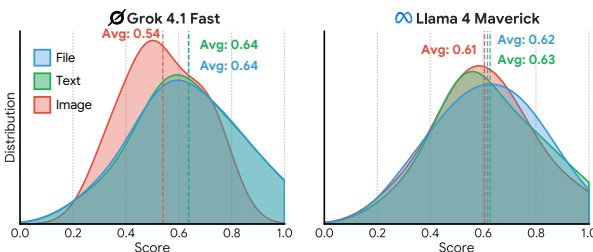

*Figure 9.* **Performance Comparison of Llama 4 Maverick and Grok 4.1 Fast across different input formats.**

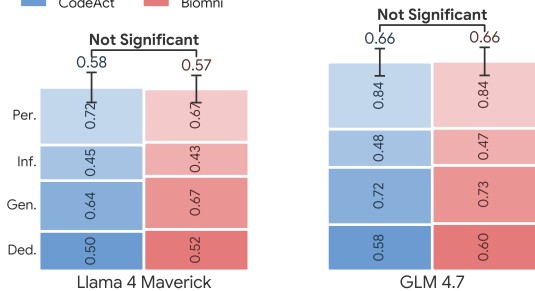

*Figure 10.* **Comparison between Biomni and CodeAct.**

scale or generation recency does not strictly guarantee superior reasoning capabilities in the health time-series domain.

> ***Finding 4:*** Models within the same architectural family exhibit highly consistent behavioral patterns, while scaling does not necessarily yield improved proficiency.

### 4.5. Does Input Format Affect Performance?

**Input format has only a moderate effect, with visual ingestion underperforming text-based formats.** We extend beyond CodeAct's file-access mechanism to assess the impact of alternative input formats, including textual and visual representations, on reasoning efficacy. To isolate the impact of format without confounding factors like context window exhaustion, we conduct this ablation on a curated subset of 10 short-sequence tasks ($L < 1000$, see Appendix C.4). Experiments using Llama 4 Maverick and Grok 4.1 Fast (Fig. 9) reveal a modality-dependent performance hierarchy. While direct text input and file access achieve near-parity with negligible score differentials, visual ingestion causes notable performance attenuation (e.g., Grok drops to 0.54). Despite this, the distributional morphology of the scores remains structurally analogous across all formats. This suggests that while visual encoding may lose fine-grained fidelity, the fundamental semantic processing of time-series data remains consistent regardless of the input modality.

> ***Finding 5:*** Input format mainly shifts absolute performance, while relative task difficulty remains consistent across formats.

### 4.6. Additional Studies

**More input information does not necessarily improve performance.** We examine a curated subset of 13 tasks (organized into 6 pairs) selected from our 110-task benchmark, spanning the sleep, surgery, metabolic, and emotion domains. Each pair shares identical settings, differing solely in whether auxiliary signals are provided to aid inference. Contrary to expectations, adding these signals often fails to improve and can even degrade performance relative to single-channel baselines. We identify two primary failure modes: contextual neglect, where agents disregard auxiliary data to rely exclusively on the target channel, and informational distraction, where complex multimodal context is misinterpreted as noise or induces redundant reasoning steps. Detailed breakdowns are available in Appendix C.5.

**Domain-specialist agents do not necessarily outperform generalist agents.** We investigate whether adopting a domain-specific agentic framework enhances performance by benchmarking against Biomni (Huang et al., 2025), a general-purpose biomedical agent. To ensure a rigorous comparison across disparate capability levels, we implemented Biomni using both GLM 4.7 (representing high-performing models) and Llama 4 Maverick (representing lower-tier models) as backbones. Counter-intuitively, we observe that the specialized Biomni architecture fails to yield statistically significant (Paired T-test) improvements over the standard CodeAct agent (Fig. 10). A granular analysis of agent trajectories (Appendix C.5) reveals the underlying cause: Biomni's retrieved action space, while comprehensive for tasks like gene prioritization or drug repurposing, suffers from toolset sparsity regarding health time-series analysis. Consequently, the agent rarely invokes its specialized library, effectively reverting to generic reasoning. This highlights a critical insight: broad biomedical domain expertise cannot compensate for the absence of modality-specific computational primitives in time-series processing.

## 5. Discussion

**Limitations.** A limitation of HEARTS is that we do not yet provide a systematic, task-agnostic evaluation of agent reasoning trajectories. While HEARTS measures final-task performance reliably, assessing the quality and faithfulness of intermediate steps remains difficult, and current approaches (e.g., LLM-as-a-judge scoring) are not reliable enough for absolute evaluation. In addition, many failures suggest that current agents lack domain prior knowledge needed for robust decisions on physiological signals; designing principled ways to incorporate such priors (e.g., via tools, constraints, or verified clinical rules) is an important direction.

**Conclusion.** We present HEARTS, a unified benchmark that evaluates LLMs on health time series beyond narrow an-

alytics and toward hierarchical reasoning. HEARTS spans 110 tasks across 12 healthcare domains and 20 signal modalities, enabling controlled evaluation of perception, inference, generation, and deduction. Our results show a consistent gap: models perform reasonably on basic signal perception but struggle on higher-level reasoning that requires long-horizon integration, causal thinking, and temporal directionality. Finally, HEARTS is released as a community-driven, living benchmark so the evaluation can expand with new datasets, tasks, and agents as the field progresses.

## Impact Statement

This paper introduces HEARTS, a unified benchmark for evaluating LLM agents on reasoning over health time series. By integrating 16 real-world datasets across 12 healthcare domains and 20 signal modalities, and by organizing 110 tasks into a hierarchical taxonomy (Perception, Inference, Generation, and Deduction), HEARTS enables controlled and reproducible assessment of model capabilities across diverse temporal scales, sampling frequencies, and clinical contexts. We expect HEARTS to support more rigorous development of health-oriented agents, improve comparability across methods, and help the community identify where current LLM-based systems succeed and where they fail, especially on long-horizon and multi-signal reasoning.

At the same time, benchmarking health time-series reasoning may raise safety, privacy, and fairness considerations. Physiological signals are sensitive, and dataset sharing, task design, and leaderboard reporting should follow strong de-identification and data governance practices to reduce re-identification and misuse risks. Model performance can vary across populations, devices, protocols, and care settings due to dataset shift and bias. As a result, benchmark performance should not be interpreted as evidence of clinical reliability. For these reasons, HEARTS should be treated as a research and evaluation resource rather than a clinical decision system; any downstream deployment would require prospective validation, robust failure handling, and compliance with privacy and medical device regulations.

## Acknowledgments

We gratefully acknowledge the support by Amazon, UCLA DataX, Anthropic, OpenAI, and Google Cloud. Any opinions, findings, conclusions, or recommendations expressed in this material are those of the author(s) and do not necessarily reflect the views of the funders.

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

# A. Details of HEARTS Design

## A.1. Data Statistics

We provide a statistical breakdown of the datasets curated in HEARTS in Table 3. To characterize the heterogeneity of the benchmark, we detail the specific signal modalities and their corresponding sampling frequencies for each dataset, alongside the number of samples utilized in HEARTS. To ensure transparency and support community-driven reproducibility, we detail the access protocols for each dataset.

*Table 3.* **Statistical summary of datasets in HEARTS.**

| Dataset | Domain | Modality used in HEARTS (Frequency) | # Test Samples | Data Accessibility |
|---|---|---|---|---|
| Capture24 | Motion | IMU (100Hz) | 1204 | Open Source |
| PAMAP2 | Motion | IMU (100Hz) | 106 | Open Source |
| Shanghai Diabetes | Metabolic | CGM (per minute) | 232 | Open Source |
| CGMacros | Metabolic | CGM (per minute), HR (per minute), Annotation | 2333 | Open Source |
| VitalDB | Surgery | MBP (10Hz/0.5Hz), EEG (100Hz), ECG (100Hz), PPG (100Hz), Annotation | 2000 | Open Source |
| SHHS | Sleep | ECG (128Hz), EEG (64Hz), EOG (64Hz), Airflow (8Hz), Thoracic (8Hz), HR (1Hz), Annotation | 4799 | Restricted Access |
| Harespod | Respiration | Respiration (100Hz), $SpO_2$ (100Hz), HR (1Hz) | 632 | Open Source |
| PhyMER | Emotion | BVP (64Hz), EDA (4Hz), TEMP (4Hz), HR (1Hz) | 1830 | Restricted Access |
| PERG-IOBA | Ophthalmology | PERG (1700Hz) | 870 | Open Source |
| GazeBase | Eye Movement | Eye Tracking (1000Hz) | 988 | Open Source |
| GLOBEM | Behavior | Aggregated Data (per day) | 1140 | Restricted Access |
| Bridge2AI-voice | Speech | Audio (100Hz) | 1600 | Restricted Access |
| VCTK | Speech | Audio (16000Hz) | 200 | Open Source |
| GrabMyo | Gesture | EMG (2048Hz) | 400 | Open Source |
| CoughVID | COVID Cough | Audio (48000Hz) | 892 | Open Source |
| Coswara | COVID Cough | Audio (48000Hz) | 1000 | Open Source |

## A.2. Task Design Detail

Below are the specifications of the task design for all 110 tasks included in HEARTS. For each task, we report a unified schema including the task name, input and output formats, evaluation metrics, and the temporal granularity. For tasks with multiple input channels, the frequency and sequence length are determined by the maximum value across all channels.

### A.2.1. CAPTURE24

**Perception - Feature Extraction**

• Step Count Calculation
  Inputs: 120 sec IMU signal recorded during walking | Output: Step count during walking
  Sequence Length: 10K-100K | Frequency: 100 Hz
  Metric: Accuracy

**Inference - Physiological Classification**

• Activity Classification
  Inputs: 150 sec 3-axis signal | Output: Activity type
  Sequence Length: 10K-100K | Frequency: 100 Hz
  Metric: Accuracy

• Activity Transition Recognition
  Inputs: 150 sec 3-axis signal | Output: Activity transition sequence
  Sequence Length: 10K-100K | Frequency: 100 Hz
  Metric: Accuracy

**Generation - Signal Imputation**

- 1-axis Signal Imputation
  Inputs: 150 sec 3-axis IMU signal with 30 sec missing on z axis | Output: 30 sec missing-axis signal
  Sequence Length: 10K-100K | Frequency: 100 Hz
  Metric: sMAPE score

- 3-axis Signal Imputation
  Inputs: 150 sec 3-axis IMU signal with 30 sec all three axis missing | Output: 30 sec missing 3-axis signal
  Sequence Length: 10K-100K | Frequency: 100 Hz
  Metric: sMAPE score

**Generation - Future Forecasting**

- 3-axis Signal Forecasting
  Inputs: 120 sec 3-axis IMU signal | Output: next 30 sec signal
  Sequence Length: 10K-100K | Frequency: 100 Hz
  Metric: sMAPE score

**Deduction - Temporal Ordering**

- Day and Night Signal Ordering
  Inputs: 3 hour night data & 3 hour day data | Output: Which one is night data?
  Sequence Length: >1M | Frequency: 100 Hz
  Metric: Accuracy

A.2.2. PAMAP2

**Inference - Event Localization**

- Activity Localization
  Inputs: 5x activity length 3-axis IMU signal | Output: Activity's start time and end time
  Sequence Length: 100K-1M | Frequency: 100 Hz
  Metric: IoU

A.2.3. SHANGHAI DIABETES

**Inference - Subject Profiling**

- Cross-Subject Diabetes Type Comparison
  Inputs: Full length CGM from two different subjects | Output: Which subject has type 1 diabetes and which has type 2 diabetes
  Sequence Length: 1K-10K | Frequency: Per minute
  Metric: Accuracy

- Diabetes Type Classification
  Inputs: Full length CGM data from a diabetic subject | Output: This subject has type 1 diabetes or type 2 diabetes
  Sequence Length: 1K-10K | Frequency: Per minute
  Metric: Accuracy

A.2.4. CGMACROS

**Perception - Statistical Calculation**

- CGM Time in Range Calculation
  Inputs: entire CGM & normal CGM range
  Output: percentage of time below normal CGM range & percentage of time above normal CGMrange
  Sequence Length: 1K-10K | Frequency: per minute
  Metric: sMAPE score

**Perception - Feature Extraction**

- Postprandial CGM iAUC Calculation
  Inputs: 2hr CGM after meal starts | Output: iAUC value
  Sequence Length: <100 | Frequency: per minute
  Metric: sMAPE score

**Inference - Event Localization**

- Meal Time Localization
  Inputs: 2hr CGM window | Output: meal start timestamp
  Sequence Length: <100 | Frequency: per minute
  Metric: sMAPE score

**Inference - Physiological Classification**

- Meal Image Classification from CGM
  Inputs: 4hr CGM time window with meal event & 4 options of meal image
  Output: identify the image corresponding to the subject's meal
  Sequence Length: <100 | Frequency: per minute
  Metric: Accuracy

**Inference - Subject Profiling**

- A1c Prediction
  Inputs: Full length CGM | Output: Select A1c range (3 options)
  Sequence Length: 1K-10K | Frequency: Per minute
  Metric: Accuracy

- Fasting GLU Prediction
  Inputs: Full length CGM | Output: Predict fasting GLU
  Sequence Length: 1K-10K | Frequency: Per minute
  Metric: sMAPE score

- Postprandial CGM Response Comparison
  Inputs: 4-hour CGM data following similar meals from two different subjects (carbohydrates and calories differ by <
  10%)
  Output: Which subject exhibits normal glucose regulation
  Sequence Length: <100 | Frequency: Per minute
  Metric: Accuracy

**Generation - Signal Imputation**

- CGM Imputation
  Inputs: 2hr CGM window with 30 minute missing | Output: Missing 30 min CGM
  Sequence Length: <100 | Frequency: Per minute
  Metric: sMAPE score

- CGM Imputation with Activity Calories Trajectory
  Inputs: 2hr CGM window with 30 minute missing & activity calories info within that window
  Output: Missing 30 min CGM
  Sequence Length: <100 | Frequency: Per minute
  Metric: sMAPE score

- CGM Imputation with HR Trajectory
  Inputs: 2hr CGM window with 30 minute missing & HR within that window | Output: Missing 30 min CGM
  Sequence Length: <100 | Frequency: Per minute
  Metric: sMAPE score

**Generation - Future Forecasting**

- CGM Forecasting
  Inputs: 1hr CGM before meal | Output: Next 30 min CGM
  Sequence Length: <100 | Frequency: Per minute
  Metric: sMAPE score

- CGM Forecasting with Meal Information
  Inputs: 1hr CGM before meal & meal info | Output: Next 30 min CGM
  Sequence Length: <100 | Frequency: Per minute
  Metric: sMAPE score

- CGM Forecasting with History CGM and Meal Info
  Inputs: Previous 3 days' CGM & meal of same subject as reference, and 1hr CGM before meal
  Output: Next 30 min CGM
  Sequence Length: 100-1K | Frequency: Per minute
  Metric: sMAPE score

- CGM Forecasting with Current Meal Information and History CGM
  Inputs: Previous 3 days' CGM & meal of same subject as reference, and 1hr CGM before meal & meal info
  Output: Next 30 min CGM
  Sequence Length: 100-1K | Frequency: Per minute
  Metric: sMAPE score

### A.2.5. VITALDB

**Perception - Statistical Calculation**

- Mean arterial pressure (MBP) Time in Range 70-100
  Inputs: Full length blood Pressure (Solar8000/ART_MBP) | Output: Percentage of time where blood pressure within normal range
  Sequence Length: 100K-1M | Frequency: 0.5 Hz
  Metric: sMAPE score

**Inference - Event Localization**

- Anesthesia Range Localization
  Inputs: Full length annotation (BIS/EEG1_WAV & BIS/EEG2_WAV) | Output: Anesthesia time range [start time, end time]
  Sequence Length: 100K-1M | Frequency: 128 Hz
  Metric: IoU

**Inference - Physiological Classification**

- Hypotension Event Classification with MBP
  Inputs: 600 sec blood pressure (Solar8000/ART_MBP) | Output: If next 5 minutes will have hypotension events
  Sequence Length: <100 | Frequency: 0.5 Hz
  Metric: Accuracy

- Hypotension Event Classification with PPG
  Inputs: 600 sec PPG (SNUADC/PLETH) | Output: If next 5 minutes will have hypotension events
  Sequence Length: 10K-100K | Frequency: 100 Hz
  Metric: Accuracy

**Inference - Subject Profiling**

- MINS (myocardial injury after non-cardiac surgery) Prediction
  Inputs: Full length Blood Pressure (Solar8000/ART_MBP) | Output: White blood cell count from lab result
  Sequence Length: 100-1K | Frequency: 1 Hz
  Metric: Accuracy

**Generation - Future Forecasting**

- MBP Forecasting
  Inputs: 120 sec blood pressure (Solar8000/ART_MBP) | Output: Next 30 sec Blood Pressure
  Sequence Length: <100 | Frequency: 0.5 Hz
  Metric: sMAPE score

- MBP Forecasting with Infusion Info
  Inputs: 120 sec blood pressure (Solar8000/ART_MBP) + All infusion data at this time window | Output: 30s MBP data
  Sequence Length: <100 | Frequency: 0.5 Hz
  Metric: sMAPE score

## Generation - Cross-modal Translation

- Drug Infusion Series to Depth of Anesthesia (BIS) Translation
  Inputs: Drug infusion series (Orchestra/PPF20_VOL & Orchestra/RFTN20_VOL & Orchestra/PPF20_CE & Orchestra/RFTN20_CE)
  Output: Corresponding BIS series
  Sequence Length: 100-1K | Frequency: 10s intervals
  Metric: sMAPE score

- EEG to Bispectral Index (BIS) Translation
  Inputs: 360 sec EEG (BIS/EEG1_WAV & BIS/EEG2_WAV) | Output: Corresponding BIS series
  Sequence Length: 100K-1M | Frequency: 128 Hz
  Metric: sMAPE score

- PPG and ECG to Blood Pressure Translation
  Inputs: 360 sec ECG (SNUADC/ECG_II) + PPG (SNUADC/PLETH)
  Output: Corresponding blood pressure series
  Sequence Length: 100-1K | Frequency: 100 Hz
  Metric: sMAPE score

## A.2.6. SHHS

### Perception - Statistical Calculation

- REM Latency Calculation
  Inputs: Whole night sleepstage annotations | Output: REM latency value
  Sequence Length: 100-1K | Frequency: Per 30sec epoch
  Metric: sMAPE score

- Sleep AHI Calculation
  Inputs: Whole night sleepstage annotations and sleep event calculations | Output: Sleep AHI value
  Sequence Length: 100-1K | Frequency: Per 30sec epoch
  Metric: sMAPE score

### Perception - Feature Extraction

- Bandpower Calculation
  Inputs: EEG C3/A2 in various length | Output: Bandpower values
  Sequence Length: 100K-1M | Frequency: 64 Hz
  Metric: sMAPE score

- Sleep Efficiency Calculation
  Inputs: Whole night sleepstage annotations | Output: Sleep efficiency value
  Sequence Length: 100-1K | Frequency: 30s epochs
  Metric: sMAPE score

### Inference - Event Localization

- Hypopnea Detection
  Inputs: Airflow & Thoracic with 5x event duration | Output: List of [start time, end time]
  Sequence Length: 1K-10K | Frequency: 8 Hz

Metric: IoU

- Arousal Detection (EEG)
  Inputs: EEG C3/A2 with 5x event duration | Output: List of [start time, end time]
  Sequence Length: 1K-10K | Frequency: 64 Hz
  Metric: IoU

- Arousal Detection (EOG)
  Inputs: EOG-E2A1 & EOG-E1A2 with 5x event duration | Output: List of [start time, end time]
  Sequence Length: 1K-10K | Frequency: 64 Hz
  Metric: IoU

**Inference - Physiological Classification**

- REM/NREM Classification
  Inputs: EOG-E2A1 & EOG-E1A2 within REM/NREM sleep | Output: REM or NREM
  Sequence Length: 100K-1M | Frequency: 64 Hz
  Metric: Accuracy

- Sleep Stage Classification
  Inputs: EEG C3/A2 of a specific sleep stage & reference bandpower of different sleep stages of same subject
  Output: Sleep stage
  Sequence Length: 1K-10K | Frequency: 64 Hz
  Metric: Accuracy

- Sleep Stage Transition Recognition
  Inputs: EEG C3/A2 from two consecutive sleep stages & bandpower of different sleep stages of same subject
  Output: Sleep stage transition sequence (e.g., N1→N2)
  Sequence Length: 1K-10K | Frequency: 64 Hz
  Metric: Accuracy

**Inference - Subject Profiling**

- Visit Level Ordering
  Inputs: ECG & Airflow & EEG C3/A2 from different visit | Output: Which one is ahead of another
  Sequence Length: >1M | Frequency: 128 Hz
  Metric: Accuracy

- Half Night Level Ordering
  Inputs: ECG & Airflow & EEG C3/A2 from first half / second half of the whole night signal
  Output: Which one is ahead of another
  Sequence Length: >1M | Frequency: 128 Hz
  Metric: Accuracy

- Episode Level Ordering
  Inputs: ECG & Airflow & EEG C3/A2 from different neighboring episodes
  Output: Which one is ahead of another
  Sequence Length: 1K-10K | Frequency: 128 Hz
  Metric: Accuracy

**Generation - Signal Imputation**

- Single-channel Imputation
  Inputs: 150s ECG signal (missing interval: 60-90s) | Output: Missing ECG signal
  Sequence Length: 10K-100K | Frequency: 128 Hz
  Metric: sMAPE score

- Conditional Imputation
  Inputs: 150 sec ECG channel (missing interval: 60-90s) & 150 sec Airflow without missing
  Output: Missing ECG signal
  Sequence Length: 10K-100K | Frequency: 128 Hz
  Metric: sMAPE score

**Generation - Future Forecasting**

- Single-channel Forecasting
  Inputs: 120 sec ECG channel | Output: Next 30sec ECG signal
  Sequence Length: 10K-100K | Frequency: 128 Hz
  Metric: sMAPE score

- Conditional Forecasting
  Inputs: 120 sec ECG channel & 120 sec Airflow signal | Output: Next 30sec ECG signal
  Sequence Length: 10K-100K | Frequency: 128 Hz
  Metric: sMAPE score

**Generation - Cross-modal Translation**

- Cross-channel Translation (EEG C3/A2 to EEG C4/A1)
  Inputs: 120 sec EEG C3/A2 channel | Output: Corresponding 120 sec EEG C4/A1 channel
  Sequence Length: 1K-10K | Frequency: 64 Hz
  Metric: sMAPE score

- Cross-channel Translation (ECG to HR)
  Inputs: 120 sec ECG | Output: Corresponding 120 sec HR
  Sequence Length: 1K-10K | Frequency: 128 Hz
  Metric: sMAPE score

**Deduction - Temporal Ordering**

- Smoker Classification
  Inputs: Whole night Airflow & ECG & EEG C3/A2 | Output: If this patient is a smoker (Yes/No)
  Sequence Length: >1M | Frequency: 128 Hz
  Metric: Accuracy

- Atrial Fibrillation (AF) Classification
  Inputs: Whole night ECG | Output: If this patient has AF (Yes/No)
  Sequence Length: >1M | Frequency: 128 Hz
  Metric: Accuracy

- Cardiovascular Disease (CVD) Death Prediction
  Inputs: Whole night ECG & Airflow & EEG C3/A2 | Output: If this patient will have fatal CVD (Yes/No)
  Sequence Length: >1M | Frequency: 128 Hz
  Metric: Accuracy

- Stroke Prediction
  Inputs: Whole night ECG & Airflow & EEG C3/A2 | Output: If this patient will have stroke (Yes/No)
  Sequence Length: >1M | Frequency: 128 Hz
  Metric: Accuracy

**Deduction - Trajectory Analysis**

- BMI Comparison Between Visit
  Inputs: Whole night ECG & Airflow & EEG C3/A2 from 2 visits of same subject
  Output: Which visit has higher BMI? (visit1 / visit2)
  Sequence Length: >1M | Frequency: 128 Hz
  Metric: Accuracy

A.2.7. HARESPOD

**Inference - Physiological Classification**

- Altitude Ranking with Respiration
  Inputs: Given 3 segments of 5 min respiration signals, ranking altitude | Output: Arrange from low to high
  Sequence Length: 10K-100K | Frequency: 100 Hz

Metric: Accuracy

- Altitude Ranking with SpO2
  Inputs: Given 3 segments of 5 min SpO2 signals, ranking altitude | Output: Arrange from low to high
  Sequence Length: 10K-100K | Frequency: 100 Hz
  Metric: Accuracy

**Inference - Subject Profiling**

- Respiration and SpO2 Pairing
  Inputs: Given 2 sets of 10min SpO2 and respiration signal
  Output: Pairing corresponding SpO2 signal and respiration signal
  Sequence Length: 10K-100K | Frequency: 100 Hz
  Metric: Accuracy

- Respiration and HR Pairing
  Inputs: Given 2 sets 10min HR and respiration signal
  Output: Pairing corresponding respiration signal and heart rate signal
  Sequence Length: 10K-100K | Frequency: 100 Hz
  Metric: Accuracy

A.2.8. PHYMER

**Inference - Physiological Classification**

- Emotion Type Classification
  Inputs: All E4 signals & self-reported arousal and valence value | Output: Emotion Type
  Sequence Length: 100K-1M | Frequency: 64 Hz
  Metric: Accuracy

- Emotion Type Classification with Only EDA Input
  Inputs: EDA & self-reported arousal and valence value | Output: Emotion Type
  Sequence Length: 1K-10K | Frequency: 4 Hz
  Metric: Accuracy

**Inference - Subject Profiling**

- Cross Subject Arousal Ranking
  Inputs: All E4 signals of two subjects watching same video & emotion label of each subject
  Output: Which subject has higher arousal?
  Sequence Length: 100K-1M | Frequency: 64 Hz
  Metric: Accuracy

- Inter-subject Emotion Recognition
  Inputs: All E4 signals of one subject watching different video & emotion type of the two sessions
  Output: Identify correct emotion (e.g., which one's emotion is angry)
  Sequence Length: 100K-1M | Frequency: 64 Hz
  Metric: Accuracy

- Personality Analysis
  Inputs: All video labels for one subject | Output: Big-5 Personality
  Sequence Length: 100K-1M | Frequency: 64 Hz
  Metric: Accuracy

**Generation - Signal Imputation**

- Single-channel Imputation
  Inputs: HR with 20 sec missing | Output: 20sec missing HR
  Sequence Length: <100 | Frequency: 1 Hz
  Metric: sMAPE score

- Conditional Imputation

Inputs: HR with 20 sec missing & corresponding BVP, EDA, TEMP in that time range & emotion labels
Output: 20 sec missing HR
Sequence Length: 1K-10K | Frequency: 64 Hz
Metric: sMAPE score

## Generation - Future Forecasting

- Single-channel Forecasting
  Inputs: HR | Output: Next 20 sec HR
  Sequence Length: <100 | Frequency: 1 Hz
  Metric: sMAPE score

- Conditional Forecasting
  Inputs: All E4 signals & emotion labels | Output: Next 20 sec HR
  Sequence Length: 1K-10K | Frequency: 1 Hz
  Metric: sMAPE score

## Generation - Cross-modal Translation

- Cross-channel Translation
  Inputs: Full length BVP & EDA & TEMP & Emotion | Output: Corresponding HR
  Sequence Length: 1K-10K | Frequency: 64 Hz
  Metric: sMAPE score

A.2.9. PERG-IOBA

## Perception - Feature Extraction

- N35, P50, N95 Feature Extraction
  Inputs: 1 PERG signal recording | Output: N35, P50 and N95 amplitude
  Sequence Length: 100-1K | Frequency: 1700 Hz
  Metric: sMAPE score

## Inference - Subject Profiling

- Eye Health Status Classification
  Inputs: 1 PERG signal recording | Output: whether the subject had eye disease
  Sequence Length: 100-1K | Frequency: 1700 Hz
  Metric: Accuracy

- Eye Disease Type Classification
  Inputs: 1 PERG signal recording | Output: Out of 4 choices, which is most likely the disease that the subject had
  Sequence Length: 100-1K | Frequency: 1700 Hz
  Metric: Accuracy

- Eye Disease Type Classification with Patient's Meta Information
  Inputs: 1 PERG signal recording and subject's age, visual acuity and sex information | Output: Out of 4 choices, which is most likely the disease that the subject had
  Sequence Length: 100-1K | Frequency: 1700 Hz
  Metric: Accuracy

- Disease Differentiation: between Macular Disease & Optic Nerve Disease & Normal
  Inputs: 1 PERG signal recording | Output: Differentiate 2 types of disease and normal status
  Sequence Length: 100-1K | Frequency: 1700 Hz
  Metric: Accuracy

A.2.10. GAZEBASE

## Inference - Event Localization

- Fixation Point Localization

Inputs: full length eye tracking data from fixation task | Output: Coordinate of the fixation point
Sequence Length: 10K-100K | Frequency: 1000 Hz
Metric: sMAPE score

**Inference - Physiological Classification**

- Eye-tracking Task Classification
  Inputs: 20 sec eye tracking data segment | Output: Task type
  Sequence Length: 10K-100K | Frequency: 1000 Hz
  Metric: Accuracy

**Generation - Signal Imputation**

- Text Reading Imputation
  Inputs: 50 sec eye tracking sequence from reading task with 10 sec missing | Output: Missing 10 sec data
  Sequence Length: 10K-100K | Frequency: 1000 Hz
  Metric: sMAPE score

**Generation - Future Forecasting**

- Horizontal Saccade Track Forecasting
  Inputs: 90 sec eye tracking sequence | Output: Next 10sec data
  Sequence Length: 10K-100K | Frequency: 1000 Hz
  Metric: sMAPE score

**Deduction - Temporal Ordering**

- Reading Sequence Recognition
  Inputs: First & middle & last 10 sec of one eye tracking sequence from reading task | Output: Sequence of these 10sec clips (e.g. A→B→C)
  Sequence Length: 10K-100K | Frequency: 1000 Hz
  Metric: Accuracy

A.2.11. GLOBEM

**Perception - Feature Extraction**

- Location Entropy Extraction
  Inputs: 2 subject's 7-days location log | Output: Which one has higher significant location entropy?
  Sequence Length: <100 | Frequency: Daily aggregates
  Metric: Accuracy

**Inference - Event Localization**

- Peak Stress Week Localization
  Inputs: 10 weeks sleep & location & screen& step data | Output: Which week is the 'peak stress week'?
  Sequence Length: <100 | Frequency: Daily aggregates
  Metric: Accuracy

**Inference - Physiological Classification**

- Depression Trajectory Classification
  Inputs: 10 weeks data | Output: Whether the subject has depression at the end of term?
  Sequence Length: <100 | Frequency: Daily aggregates
  Metric: Accuracy

- COVID Year Recognition
  Inputs: General behavior patterns (travel distance, step count) of different years
  Output: which one is from post COVID?
  Sequence Length: <100 | Frequency: Daily aggregates

Metric: Accuracy

## Generation - Future Forecasting

- Step Count Forecasting
  Inputs: Given past 14 days step count | Output: Next 1day's step count
  Sequence Length: <100 | Frequency: Daily aggregates
  Metric: sMAPE score

## Inference - Subject Profiling

- Circadian Routine Comparison
  Inputs: 2 subject's ten week sleep log (get up / go to sleep time), location log and step count
  Output: Which one has higher circadian routine score (circdnrtn)?
  Sequence Length: <100 | Frequency: Daily aggregates
  Metric: Accuracy

A.2.12. B**RIDGE**2AI-**VOICE**

## Perception - Feature Extraction

- F0 Range Extraction
  Inputs: 1 spectrogram of a recording | Output: Value of mean F0 (fundamental frequency)
  Sequence Length: 10K - 100K | Frequency: 100 Hz
  Metric: sMAPE score

- Harmonics to Noise Ratio (HNR) Extraction
  Inputs: 1 spectrogram of a recording | Output: Value of mean HNR
  Sequence Length: 10K - 100K | Frequency: 100 Hz
  Metric: sMAPE score

- Shimmer Extraction
  Inputs: 1 spectrogram of a recording | Output: Value of mean shimmer
  Sequence Length: 10K - 100K | Frequency: 100 Hz
  Metric: sMAPE score

- Jitter Extraction
  Inputs: 1 spectrogram of a recording | Output: Value of mean jitter
  Sequence Length: 10K - 100K | Frequency: 100 Hz
  Metric: sMAPE score

## Perception - Feature Extraction

- Articulation Rate Calculation
  Inputs: 1 spectrogram of a recording | Output: Articulation rate
  Sequence Length: 10K - 100K | Frequency: 100 Hz
  Metric: sMAPE score

## Inference - Physiological Classification

- Cross-task Voice Comparison
  Inputs: Spectrograms of 2 recordings | Output: Matching transcripts with the spectrogram
  Sequence Length: 10K - 100K | Frequency: 100 Hz
  Metric: Accuracy

## Inference - Subject Profiling

- Parkinson's Diagnosis
  Inputs: 1 spectrogram of a recording | Output: Whether the subject has Parkinson's disease
  Sequence Length: 10K - 100K | Frequency: 100 Hz
  Metric: Accuracy

**Deduction - Temporal Ordering**

- Reversed Signal Detection
  Inputs: 1 spectrogram of a recording | Output: Whether the spectrogram was reversed
  Sequence Length: 10K - 100K | Frequency: 100 Hz
  Metric: Accuracy

### A.2.13. VCTK

**Deduction - Temporal Ordering**

- Reversed Signal Detection
  Inputs: 1s audio waveform | Output: Whether the waveform was reversed
  Sequence Length: 100K - 1M | Frequency: 48000 Hz
  Metric: Accuracy

### A.2.14. GRABMYO

**Inference - Physiological Classification**

- Gesture Classification with Reference
  Inputs: 28 channels of EMG signal and reference EMG signals of all gestures | Output: Gesture prediction
  Sequence Length: 1k - 10K | Frequency: 2048 Hz
  Metric: Accuracy

- Subject Identification
  Inputs: 28 channels of EMG signal and reference EMG signals of same gesture from a subject recorded in different session | Output: Whether EMG signal was from the same subject
  Sequence Length: 1k - 10K | Frequency: 2048 Hz
  Metric: Accuracy

### A.2.15. COUGHVID

**Perception - Feature Extraction**

- MFCC Mean & STD Calculation
  Inputs: 1-12 seconds audio signal | Output: Mean and standard deviation of MFCC values
  Sequence Length: 100K-1M | Frequency: 48K Hz
  Metric: sMAPE score

**Inference - Physiological Classification**

- Health Status Classification
  Inputs: 1-12 seconds audio signal | Output: Whether the subject is healthy
  Sequence Length: 100K-1M | Frequency: 48K Hz
  Metric: Accuracy

- COVID Status Classification
  Inputs: 1-12 seconds audio signal | Output: COVID-19 positive/negative prediction
  Sequence Length: 100K-1M | Frequency: 48K Hz
  Metric: Accuracy

- Diagnosis Classification
  Inputs: 1-12 seconds audio signal | Output: Choose one most likely diagnosis out of 5 choices
  Sequence Length: 100K-1M | Frequency: 48K Hz
  Metric: Accuracy

**Inference - Subject Profiling**

- Cough Detection with Good Quality Samples
  Inputs: 1-12 seconds audio signal | Output: Binary detection of cough presence

Sequence Length: 100K-1M | Frequency: 48K Hz
Metric: Accuracy

- Cough Detection with Poor Quality Samples
  Inputs: 1-12 seconds audio signal | Output: Binary detection of cough presence
  Sequence Length: 100K-1M | Frequency: 48K Hz
  Metric: Accuracy

### A.2.16. COSWARA

**Inference - Physiological Classification**

- Audio Type Classification
  Inputs: Audio | Output: Audio type (breathing/speech/cough)
  Sequence Length: >1M | Frequency: 48 kHz
  Metric: Accuracy

- Diagnosis Classification with Speech Audio
  Inputs: Speech audio | Output: Healthy or COVID positive
  Sequence Length: >1M | Frequency: 48 kHz
  Metric: Accuracy

**Inference - Subject Profiling**

- Diagnosis Classification with Cough Audio
  Inputs: Cough audio | Output: Healthy or COVID positive
  Sequence Length: 100K-1M | Frequency: 48 kHz
  Metric: Accuracy

- Diagnosis Classification with Cough Audio and Symptom Information
  Inputs: cough audio & symptom info | Output: Healthy or COVID positive
  Sequence Length: >1M | Frequency: 48 kHz
  Metric: Accuracy

- Diagnosis Classification with Symptom Information Only
  Inputs: Symptom info | Output: Healthy or COVID positive
  Sequence Length: NA | Frequency: N/A
  Metric: Accuracy

### A.3. Example Prompt

Here we showcase the example prompts given to the LLM for each task category.

### A.3.1. PERCEPTION

---

**Satistical Calculation**

**Prompt:** The continuous glucose monitors (CGM) data for this subject is provided in `input/data.csv`. There are two columns in this csv file: one is timestamp containing the time of each reading, and the other column `Libre GL` contains glucose values (mg/dL).
Calculate percentage of time CGM is below and above normal range ($70 - 180$ mg/dL). Please calculate and output your final answer as a JSON object without any other text in the following format:
```
{
"below":  [float, percentage of time CGM < 70 mg/dL],
"above":  [float, percentage of time CGM > 180 mg/dL]
}
```

---

## Feature Extraction

**Prompt:** One EEG C3/A2 raw signal is saved in the file `input/data.npy`. Its sampling frequency is $64$ Hz. Calculate the relative bandpower of the signal in frequency range $0.5 - 40$ Hz.

Please use Welch's method with `nperseg=1024` to estimate the power spectral density, and use Simpson's method to integrate the power spectral density. Please output your final answer in the following JSON format without any other text:

```
{
"delta":  [ratio of delta band (0.5-4 Hz)],
"theta":  [ratio of theta band (4-8 Hz)],
"alpha":  [ratio of alpha band (8-12 Hz)],
"sigma":  [ratio of sigma band (12-16 Hz)],
"beta":   [ratio of beta band (16-30 Hz)],
"gamma":  [ratio of gamma band (30-40 Hz)]
}
```

## A.3.2. INFERENCE

## Event Localization

**Prompt:** The AF (Air Flow) and THX (Thoracic respiratory) signals are provided at `input/AF.npy` and `input/THX.npy`. Both signals are sampled at $8$Hz, and this time window contains hypopnea events.

Please analyze the signals and output your final answer of a list of hypopnea events (start and end timestamps) in the following JSON format without any other text:

```
{
"start":  [float, start timestamp in seconds],
"end":    [float, end timestamp in seconds]
}
```

## Physiological Classification

**Prompt:** There are two EOG signals saved as numpy arrays: `input/EOG_E2_A1.npy` (EOG channel E2-A1), and `input/EOG_E1_A2.npy` (EOG channel E1-A2). Both signals are sampled at $64$Hz.

Please analyze the signals and determine whether this is REM sleep or NREM sleep in JSON format. Please output your final result in a JSON object without any other text:

```
{
"stage":  [NR|R]
}
```

---

**Subject Profiling**

**Prompt:** A PERG (Pattern Electroretinogram) recording from both eyes is provided. The right eye signal is in `input/re_signal.csv` and left eye signal is in `input/le_signal.csv`. Each file contains two columns: `time_ms` (time in milliseconds) and `amplitude` (signal amplitude in $\mu V$).
Your task is to analyze the PERG recording and determine if the subject has normal eyes or has disease on eye.
Possible classifications:
- `"normal"`: The subject has normal eyes
- `"disease"`: The subject has disease on eyes
Based on your analysis of the waveform characteristics, choose the most appropriate classification.
Output your answer in JSON format:
```
{
"eye status":  [your classification]
}
```

---

A.3.3. GENERATION

**Signal Imputation**

**Prompt:** The ECG signal with missing values is provided at `input/ECG_missing.npy`. The signal is a 128 Hz time series of total length 150 seconds, where missing values are represented as 0. Please fill in the missing values in the ECG signal. Note that the missing values are consecutive and correspond to a 30-second segment within the 150-second window.
Save your imputed signal in a Python dictionary format with the following structure:
```
{
"ECG":  [your imputed signal as a NumPy array]
}
```
Then, save this dictionary to the output path `output/imputed_signal.npz`. The numpy array should only contain the imputed values for the missing segments, while the observed values should not included. Your array should have the shape (3840,). Ensure that your imputed values are realistic and consistent with the observed data.

---

**Future Forecasting**

**Prompt:** The ECG signal for the past 120 seconds is provided at `input/ECG.npy`. The signal is a 128 Hz time series. Please forecast the next 30 seconds of the ECG signal based on the provided data.
Save your forecasted signal in a Python dictionary format with the following structure:
```
{
"ECG":  [your forecasted signal as a NumPy array]
}
```
Then, save this dictionary to the output path `output/forecasted-signal.npz`. The numpy array should have the shape (3840,). Ensure that your forecasted values are realistic and consistent with the observed data.

---

**Cross-modal Translation**

**Prompt:** The EEG signal from channel EEG C3/A2 for the past 120 seconds is provided at `input/EEG_C3_A2.npy`. The signal is a 64 Hz time series.

Please translate this signal to the corresponding EEG C4/A1 channel for the same time period. The EEG C4/A1 channel is also a 64 Hz time series.

Save your translated signal in a Python dictionary format with the following structure:

{

`"EEG_C4_A1":  [your translated signal as a NumPy array]`

}

Then, save this dictionary to the output path `output/translated_signal.npz`. The numpy array should have the shape (7680,). Ensure that your translated values are realistic and consistent with the observed data.

### A.3.4. DEDUCTION

**Temporal Ordering**

**Prompt:** Two sets of 3-hour 3-axis accelerometer signals are provided in the format `input/[axis]_1.npy` and `input/[axis]_2.npy` where `axis` is x, y, or z. The signals are sampled at 100 Hz. One set is from daytime (11:00 AM - 2:00 PM) and the other is from nighttime (12:00 AM - 3:00 AM).

Determine which set of signals is from daytime and which is from nighttime. Output your final answer as a JSON object without any other text, in the following format:

{

`"is_daytime_first":  [1|0]`

}

where 0 means `input/[axis]_1.npy` is from daytime and `input/[axis]_2.npy` is from nighttime, 1 means `input/[axis]_2.npy` is from daytime and `input/[axis]_1.npy` is from nighttime.

**Trajectory Analysis**

**Prompt:** You are given ECG, AF, EEG C3/A2 signals from two visits of the same subject. ECG represents Electrocardiogram; AF represents Airflow; EEG C3/A2 represents Electroencephalogram. The signals are provided as 1D numpy arrays in `input/[ch]_visit1.npy` and `input/[ch]_visit2.npy`, where `[ch]` is one of [`ECG, AF, EEG_C3_A2`]. Your task is to predict which visit has higher BMI. Output your final answer as a JSON object without any other text, in the following format:

{

`"higher_bmi":  ["visit1"|"visit2"]`

}

### A.4. Living Ecosystem

Unlike traditional benchmarks that function as static snapshots of performance, our framework establishes a "Living Ecosystem" designed for continuous evolution and community-driven expansion. At the core of this ecosystem lies a rigorous decoupled architecture, exemplified by the separation of the `ExperimentBase` and `AgentBase` abstract classes. By enforcing a unified API through Python's ABC (Abstract Base Class) structures, we have created a standardized interface that allows researchers to inject new reasoning tasks or integrate novel agentic frameworks without altering the underlying evaluation engine. This modularity transforms the benchmark from a fixed dataset into a dynamic scaffolding where the "Task" and the "Solver" are interchangeable components, ensuring the system grows organically alongside the rapid pace of LLM advancement.

A.4.1. TASK TEMPLATE

The "Living" aspect of the ecosystem is powered by the `ExperimentBase` class, which serves as a universal blueprint for community contribution. By abstracting the experiment lifecycle into five distinct, enforceable stages: `prepare_data`, `data_iterator`, `run_agent`, `parse_output`, and `calculate_metrics`, we lower the barrier to entry for introducing complex reasoning domains. A community contributor need only implement these specific methods to plug a new task using existing dataset or a new set of tasks from a new dataset into the ecosystem. This rigid yet flexible structure ensures that while the content of the benchmark diversifies through community effort, the scientific rigor of the evaluation remains consistent. The framework handles the orchestration, logging, and state management, allowing contributors to focus purely on defining the logic of the task itself.

```python
class ExperimentBase(ABC):
    """
    Abstract base class for orchestrating agent-based experiments.

    This class defines the standard lifecycle of an experiment:
    1. Data preparation (`prepare_data`)
    2. Input iteration (`data_iterator`)
    3. Agent execution (`run_agent`)
    4. Output parsing (`parse_output`)
    5. Metric calculation (`calculate_metrics`)

    Attributes:
        task (str): The name of the specific task being experimented on.
        num_test (int): The maximum number of test samples to run.
        logs_dir (Path): The directory where logs will be stored.
        agent (Optional[AgentBase]): The agent instance to be evaluated.
    """

    def __init__(
        self,
        task: str,
        num_test: int = 50,
        logs_dir: Optional[Path] = None,
        agent: Optional[AgentBase] = None,
    ):
        """
        Initialize the ExperimentBase.

        Args:
            task (str): The identifier for the experiment task
            num_test (int, optional): The number of samples to test. Defaults to 50.
            logs_dir (Optional[Path], optional): Custom path for logging. If None
            agent (Optional[AgentBase], optional): The agent instance to run the
                experiment against. Defaults to None.
        """
        self.num_test = num_test
        # Set default logs directory if not provided and task is specified
        self.logs_dir = logs_dir
        if self.logs_dir and not self.logs_dir.exists():
            self.logs_dir.mkdir(parents=True, exist_ok=True)
        self.task = task
        self.agent = agent

    @abstractmethod
    def prepare_data(self) -> None:
        """
        Prepare the dataset and environment before the experiment loop begins.

        This method should handle tasks such as:
        - Downloading datasets.
        - Loading raw files into memory.
        - Data preprocessing
        - Setting up necessary global state for the experiment.
        """
        pass

    @abstractmethod
    def data_iterator(self) -> Generator[Dict[str, Any], None, None]:
        """
        Yield individual data samples for the experiment loop.

        Returns:
            Generator[Dict[str, Any], None, None]: A generator yielding a dictionary
            representing a single test case (containing 'input_data', 'GT', 'metadata').
        """
        pass
```

```
@abstractmethod
async def run_agent(self, data: Dict[str, Any]) -> Dict[str, Any]:
    """
    Execute the agent logic for a single data sample.

    Args:
        data (Dict[str, Any]): The input data for the current step (yielded
            by `data_iterator`).

    Returns:
        Dict[str, Any]: The raw result from the agent execution, usually containing
        the agent's response, trace, or internal state.
    """
    pass

@abstractmethod
def parse_output(
    self, content: Optional[str] = None, query_id: Optional[str] = None
) -> Tuple[Dict[str, Any], Any]:
    """
    Parse the raw output from the agent into a structured format for evaluation.

    This method supports two modes:
    1. Parsing a direct string (`content`).
    2. Loading output from a file associated with `query_id` if the agent
       writes to disk.

    Args:
        content (Optional[str], optional): The raw string output from the agent.
        query_id (Optional[str], optional): The unique ID of the query, used to
            locate output files if `content` is not sufficient.

    Returns:
        Tuple[Dict[str, Any], Any]: A tuple containing:
            - A dictionary with the parsed solution/answer.
            - A reason for failure (if any), or None if successful.
    """
    pass

@abstractmethod
def calculate_metrics(self, result_list: List[dict]) -> ExperimentMetrics:
    """
    Compute aggregate performance metrics for the entire experiment.

    Args:
        result_list (List[dict]): A list of dictionaries, where each dictionary
            contains the parsed results and ground truth for a single test case.

    Returns:
        ExperimentMetrics: A structured object containing calculated metrics
        (e.g., Accuracy, IoU, SMAPE).
    """
    pass
```

### A.4.2. AGENT TEMPLATE

The robustness of this ecosystem is secured by the `AgentBase` class, which standardizes the interaction between the evaluation environment and the rapidly proliferating landscape of LLMs. By distilling complex model interactions into a single, unified asynchronous query method, the framework becomes model-agnostic. Whether evaluating a proprietary model like GPT-4 or a local, open-source agent, the system treats them as uniform entities adhering to the same protocol. This polymorphism ensures that our benchmark is not brittle: it is ready to assess the agents of tomorrow immediately. Consequently, the framework acts as a bridge, allowing any standardized agent to attempt any community-contributed task, effectively creating a many-to-many testing matrix that defines a truly thriving, collaborative ecosystem.

```
class AgentBase(ABC):
    """
    Abstract base class defining the interface for AI agents.

    This class serves as a blueprint for creating specific agent implementations
    that interact with various models (LLMs). It enforces a consistent structure
    for initialization and querying, ensuring that all subclasses provide the
    necessary logic to handle prompts, context data, logging, and response generation.

    Attributes:
        model_name (str): The identifier of the specific model being used (e.g., 'gpt-4', 'claude-3').
        name (str): The name of the agent instance, defaulting to the class name.
```

```python
        """

    def __init__(self, model_name: str, **kwargs):
        """
        Initialize the AgentBase instance.

        Args:
            model_name (str): The name or identifier of the underlying model to be used
                by the agent (e.g., "gpt-4-turbo").
            **kwargs: Additional keyword arguments for configuration or model-specific
                parameters. These can be captured and used by subclasses.
        """
        self.model_name = model_name
        self.name = self.__class__.__name__

    @abstractmethod
    async def query(
        self, prompt: str, data: Dict[str, Any], logs_dir: Path, query_id: str
    ) -> str:
        """
        Execute an asynchronous query against the agent.

        This abstract method must be implemented by subclasses to define how the
        agent processes a prompt and data. It is responsible for formatting the
        input, interacting with the model API, handling the conversation history,
        and persisting logs to the specified directory.

        Args:
            prompt (str): The main instructional text or query to send to the agent.
            data (Dict[str, Any]): Contextual data required for the query. This may include
                signals, channel information, user history, or other relevant metadata.
            logs_dir (Path): The filesystem path where logs, artifacts, and conversation
                traces should be saved for debugging and auditing.
            query_id (str): A unique identifier for the specific query execution, used
                for tracing and organizing log files.

        Returns:
            str: The final textual response content generated by the agent.

        Raises:
            NotImplementedError: If the subclass does not implement this method.
        """
        pass
```

## B. Experiment Setting Details

### B.1. Naive Baseline

To contextualize model performance across diverse metrics, we introduce a set of naive baselines tailored to each task type. The detailed calculation methods are: (1) For tasks evaluated via Accuracy, the baseline employs a uniform random guessing strategy (1/num_options). (2) For IoU-based localization tasks, we generate a random interval $[t, t+d]$, where the start time $t$ is uniformly sampled from the valid range and the duration $d$ equals the average event length in the test set. (3) For tasks evaluated by the sMAPE-score, the baseline strategy varies by category: we utilize the population mean for Cross-modal Translation tasks and the input signal's mean as a constant prediction for Future Forecasting and Signal Imputation tasks.

Notably, the baseline for perception tasks defaults to a perfect reference score of 1.0, reflecting the clear, deterministic analytical solutions inherent to such tasks. For instance, calculating sleep efficiency relies on a strict mathematical definition (total sleep time / time in bed), naturally yielding a perfect score when executed correctly.

### B.2. Model Intelligence Index

To quantify the general reasoning capabilities of the LLMs in our study, we adopt the Intelligence Index methodology proposed by Kim et al., 2025. This composite metric is designed to isolate intrinsic model performance across general-purpose reasoning, knowledge retrieval, mathematics, and coding, explicitly independent of agentic workflows or multi-agent collaboration structures. The index is constructed via an equal-weighted aggregation of eight standardized evaluation suites: MMLU-Pro (Wang et al., 2024b), GPQA Diamond (Rein et al., 2023), HLE (Phan et al., 2025), AIME 2025, SciCode (Tian et al., 2024), LiveCodeBench (Jain et al., 2024), IFBench (Pyatkin et al., 2025), and AA-LCR (Artificial Analysis, 2025). To ensure consistency and reliability, all models are evaluated under uniform zero-shot, instruction-prompted conditions using robust equality checks and pass@1 scoring. Table 4 presents the calculated Intelligence Index for all evaluated LLMs. The

intelligence indices for GLM 5 and Gemini 3.1 Pro are omitted, as GLM 5 lacks official AIME 2025 and LiveCodeBench scores, and Gemini 3.1 Pro lacks an official LiveCodeBench pass@1 scoring.

*Table 4.* **Intelligence Index of all LLMs evaluated on HEARTS.**

| Model | Index | AA-LCR | HLE | MMLU-Pro | GPQA Diamond | AIME 25 | LiveCode | SciCode | IFBench |
|---|---|---|---|---|---|---|---|---|---|
| ∞ Llama 4 Maverick | 42 | 46 | 5 | 81 | 67 | 19 | 40 | 33 | 43 |
| Qwen3 Coder 480B | 45 | 42 | 4 | 79 | 62 | 39 | 59 | 36 | 41 |
| GPT 4.1 Mini | 45 | 42 | 5 | 78 | 66 | 46 | 48 | 40 | 38 |
| Nemotron Nano 12B V2 | 48 | 40 | 5 | 76 | 57 | 75 | 69 | 26 | 32 |
| Claude 4.5 Haiku | 58 | 70 | 4 | 80 | 67 | 84 | 62 | 43 | 54 |
| Gemini 2.5 Flash | 58 | 57 | 13 | 84 | 79 | 78 | 63 | 41 | 52 |
| MiniMax M2 | 63 | 61 | 13 | 82 | 78 | 78 | 83 | 36 | 72 |
| Qwen3 235B | 63 | 67 | 15 | 83 | 79 | 91 | 79 | 42 | 51 |
| DeepSeek V3.1 | 64 | 65 | 15 | 83 | 79 | 90 | 80 | 41 | 57 |
| Gemini 2.5 Pro | 65 | 66 | 21 | 86 | 84 | 88 | 80 | 43 | 49 |
| Grok 4.1 Fast | 66 | 68 | 18 | 85 | 85 | 89 | 82 | 44 | 53 |
| GPT 5 Mini | 68 | 68 | 20 | 84 | 83 | 91 | 84 | 39 | 75 |
| Kimi K2 Thinking | 68 | 66 | 22 | 85 | 84 | 95 | 85 | 42 | 68 |
| GLM 4.7 | 70 | 64 | 25 | 86 | 86 | 95 | 89 | 45 | 68 |
| GLM 5 | - | 63 | 27 | 86 | 82 | - | - | 46 | 72 |
| Gemini 3.1 Pro | - | 73 | 45 | 91 | 94 | 95 | - | 59 | 77 |

## B.3. SOTA Machine Learning Baseline

To strictly quantify the performance disparity between generalist LLMs and specialized models, we curate a registry of state-of-the-Art machine learning baselines for a representative subset of 32 tasks within HEARTS. We acknowledge that these external baselines are drawn from diverse literature, datasets, and may operate under varying experimental protocols (e.g., specific data splits or preprocessing pipelines). Consequently, these comparisons should not be interpreted as a strictly controlled competitive evaluation, but rather as an indicative upper bound of domain-specific capability. Despite these methodological variations, we ensure semantic alignment in task definitions to provide valid insights into the limitations of current LLM reasoning. Table 5 details the specific model architectures, reported performance metrics, and original sources for these 32 reference tasks.

# C. More Results

## C.1. Performance Details

In this section we report the comparative performance of 16 LLMs on HEARTS. Fig. 11 presents leaderboard for HEARTS. All models are ranked by their overall scores.

Table 6 provides a granular performance breakdown for all 16 LLMs and naive baselines across the full suite of 110 tasks.

*Table 5.* **Benchmarking LLM Agents against SOTA ML Models across 32 tasks on HEARTS.** Scores represent the average and best performance achieved by the LLMs compared to SOTA baselines.

| Dataset | Task | Avg. LLMs | Best LLM | SOTA ML | Source |
|---|---|---|---|---|---|
| *Individual-level Analysis* | | | | | |
| Bridge2AI-voice | Voice Parkinson's Detection | 0.52 | 0.56 | 0.92 | Jeong et al., 2024 |
| CGMacros | CGM Diabetes Classification | 0.54 | 0.65 | 0.97 | Ayat et al., 2024 |
| CGMacros | Glycemic Response Comparison | 0.83 | 0.89 | 0.87 | Herrero et al., 2022 |
| CoughVID | Cough Event Detection | 0.54 | 0.60 | 0.88 | Orlandic et al., 2021 |
| PERG-IOBA | PERG Eye Disease Detection | 0.58 | 0.62 | 0.68 | Bowd et al., 2009 |
| PhyMER | Cross-session Emotion Match | 0.57 | 0.73 | 0.70 | Singh et al., 2025b |
| SHHS | ECG Atrial Fibrillation | 0.56 | 0.68 | 0.99 | Cai et al., 2020 |
| SHHS | Cardiovasc. Mortality Pred. | 0.49 | 0.54 | 0.88 | Thapa et al., 2026 |
| SHHS | PSG Smoker Classification | 0.52 | 0.56 | 0.94 | Çay et al., 2024 |
| SHHS | PSG Stroke Prediction | 0.51 | 0.55 | 0.82 | Thapa et al., 2026 |
| VitalDB | Perioperative Myocardial Injury | 0.52 | 0.60 | 0.79 | Oh et al., 2023a |
| *Physiology Classification* | | | | | |
| Capture24 | Accel. Activity Recognition | 0.12 | 0.17 | 0.90 | Burq & Sridhar, 2023 |
| Coswara | Cough COVID-19 Detection | 0.46 | 0.56 | 0.97 | Pahar et al., 2022 |
| Coswara | Cough COVID-19 (w/ Symptoms) | 0.71 | 0.76 | 0.97 | Pahar et al., 2022 |
| Coswara | Speech COVID-19 Detection | 0.47 | 0.58 | 0.87 | Pahar et al., 2022 |
| CoughVID | Audio COVID-19 Class. | 0.50 | 0.56 | 0.91 | Pahar et al., 2022 |
| CoughVID | Audio Health Status Class. | 0.50 | 0.57 | 0.91 | Kumar et al., 2025 |
| GazeBase | Eye-tracking Task Class. | 0.31 | 0.46 | 0.82 | Sadhu et al., 2025 |
| GLOBEM | Behavioral Depression Pred. | 0.53 | 0.59 | 0.70 | Bhattacharya et al., 2024 |
| GrabMyo | EMG Gesture Classification | 0.29 | 0.48 | 0.90 | Cansiz et al., 2025 |
| GrabMyo | EMG Subject Identification | 0.55 | 0.61 | 0.79 | Cansiz et al., 2025 |
| Harespod | Alt.-Respiration Ranking | 0.20 | 0.31 | 0.96 | Kim & Kim, 2025 |
| PhyMER | Multimodal Emotion Class. | 0.38 | 0.43 | 0.70 | Singh et al., 2025b |
| PhyMER | EDA Emotion Classification | 0.14 | 0.18 | 0.72 | Singh et al., 2025b |
| SHHS | EOG REM/NREM Class. | 0.49 | 0.58 | 0.73 | Yetton et al., 2016 |
| SHHS | EEG Sleep Stage Class. | 0.41 | 0.54 | 0.81 | Gunnarsdottir et al., 2018 |
| VitalDB | ABP Hypotension Prediction | 0.72 | 0.77 | 0.88 | Shim et al., 2025 |
| VitalDB | PPG Hypotension Prediction | 0.50 | 0.55 | 0.88 | Shim et al., 2025 |
| *Localization* | | | | | |
| GazeBase | Gaze Fixation Localization | 0.36 | 0.40 | 0.87 | Svaricek et al., 2025 |
| GLOBEM | Peak Stress Week Ident. | 0.31 | 0.40 | 0.48 | Booth et al., 2022 |
| PAMAP2 | Activity Segment Localization | 0.17 | 0.25 | 0.99 | Bollampally et al., 2024 |
| *Cross-visit Comparison* | | | | | |
| SHHS | Longitudinal BMI Prediction | 0.49 | 0.55 | 0.74 | Li et al., 2021 |

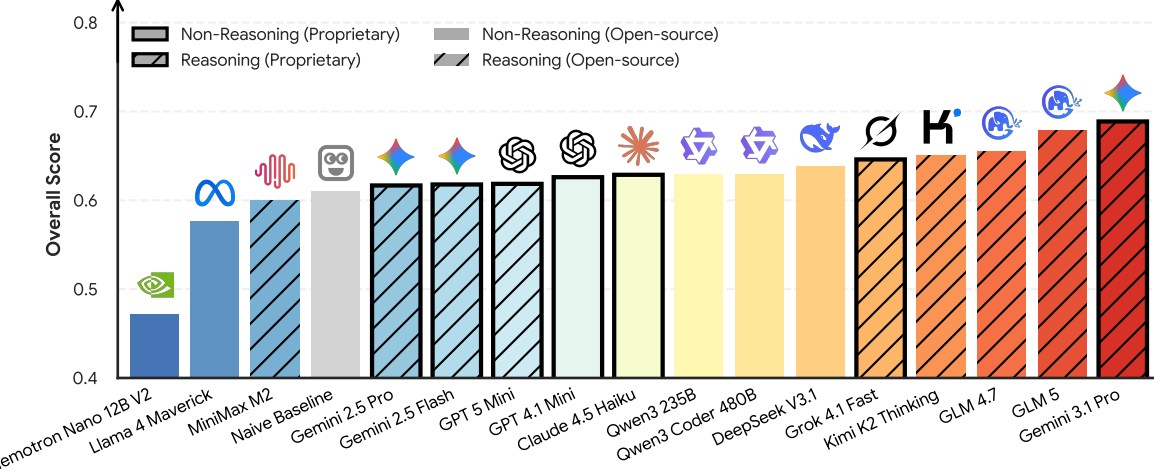

*Figure 11.* **HEARTS Leaderboard.** We report results for 16 state-of-the-art LLMs.

*Table 6.* **Performance across datasets and tasks of all LLMs**

| Dataset | Task | Naive Baseline | Claude 4.5 Haiku | DeepSeek V3.1 | Gemini 2.5 Flash | Gemini 2.5 Pro | Gemini 3.1 Pro | GLM 4.7 | GLM 5 | GPT 4.1 Mini | GPT 5 Mini | Grok 4.1 Fast | Kimi K2 Thinking | Llama 4 Maver- ick | MiniMax M2 | Nemotron Nano 12B V2 | Qwen3 235B | Qwen3 Coder 480B |
|---|---|---|---|---|---|---|---|---|---|---|---|---|---|---|---|---|---|---|
| Bridge2AI-voice | Articulation Rate Calculation | 1.00 | 0.79 | 0.75 | 0.69 | 0.83 | 0.81 | 0.80 | 0.86 | 0.67 | 0.73 | 0.84 | 0.80 | 0.47 | 0.78 | 0.55 | 0.79 | 0.81 |
| Bridge2AI-voice | Cross-task Voice Comparison | 0.25 | 0.32 | 0.36 | 0.27 | 0.26 | 0.43 | 0.30 | 0.33 | 0.33 | 0.39 | 0.29 | 0.35 | 0.38 | 0.32 | 0.28 | 0.28 | 0.32 |
| Bridge2AI-voice | F0 Range Extraction | 1.00 | 0.75 | 0.80 | 0.82 | 0.82 | 0.80 | 0.83 | 0.78 | 0.66 | 0.67 | 0.82 | 0.78 | 0.65 | 0.83 | 0.80 | 0.79 | 0.82 |
| Bridge2AI-voice | Harmonics to Noise Ratio (HNR) Extraction | 1.00 | 0.52 | 0.48 | 0.57 | 0.53 | 0.69 | 0.58 | 0.72 | 0.35 | 0.34 | 0.50 | 0.61 | 0.30 | 0.60 | 0.59 | 0.47 | 0.66 |
| Bridge2AI-voice | Jitter Extraction | 1.00 | 0.17 | 0.27 | 0.19 | 0.11 | 0.06 | 0.27 | 0.26 | 0.16 | 0.06 | 0.21 | 0.39 | 0.10 | 0.08 | 0.19 | 0.32 | 0.27 |
| Bridge2AI-voice | Parkinsons Diagnosis | 0.50 | 0.52 | 0.56 | 0.53 | 0.49 | 0.51 | 0.51 | 0.49 | 0.56 | 0.48 | 0.51 | 0.50 | 0.52 | 0.51 | 0.54 | 0.51 | 0.48 |
| Bridge2AI-voice | Shimmer Extraction | 1.00 | 0.58 | 0.62 | 0.55 | 0.65 | 0.75 | 0.63 | 0.70 | 0.63 | 0.45 | 0.73 | 0.61 | 0.54 | 0.58 | 0.38 | 0.49 | 0.62 |
| Bridge2AI-voice | Reversed Signal Detection | 0.50 | 0.57 | 0.83 | 0.60 | 0.75 | 0.63 | 0.64 | 0.77 | 0.54 | 0.57 | 0.64 | 0.66 | 0.48 | 0.48 | 0.49 | 0.67 | 0.54 |
| Capture24 | Activity Classification | 0.07 | 0.10 | 0.10 | 0.08 | 0.13 | 0.13 | 0.14 | 0.14 | 0.14 | 0.10 | 0.14 | 0.17 | 0.11 | 0.13 | 0.05 | 0.14 | 0.12 |
| Capture24 | Activity Transition Recognition | 0.02 | 0.00 | 0.00 | 0.00 | 0.00 | 0.00 | 0.00 | 0.00 | 0.00 | 0.00 | 0.00 | 0.00 | 0.00 | 0.00 | 0.00 | 0.00 | 0.00 |
| Capture24 | 1-axis Signal Imputation | 0.73 | 0.78 | 0.83 | 0.78 | 0.85 | 0.84 | 0.81 | 0.83 | 0.83 | 0.81 | 0.84 | 0.84 | 0.80 | 0.79 | 0.59 | 0.80 | 0.83 |
| Capture24 | Step Count Calculation | 1.00 | 0.93 | 0.89 | 0.73 | 0.85 | 0.95 | 0.93 | 0.94 | 0.92 | 0.96 | 0.92 | 0.92 | 0.79 | 0.86 | 0.56 | 0.82 | 0.91 |
| Capture24 | 3-axis Signal Forecasting | 0.72 | 0.82 | 0.78 | 0.83 | 0.80 | 0.81 | 0.81 | 0.81 | 0.76 | 0.79 | 0.81 | 0.81 | 0.82 | 0.76 | 0.73 | 0.80 | 0.83 |
| Capture24 | 3-axis Signal Imputation | 0.73 | 0.70 | 0.63 | 0.82 | 0.67 | 0.80 | 0.79 | 0.83 | 0.76 | 0.71 | 0.77 | 0.83 | 0.76 | 0.76 | 0.58 | 0.68 | 0.75 |
| Capture24 | Day and Night Signal Ordering | 0.50 | 0.73 | 0.89 | 0.79 | 0.83 | 0.94 | 0.88 | 0.95 | 0.82 | 0.98 | 0.91 | 0.83 | 0.60 | 0.68 | 0.46 | 0.91 | 0.74 |
| CGMacros | A1c Prediction | 0.33 | 0.64 | 0.62 | 0.63 | 0.57 | 0.67 | 0.61 | 0.58 | 0.57 | 0.53 | 0.65 | 0.54 | 0.53 | 0.44 | 0.29 | 0.49 | 0.47 |
| CGMacros | CGM Time in Range Calculation | 1.00 | 1.00 | 1.00 | 0.95 | 0.99 | 1.00 | 1.00 | 1.00 | 0.97 | 0.91 | 0.98 | 1.00 | 0.99 | 0.87 | 0.54 | 0.88 | 0.99 |
| CGMacros | Fasting GLU Prediction | 0.90 | 0.78 | 0.74 | 0.32 | 0.50 | 0.75 | 0.76 | 0.77 | 0.78 | 0.78 | 0.78 | 0.72 | 0.65 | 0.72 | 0.69 | 0.78 | 0.76 |
| CGMacros | Postprandial CGM iAUC Calculation | 1.00 | 0.93 | 0.89 | 0.94 | 0.87 | 0.99 | 0.95 | 0.92 | 0.99 | 0.99 | 0.96 | 0.95 | 0.79 | 0.88 | 0.33 | 0.93 | 0.83 |
| CGMacros | CGM Forecasting with Histroy CGM and Meal Info | 0.67 | 0.55 | 0.54 | 0.54 | 0.55 | 0.58 | 0.57 | 0.54 | 0.49 | 0.52 | 0.58 | 0.59 | 0.41 | 0.55 | 0.42 | 0.54 | 0.53 |
| CGMacros | CGM Forecasting with Current Meal Information and Histroy CGM and Meal Information | 0.66 | 0.52 | 0.48 | 0.39 | 0.57 | 0.61 | 0.58 | 0.54 | 0.52 | 0.53 | 0.51 | 0.54 | 0.33 | 0.51 | 0.45 | 0.55 | 0.54 |
| CGMacros | CGM Forecasting with Meal Information | 0.66 | 0.52 | 0.52 | 0.40 | 0.54 | 0.61 | 0.61 | 0.55 | 0.54 | 0.46 | 0.54 | 0.54 | 0.45 | 0.51 | 0.30 | 0.47 | 0.49 |
| CGMacros | CGM Forecasting | 0.67 | 0.47 | 0.50 | 0.54 | 0.49 | 0.66 | 0.56 | 0.53 | 0.48 | 0.49 | 0.49 | 0.54 | 0.41 | 0.51 | 0.51 | 0.53 | 0.52 |
| CGMacros | Meal Image Classification from CGM | 0.25 | 0.29 | - | 0.10 | 0.25 | 0.13 | - | 0.18 | 0.26 | 0.27 | 0.26 | - | 0.29 | - | 0.23 | - | - |
| CGMacros | Postprandial CGM Response Comparison | 0.50 | 0.82 | 0.84 | 0.89 | 0.81 | 0.89 | 0.88 | 0.88 | 0.89 | 0.84 | 0.87 | 0.87 | 0.85 | 0.78 | 0.54 | 0.83 | 0.85 |
| CGMacros | Meal Time Localization | 0.71 | 0.63 | 0.64 | 0.60 | 0.61 | 0.68 | 0.83 | 0.59 | 0.68 | 0.63 | 0.67 | 0.69 | 0.62 | 0.64 | 0.62 | 0.63 | 0.67 |
| CGMacros | CGM imputation with Activity Calories Trajectory | 0.72 | 0.72 | 0.55 | 0.53 | 0.45 | 0.74 | 0.68 | 0.75 | 0.63 | 0.55 | 0.67 | 0.71 | 0.42 | 0.69 | 0.40 | 0.58 | 0.71 |
| CGMacros | CGM imputation | 0.71 | 0.98 | 0.98 | 0.99 | 0.99 | 0.99 | 0.99 | 0.98 | 0.98 | 0.98 | 0.98 | 0.98 | 0.93 | 0.98 | 0.59 | 0.98 | 0.98 |
| CGMacros | CGM Imputation with HR Trajectory | 0.70 | 0.98 | 0.95 | 0.94 | 0.94 | 0.97 | 0.97 | 0.98 | 0.95 | 0.97 | 0.97 | 0.96 | 0.96 | 0.96 | 0.71 | 0.95 | 0.98 |
| Coswara | Audio Type Classification | 0.33 | 0.32 | 0.29 | 0.36 | 0.00 | 0.79 | 0.45 | 0.39 | 0.36 | 0.33 | 0.41 | 0.35 | 0.28 | 0.39 | 0.33 | 0.33 | 0.39 |
| Coswara | Diagnosis Classification with Cough Audio | 0.50 | 0.47 | 0.53 | 0.53 | 0.00 | 0.33 | 0.56 | 0.49 | 0.47 | 0.47 | 0.48 | 0.47 | 0.52 | 0.51 | 0.46 | 0.51 | 0.52 |

*Continued on next page*

| Dataset | Task | Naive Baseline | Claude 4.5 Haiku | DeepSeek V3.1 | Gemini 2.5 Flash | Gemini 2.5 Pro | Gemini 3.1 Pro | GLM 4.7 | GLM 5 | GPT 4.1 Mini | GPT 5 Mini | Grok 4.1 Fast | Kimi K2 Thinking | Llama 4 Maverick | MiniMax M2 | Nemotron Nano 12B V2 | Qwen3 235B | Qwen3 Coder 480B |
|---|---|---|---|---|---|---|---|---|---|---|---|---|---|---|---|---|---|---|
| Coswara | Diagnosis Classification with Symptom Information Only | 0.50 | 0.76 | 0.78 | 0.76 | 0.81 | 0.78 | 0.79 | 0.74 | 0.76 | 0.78 | 0.79 | 0.81 | 0.78 | 0.70 | 0.80 | 0.74 | 0.80 |
| Coswara | Diagnosis Classification with Cough Audio and Symptom Information | 0.50 | 0.66 | 0.74 | 0.72 | 0.70 | 0.83 | 0.73 | 0.72 | 0.74 | 0.70 | 0.74 | 0.69 | 0.76 | 0.68 | 0.74 | 0.70 | 0.71 |
| Coswara | Diagnosis Classification with Speech Audio | 0.50 | 0.58 | 0.47 | 0.49 | 0.00 | 0.51 | 0.49 | 0.50 | 0.52 | 0.48 | 0.54 | 0.51 | 0.48 | 0.52 | 0.54 | 0.47 | 0.53 |
| Cough VID | Cough Detection wtih Good Quality Samples | 0.50 | 0.50 | 0.54 | 0.57 | 0.55 | 0.59 | 0.59 | 0.61 | 0.54 | 0.54 | 0.58 | 0.46 | 0.60 | 0.56 | 0.46 | 0.49 | 0.55 |
| Cough VID | Cough Detection with Poor Quality Samples | 0.50 | 0.55 | 0.65 | 0.62 | 0.00 | 0.60 | 0.57 | 0.59 | 0.50 | 0.55 | 0.41 | 0.73 | 0.73 | 0.50 | 0.57 | 0.42 | 0.38 |
| Cough VID | COVID Status Classification | 0.50 | 0.50 | 0.45 | 0.54 | 0.44 | 0.51 | 0.54 | 0.51 | 0.52 | 0.48 | 0.49 | 0.54 | 0.51 | 0.46 | 0.47 | 0.53 | 0.56 |
| Cough VID | Diagnosis Classification | 0.20 | 0.20 | 0.18 | 0.17 | 0.10 | 0.07 | 0.20 | 0.18 | 0.23 | 0.11 | 0.20 | 0.19 | 0.22 | 0.19 | 0.13 | 0.23 | 0.17 |
| Cough VID | Health Status Classification | 0.50 | 0.48 | 0.55 | 0.50 | 0.49 | 0.57 | 0.47 | 0.45 | 0.57 | 0.51 | 0.51 | 0.48 | 0.57 | 0.50 | 0.46 | 0.46 | 0.49 |
| Cough VID | MFCC Mean & STD Calculation | 1.00 | 0.99 | 0.99 | 1.00 | 0.79 | 1.00 | 0.97 | 0.99 | 0.99 | 1.00 | 0.73 | 0.88 | 0.98 | 0.94 | 0.42 | 0.98 | 1.00 |
| GazeBase | Fixation Point Localization | 0.35 | 0.36 | 0.37 | 0.38 | 0.39 | 0.37 | 0.36 | 0.38 | 0.37 | 0.37 | 0.20 | 0.37 | 0.38 | 0.38 | 0.40 | 0.37 | 0.38 |
| GazeBase | Horizontal Saccade Track Forecasting | 0.79 | 0.78 | 0.79 | 0.80 | 0.83 | 0.90 | 0.82 | 0.84 | 0.81 | 0.84 | 0.84 | 0.81 | 0.76 | 0.80 | 0.50 | 0.80 | 0.80 |
| GazeBase | Text Reading Imputation | 0.86 | 0.83 | 0.84 | 0.85 | 0.78 | 0.83 | 0.84 | 0.84 | 0.82 | 0.84 | 0.85 | 0.83 | 0.85 | 0.84 | 0.91 | 0.84 | 0.84 |
| GazeBase | Reading Sequence Recognition | 0.17 | 0.29 | 0.21 | 0.10 | 0.03 | 0.19 | 0.40 | 0.31 | 0.34 | 0.09 | 0.23 | 0.38 | 0.23 | 0.20 | 0.17 | 0.17 | 0.29 |
| GazeBase | Eye-tracking Task Classification | 0.17 | 0.24 | 0.24 | 0.24 | 0.46 | 0.33 | 0.40 | 0.38 | 0.42 | 0.29 | 0.45 | 0.35 | 0.22 | 0.28 | 0.18 | 0.30 | 0.30 |
| GLOBEM | Circadian Routine Comparison | 0.50 | 0.52 | 0.57 | 0.61 | 0.60 | 0.48 | 0.55 | 0.53 | 0.63 | 0.53 | 0.57 | 0.55 | 0.65 | 0.54 | 0.59 | 0.55 | 0.64 |
| GLOBEM | COVID Year Recognition | 0.50 | 0.96 | 0.94 | 0.93 | 0.90 | 0.60 | 0.92 | 0.92 | 0.94 | 0.90 | 0.93 | 0.93 | 0.95 | 0.88 | 0.49 | 0.92 | 0.92 |
| GLOBEM | Depression Trajectory Classification | 0.50 | 0.54 | 0.48 | 0.51 | 0.54 | 0.53 | 0.59 | 0.55 | 0.56 | 0.55 | 0.49 | 0.55 | 0.50 | 0.51 | 0.52 | 0.53 | 0.51 |
| GLOBEM | Location Entropy Extraction | 1.00 | 0.80 | 0.72 | 0.62 | 0.68 | 0.74 | 0.74 | 0.78 | 0.80 | 0.76 | 0.83 | 0.72 | 0.78 | 0.79 | 0.53 | 0.69 | 0.77 |
| GLOBEM | Peak Stress Week Localization | 0.10 | 0.28 | 0.29 | 0.26 | 0.31 | 0.33 | 0.30 | 0.32 | 0.30 | 0.31 | 0.30 | 0.30 | 0.30 | 0.38 | 0.40 | 0.29 | 0.31 |
| GLOBEM | Step Count Forecasting | 0.68 | 0.78 | 0.77 | 0.76 | 0.76 | 0.78 | 0.78 | 0.79 | 0.76 | 0.74 | 0.76 | 0.75 | 0.78 | 0.78 | 0.65 | 0.77 | 0.78 |
| GrabMyo | Gesture Classification with Reference | 0.06 | 0.46 | 0.34 | 0.38 | 0.42 | 0.56 | 0.48 | 0.52 | 0.11 | 0.24 | 0.32 | 0.39 | 0.14 | 0.35 | 0.06 | 0.10 | 0.32 |
| GrabMyo | Subject Identification | 0.50 | 0.55 | 0.58 | 0.60 | 0.61 | 0.60 | 0.56 | 0.62 | 0.52 | 0.48 | 0.60 | 0.55 | 0.47 | 0.55 | 0.57 | 0.56 | 0.56 |
| Harespod | Altitude Ranking with Respiration | 0.17 | 0.20 | 0.28 | 0.15 | 0.22 | 0.13 | 0.23 | 0.20 | 0.31 | 0.14 | 0.14 | 0.21 | 0.23 | 0.25 | 0.00 | 0.18 | 0.22 |
| Harespod | Altitude Ranking with SpO2 | 0.17 | 0.32 | 0.32 | 0.33 | 0.37 | 0.39 | 0.33 | 0.37 | 0.33 | 0.31 | 0.32 | 0.33 | 0.27 | 0.37 | 0.00 | 0.26 | 0.38 |
| Harespod | Respiration and HR Pairing | 0.50 | 0.60 | 0.53 | 0.46 | 0.58 | 0.57 | 0.56 | 0.67 | 0.69 | 0.61 | 0.65 | 0.61 | 0.56 | 0.69 | 0.50 | 0.55 | 0.60 |
| Harespod | Respiration and SpO2 Pairing | 0.50 | 0.72 | 0.50 | 0.49 | 0.48 | 0.61 | 0.60 | 0.75 | 0.41 | 0.74 | 0.66 | 0.56 | 0.52 | 0.68 | 0.51 | 0.46 | 0.59 |
| PAMAP2 | Activity Localization | 0.15 | 0.23 | 0.22 | 0.18 | 0.19 | 0.79 | 0.25 | 0.27 | 0.15 | 0.08 | 0.21 | 0.20 | 0.11 | 0.18 | 0.03 | 0.17 | 0.21 |
| PERG-IOBA | Eye Disease Type Classification with Patient's Meta Information | 0.25 | 0.28 | 0.32 | 0.44 | 0.27 | 0.50 | 0.34 | 0.36 | 0.40 | 0.35 | 0.31 | 0.39 | 0.38 | 0.30 | 0.35 | 0.42 | 0.34 |
| PERG-IOBA | Eye Disease Type Classification | 0.25 | 0.24 | 0.26 | 0.35 | 0.29 | 0.34 | 0.37 | 0.39 | 0.40 | 0.34 | 0.30 | 0.40 | 0.31 | 0.37 | 0.30 | 0.46 | 0.37 |
| PERG-IOBA | Disease Differentiation: between Macular Disease & Optic Nerve Disease & Normal | 0.50 | 0.38 | 0.54 | 0.55 | 0.52 | 0.61 | 0.53 | 0.54 | 0.58 | 0.56 | 0.53 | 0.52 | 0.56 | 0.56 | 0.57 | 0.56 | 0.49 |
| PERG-IOBA | Eye Health Status Classification | 0.50 | 0.54 | 0.55 | 0.62 | 0.60 | 0.57 | 0.57 | 0.54 | 0.56 | 0.61 | 0.61 | 0.54 | 0.57 | 0.62 | 0.57 | 0.59 | 0.58 |
| PERG-IOBA | N35, P50, N95 Feature Extraction | 1.00 | 1.00 | 1.00 | 0.88 | 0.85 | 0.99 | 1.00 | 1.00 | 1.00 | 0.98 | 0.92 | 1.00 | 1.00 | 0.96 | 0.54 | 1.00 | 1.00 |
| PhyMER | Cross Subject Arousal Ranking | 0.50 | 0.50 | 0.48 | 0.51 | 0.52 | 0.53 | 0.54 | 0.53 | 0.51 | 0.56 | 0.51 | 0.58 | 0.50 | 0.44 | 0.60 | 0.51 | 0.54 |

*Continued on next page*

| Dataset | Task | Naive Baseline | Claude 4.5 Haiku | DeepSeek V3.1 | Gemini 2.5 Flash | Gemini 2.5 Pro | Gemini 3.1 Pro | GLM 4.7 | GLM 5 | GPT 4.1 Mini | GPT 5 Mini | Grok 4.1 Fast | Kimi K2 Thinking | Llama 4 Maverick | MiniMax M2 | Nemotron Nano 12B V2 | Qwen3 235B | Qwen3 Coder 480B |
|---|---|---|---|---|---|---|---|---|---|---|---|---|---|---|---|---|---|---|
| PhyMER | Conditional Forecasting | 0.21 | 0.54 | 0.39 | 0.38 | 0.50 | 0.57 | 0.58 | 0.55 | 0.40 | 0.40 | 0.47 | 0.50 | 0.42 | 0.43 | 0.07 | 0.45 | 0.49 |
| PhyMER | Conditional Imputation | 0.23 | 0.62 | 0.54 | 0.60 | 0.62 | 0.69 | 0.68 | 0.66 | 0.56 | 0.57 | 0.63 | 0.58 | 0.33 | 0.56 | 0.26 | 0.52 | 0.59 |
| PhyMER | Cross-channel Translation | 0.43 | 0.72 | 0.70 | 0.69 | 0.78 | 0.77 | 0.80 | 0.80 | 0.68 | 0.79 | 0.79 | 0.62 | 0.65 | 0.67 | 0.35 | 0.68 | 0.77 |
| PhyMER | Emotion Type Classification | 0.14 | 0.38 | 0.39 | 0.35 | 0.34 | 0.38 | 0.41 | 0.35 | 0.42 | 0.43 | 0.41 | 0.37 | 0.30 | 0.35 | 0.29 | 0.43 | 0.41 |
| PhyMER | Emotion Type Classification with Only EDA Input | 0.14 | 0.10 | 0.16 | 0.11 | 0.12 | 0.19 | 0.17 | 0.14 | 0.18 | 0.14 | 0.15 | 0.13 | 0.17 | 0.13 | 0.13 | 0.14 | 0.16 |
| PhyMER | Inter-subject Emotion Recognition | 0.50 | 0.52 | 0.54 | 0.55 | 0.55 | 0.54 | 0.52 | 0.60 | 0.61 | 0.63 | 0.55 | 0.63 | 0.55 | 0.46 | 0.73 | 0.60 | 0.56 |
| PhyMER | Personality Analysis | 0.25 | 0.27 | 0.30 | 0.20 | 0.19 | 0.23 | 0.28 | 0.24 | 0.24 | 0.24 | 0.21 | 0.19 | 0.23 | 0.16 | 0.21 | 0.31 | 0.27 |
| PhyMER | Single-channel Forecasting | 0.21 | 0.56 | 0.56 | 0.34 | 0.56 | 0.58 | 0.59 | 0.59 | 0.50 | 0.54 | 0.56 | 0.62 | 0.43 | 0.53 | 0.35 | 0.50 | 0.60 |
| PhyMER | Single-channel Imputation | 0.23 | 0.69 | 0.73 | 0.72 | 0.72 | 0.76 | 0.73 | 0.71 | 0.72 | 0.72 | 0.73 | 0.73 | 0.70 | 0.74 | 0.40 | 0.73 | 0.72 |
| Shanghai Diabetes | Diabetes Type Classification | 0.50 | 0.60 | 0.61 | 0.54 | 0.78 | 0.60 | 0.52 | 0.72 | 0.69 | 0.62 | 0.62 | 0.73 | 0.75 | 0.78 | 0.54 | 0.59 | 0.61 |
| Shanghai Diabetes | Cross-Subject Diabetes Type Comparison | 0.50 | 0.76 | 0.87 | 0.87 | 0.88 | 0.89 | 0.82 | 0.85 | 0.86 | 0.81 | 0.82 | 0.80 | 0.81 | 0.76 | 0.48 | 0.85 | 0.81 |
| SHHS | Atrial Fibrillation (AF) Classification | 0.50 | 0.54 | 0.55 | 0.57 | 0.62 | 0.60 | 0.62 | 0.55 | 0.48 | 0.57 | 0.68 | 0.55 | 0.53 | 0.53 | 0.53 | 0.53 | 0.55 |
| SHHS | Sleep AHI Calculation | 1.00 | 0.92 | 0.87 | 0.85 | 0.37 | 0.96 | 0.91 | 0.93 | 0.89 | 0.83 | 0.97 | 0.89 | 0.38 | 0.88 | 0.31 | 0.84 | 0.89 |
| SHHS | Arousal Detection (EEG) | 0.00 | 0.26 | 0.19 | 0.09 | 0.15 | 0.39 | 0.27 | 0.34 | 0.16 | 0.11 | 0.26 | 0.30 | 0.10 | 0.19 | 0.10 | 0.14 | 0.20 |
| SHHS | Arousal Detection (EOG) | 0.00 | 0.20 | 0.12 | 0.10 | 0.09 | 0.29 | 0.18 | 0.27 | 0.15 | 0.11 | 0.20 | 0.21 | 0.10 | 0.20 | 0.09 | 0.12 | 0.16 |
| SHHS | Bandpower Calculation | 1.00 | 1.00 | 1.00 | 0.92 | 0.99 | 1.00 | 0.99 | 1.00 | 1.00 | 0.98 | 1.00 | 0.98 | 0.99 | 0.97 | 0.43 | 0.97 | 1.00 |
| SHHS | BMI Comparison Between Visit | 0.50 | 0.50 | 0.53 | 0.46 | 0.45 | 0.39 | 0.47 | 0.52 | 0.55 | 0.48 | 0.45 | 0.43 | 0.49 | 0.55 | 0.54 | 0.48 | 0.47 |
| SHHS | Conditional Forecasting | 0.94 | 0.91 | 0.91 | 0.92 | 0.93 | 0.92 | 0.91 | 0.91 | 0.93 | 0.90 | 0.91 | 0.91 | 0.92 | 0.91 | 0.91 | 0.92 | 0.92 |
| SHHS | Conditional Imputation | 0.93 | 0.88 | 0.90 | 0.77 | 0.92 | 0.90 | 0.89 | 0.91 | 0.90 | 0.90 | 0.90 | 0.89 | 0.87 | 0.90 | 0.76 | 0.91 | 0.89 |
| SHHS | Cross-channel Translation (ECG to HR) | 0.38 | 0.53 | 0.58 | 0.61 | 0.71 | 0.74 | 0.56 | 0.56 | 0.37 | 0.58 | 0.69 | 0.66 | 0.54 | 0.59 | 0.22 | 0.59 | 0.66 |
| SHHS | Cross-channel Translation (EEG-C3A2 to EEG-C4A1) | 0.92 | 0.90 | 0.90 | 0.92 | 0.92 | 0.92 | 0.91 | 0.90 | 0.91 | 0.91 | 0.91 | 0.91 | 0.90 | 0.91 | 0.90 | 0.90 | 0.90 |
| SHHS | Cardiovascular Disease (CVD) Death Prediction | 0.50 | 0.45 | 0.44 | 0.52 | 0.54 | 0.53 | 0.46 | 0.49 | 0.53 | 0.51 | 0.47 | 0.46 | 0.52 | 0.40 | 0.51 | 0.50 | 0.52 |
| SHHS | Hypopnea Detection | 0.02 | 0.31 | 0.23 | 0.23 | 0.12 | 0.30 | 0.31 | 0.36 | 0.14 | 0.09 | 0.22 | 0.30 | 0.06 | 0.28 | 0.11 | 0.12 | 0.21 |
| SHHS | REM Latency Calculation | 1.00 | 0.92 | 0.92 | 0.90 | 0.91 | 0.93 | 0.91 | 0.91 | 0.92 | 0.92 | 0.92 | 0.88 | 0.93 | 0.89 | 0.59 | 0.92 | 0.92 |
| SHHS | REM/NREM Classification | 0.50 | 0.51 | 0.52 | 0.40 | 0.56 | 0.30 | 0.53 | 0.47 | 0.58 | 0.54 | 0.52 | 0.51 | 0.49 | 0.41 | 0.35 | 0.52 | 0.48 |
| SHHS | Single-channel Forecasting | 0.93 | 0.90 | 0.91 | 0.91 | 0.91 | 0.91 | 0.91 | 0.91 | 0.92 | 0.89 | 0.91 | 0.90 | 0.91 | 0.91 | 0.86 | 0.91 | 0.91 |
| SHHS | Single-channel Imputation | 0.93 | 0.86 | 0.70 | 0.74 | 0.92 | 0.91 | 0.87 | 0.91 | 0.73 | 0.89 | 0.90 | 0.86 | 0.87 | 0.88 | 0.92 | 0.86 | 0.87 |
| SHHS | Sleep Efficiency Calculation | 1.00 | 1.00 | 0.99 | 0.95 | 0.99 | 1.00 | 0.94 | 0.88 | 1.00 | 1.00 | 0.99 | 0.97 | 0.87 | 0.02 | 0.67 | 0.94 | 0.96 |
| SHHS | Smoker Classification | 0.50 | 0.53 | 0.48 | 0.53 | 0.50 | 0.53 | 0.52 | 0.50 | 0.52 | 0.56 | 0.56 | 0.51 | 0.47 | 0.55 | 0.48 | 0.49 | 0.54 |
| SHHS | Sleep Stage Classification | 0.25 | 0.45 | 0.45 | 0.45 | 0.42 | 0.17 | 0.42 | 0.24 | 0.43 | 0.41 | 0.45 | 0.41 | 0.28 | 0.54 | 0.29 | 0.39 | 0.42 |
| SHHS | Sleep Stage Transition Recognition | 0.08 | 0.11 | 0.23 | 0.43 | 0.84 | 0.39 | 0.31 | 0.22 | 0.09 | 0.10 | 0.20 | 0.28 | 0.08 | 0.16 | 0.16 | 0.13 | 0.09 |
| SHHS | Stroke Prediction | 0.50 | 0.47 | 0.55 | 0.47 | 0.47 | 0.62 | 0.52 | 0.51 | 0.49 | 0.48 | 0.52 | 0.53 | 0.53 | 0.49 | 0.53 | 0.51 | 0.54 |
| SHHS | Episode Level Ordering | 0.50 | 0.52 | 0.55 | 0.53 | 0.52 | 0.94 | 0.54 | 0.51 | 0.50 | 0.48 | 0.55 | 0.48 | 0.53 | 0.46 | 0.51 | 0.54 | 0.47 |
| SHHS | Half Night Level Ordering | 0.50 | 0.70 | 0.76 | 0.83 | 0.85 | 0.83 | 0.67 | 0.74 | 0.73 | 0.74 | 0.76 | 0.75 | 0.60 | 0.67 | 0.52 | 0.81 | 0.57 |
| SHHS | Visit Level Ordering | 0.50 | 0.53 | 0.56 | 0.68 | 0.69 | 0.67 | 0.56 | 0.59 | 0.52 | 0.53 | 0.55 | 0.55 | 0.56 | 0.54 | 0.48 | 0.59 | 0.56 |
| VCTK | Reversed Signal Detection | 0.50 | 0.49 | 0.50 | 0.67 | 0.47 | 0.67 | 0.48 | 0.89 | 0.47 | 0.59 | 0.55 | 0.58 | 0.49 | 0.50 | 0.51 | 0.58 | 0.43 |
| VitalDB | Anesthesia Range | 0.33 | 0.30 | 0.23 | 0.33 | 0.70 | 0.86 | 0.56 | 0.68 | 0.36 | 0.16 | 0.66 | 0.54 | 0.75 | 0.21 | 0.04 | 0.44 | 0.40 |
| VitalDB | MINS (myocardial injury after non-cardiac surgery) Prediction with MBP | 0.50 | 0.60 | 0.47 | 0.55 | 0.53 | 0.50 | 0.52 | 0.53 | 0.51 | 0.51 | 0.54 | 0.54 | 0.52 | 0.54 | 0.49 | 0.49 | 0.55 |
| VitalDB | Drug Infusion Series to Depth of Anesthesia (BIS) Translation | 0.72 | 0.41 | 0.46 | 0.48 | 0.40 | 0.61 | 0.54 | 0.52 | 0.52 | 0.49 | 0.53 | 0.58 | 0.16 | 0.58 | 0.36 | 0.36 | 0.55 |
| VitalDB | PPG and ECG to Blood Pressure Translation | 0.40 | 0.54 | 0.48 | 0.50 | 0.51 | 0.50 | 0.51 | 0.50 | 0.52 | 0.51 | 0.51 | 0.53 | 0.52 | 0.52 | 0.34 | 0.49 | 0.52 |

*Continued on next page*

| Dataset | Task | Naive Baseline | Claude 4.5 Haiku | DeepSeek V3.1 | Gemini 2.5 Flash | Gemini 2.5 Pro | Gemini 3.1 Pro | GLM 4.7 | GLM 5 | GPT 4.1 Mini | GPT 5 Mini | Grok 4.1 Fast | Kimi K2 Thinking | Llama 4 Maverick | MiniMax M2 | Nemotron Nano 12B V2 | Qwen3 235B | Qwen3 Coder 480B |
|---|---|---|---|---|---|---|---|---|---|---|---|---|---|---|---|---|---|---|
| VitalDB | EEG to Bispectral Index (BIS) Translation | 0.70 | 0.50 | 0.48 | 0.46 | 0.59 | 0.60 | 0.53 | 0.61 | 0.44 | 0.55 | 0.35 | 0.48 | 0.47 | 0.58 | 0.46 | 0.52 | 0.58 |
| VitalDB | MBP Forecasting | 0.29 | 0.58 | 0.54 | 0.53 | 0.52 | 0.61 | 0.54 | 0.58 | 0.57 | 0.55 | 0.58 | 0.58 | 0.54 | 0.55 | 0.33 | 0.58 | 0.57 |
| VitalDB | Hypotension Event Classification with MBP | 0.50 | 0.73 | 0.72 | 0.73 | 0.68 | 0.80 | 0.75 | 0.74 | 0.77 | 0.72 | 0.74 | 0.75 | 0.70 | 0.77 | 0.55 | 0.74 | 0.75 |
| VitalDB | MBP Forecasting with Infusion Info | 0.27 | 0.55 | 0.54 | 0.46 | 0.52 | 0.60 | 0.55 | 0.56 | 0.54 | 0.53 | 0.58 | 0.57 | 0.55 | 0.54 | 0.33 | 0.54 | 0.57 |
| VitalDB | Hypotension Event Classification with PPG | 0.50 | 0.48 | 0.47 | 0.49 | 0.51 | 0.50 | 0.55 | 0.52 | 0.47 | 0.49 | 0.55 | 0.48 | 0.52 | 0.50 | 0.51 | 0.48 | 0.48 |
| VitalDB | Mean arterial pressure (MBP) Time in Range 70-100 | 1.00 | 1.00 | 1.00 | 1.00 | 1.00 | 1.00 | 1.00 | 1.00 | 1.00 | 1.00 | 0.93 | 1.00 | 1.00 | 0.99 | 0.91 | 0.99 | 1.00 |

## C.2. Supplement Analysis Across Task Categories

To contextualize our analysis in Sec. 4.2, we provide supplement analysis in this section.

### C.2.1. PERCEPTION TASK

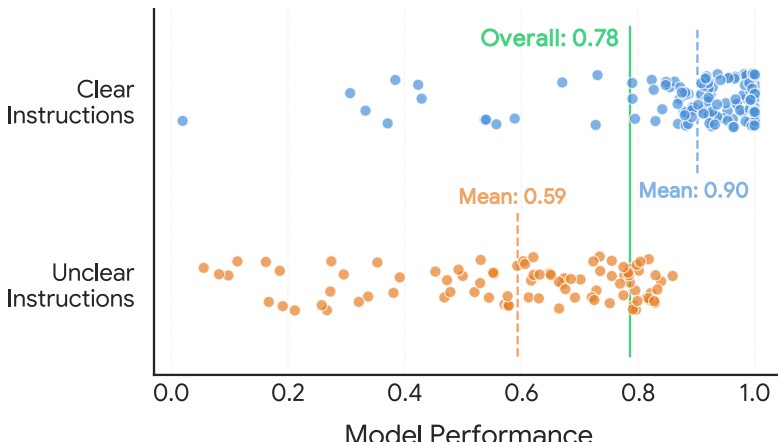

*Figure 12.* **Impact of instruction clarity in Perception category.** Each dot represents the performance of a single model on a specific perception task.

The data visualized in Fig. 12 illustrates a critical insight: while LLMs possess a strong baseline level for perception tasks in general, their reliability is heavily dependent on how the problem is framed. It reveals a stark divergence between the two conditions. If LLMs were provided with step-by-step guidance, performance is propelled to a mean of 0.90, effectively unlocking the model's full potential. Conversely, the scattered orange distribution below shows that leaving instructions result-oriented creates significant volatility, dragging the mean score down to 0.59. A fundamental limitation is revealed: current LLMs still lack robust autonomous reasoning capabilities. They rely heavily on detailed instructions to perform effectively, struggling to bridge the gap from a simple target to a complex solution on their own.

### C.2.2. GENERATION TASK

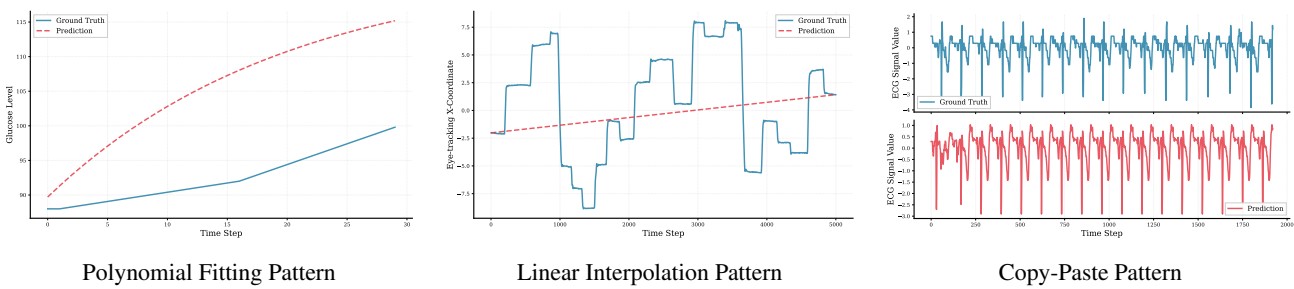

| Polynomial Fitting Pattern | Linear Interpolation Pattern | Copy-Paste Pattern |

*Figure 13.* **Common Generation Patterns exhibited by LLMs for sequence imputation and forecasting.** The three most prevalent strategies are: polynomial curve fitting, linear interpolation, and copy-pasting observed segments.

Fig. 13 shows 3 most common patterns of how LLMs approach time-series generation tasks. The visualization reveals a critical tendency: rather than modeling the intricate stochastic dynamics or causal dependencies of the data, LLMs frequently collapse predictions into elementary mathematical functions. As observed in the polynomial fitting pattern, the model smooths distinct glucose level fluctuations into a generic curve, drifting significantly from the actual trend. Similarly, the linear interpolation ignores the structured and quasi-periodic behavior of eye-tracking coordinates, cutting through the data with a naive straight line. The copy-paste behavior further highlights this lack of genuine reasoning capability, where the LLM simply replicates previous segments of an ECG signal. These failures suggest that instead of solving problems from deep analysis of time series itself, generation tasks are governed by low-complexity heuristics.

## C.3. Heatmap on Input Modality

We provide a granular visualization of model performance across distinct input modalities, supplementing the domain-level analysis in Sec. 4.3. As illustrated in Fig. 14, the heatmap reveals a striking stratification of difficulty inherent to specific signal types, mirroring the inter-model consistency observed in our domain analysis. This indicates that performance is primarily governed by the intrinsic semantic accessibility of the signal representation rather than model-specific architectural biases. Specifically, we observe a universal competence in processing high-level, structured signals (e.g., HR, Aggregated Data, and BP), which yield consistently positive Kappa scores across the model cohort. Conversely, waveform-dense modalities such as ECG and CGM impose a uniform performance bottleneck, resulting in negative scores even for advanced models. This suggests that current LLM tokenization strategies may be fundamentally misaligned with the continuous, high-frequency nature of these specific biosignals, creating a 'representation gap' that scaling alone cannot easily bridge.

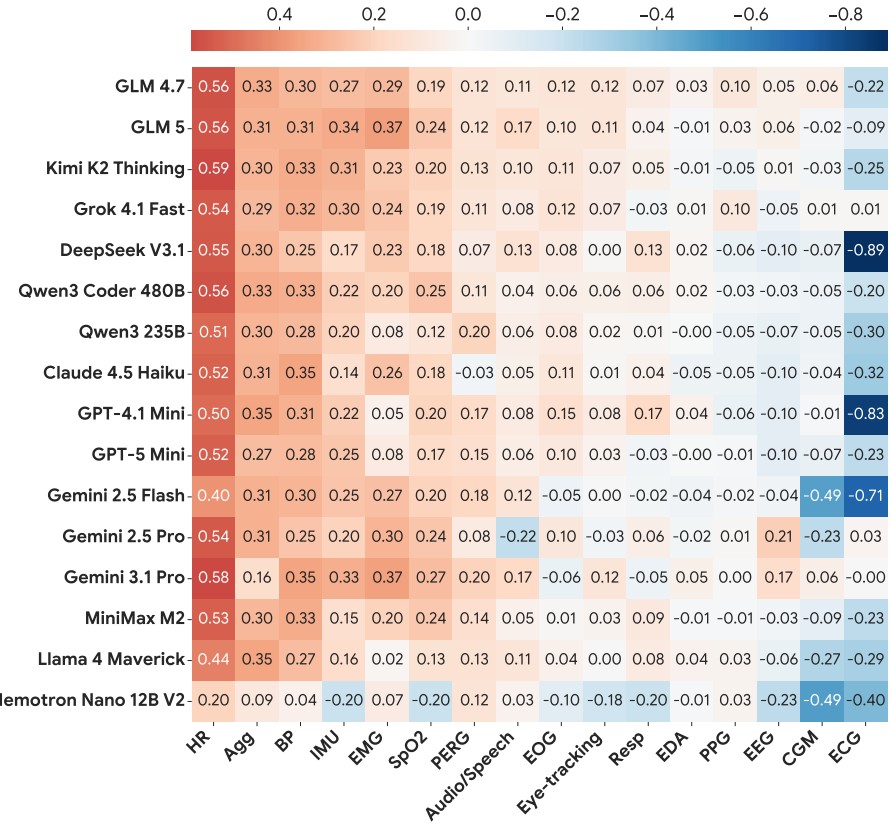

*Figure 14.* **Model performance heatmap across input modality.**

## C.4. Input Format Experiments

To support the input format evaluation discussed in Sec. 4.5, we specify 10 representative tasks and their corresponding input time series, whose text-based representations fit within the LLM context window. Detailed performance results for each task are reported in Table 7.

## C.5. Supplementary Material for Additional Studies

### C.5.1. INFLUENCE OF INFORMATION QUALITY

The experiments in this section investigated a critical question: can Large Language Models (LLMs) effectively synthesize auxiliary information to achieve their goals? This serves as a vital probe for their reasoning capabilities. As shown in Fig. 15, the results reveal that adding data does not automatically improve performance. In the Coswara dataset, where the additional input consisted of clear symptom descriptions, the model could directly apply this context to aid its diagnosis. However, in the other three datasets, the auxiliary inputs were raw signals, abstract data formats that the model could

*Table 7.* **Results breakdown for input format experiments.**

| Dataset | Task | ⌀ Grok 4.1 Fast | | | ∞ Llama 4 Maverick | | |
|---|---|---|---|---|---|---|---|
| | | **File** | **Text** | **Image** | **File** | **Text** | **Image** |
| CGMacros | CGM Forecasting | 0.49 | 0.55 | 0.47 | 0.41 | 0.54 | 0.44 |
| | CGM imputation | 0.98 | 0.98 | 0.73 | 0.93 | 0.99 | 0.96 |
| GLOBEM | Circadian Routine Comparison | 0.57 | 0.57 | 0.49 | 0.65 | 0.53 | 0.57 |
| | Depression Trajectory Classification | 0.50 | 0.50 | 0.50 | 0.50 | 0.54 | 0.53 |
| | Location Entropy Extraction | 0.83 | 0.83 | 0.52 | 0.78 | 0.82 | 0.69 |
| | Peak Stress Week Localization | 0.30 | 0.30 | 0.36 | 0.30 | 0.30 | 0.29 |
| | Step Count Forecasting | 0.76 | 0.78 | 0.73 | 0.78 | 0.78 | 0.77 |
| PERG_IOBA | Eye Health Status Classification | 0.62 | 0.56 | 0.60 | 0.58 | 0.52 | 0.60 |
| VitalDb | MBP Forecasting | 0.58 | 0.60 | 0.33 | 0.54 | 0.55 | 0.51 |
| | Hypotension Event Classification with PPG | 0.74 | 0.72 | 0.68 | 0.71 | 0.70 | 0.71 |

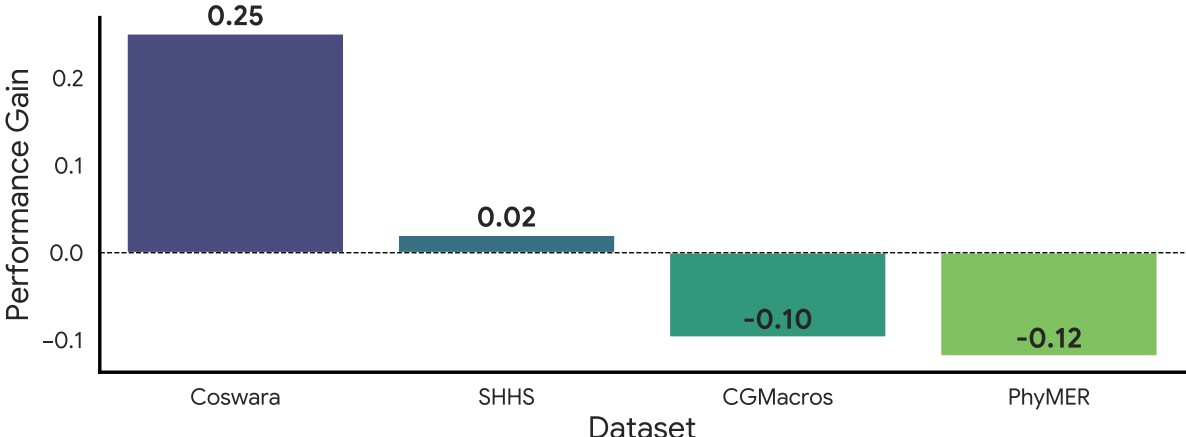

*Figure 15.* **Performance Gain with Auxiliary Information**

not intuitively interpret. Consequently, performance gains in these cases were negligible or even detrimental, exposing a significant limitation in the models' ability to synthesize complex, indirect data. This evidence consolidates our conclusion: more input information does not inherently result in better outcomes.

### C.5.2. INFLUENCE OF DOMAIN-SPECIFIC AGENT

As demonstrated in Sec. 4.6, a domain-specialist agent does not inherently outperform generalist models. Comparing the trajectories from the CodeAct and Biomni frameworks below, it is evident that the behavior of two agents are similar, and the specialized agent (Biomni) failed to provide additional utility in resolving the task.

**CodeAct reasoning trajectory**

---

**User**

You are provided with 3-axis accelerometer data from a wearable device. Your task is to classify the physical activity being performed during the recorded segment. The 14 possible activities are: sleeping, manual-work, sitting, heavy loads, leisure, sports/gym, standing, transportation, occupation, activity, vehicle, mixed-activity, walking, and household-chores. The 3-axis data is provided as three separate arrays corresponding to the x, y, and z axes of the accelerometer in 'input/x.npy', 'input/y.npy', and 'input/z.npy'. The signals are sampled at 100 Hz. Please analyze the signal and determine which activity it represents in json format. Your answer must be one of the 14 possible activities. Please output your final result in a json object without any other text: { "activity": "activity_name" }

---

## Agent

### Thought

I need to analyze 3-axis accelerometer data to classify physical activity. Let me first load the data from the three files and explore the signal characteristics to determine what activity is being performed.

The 14 possible activities are: - sleeping - manual-work - sitting - heavy loads - leisure - sports/gym - standing - transportation - occupation - activity - vehicle - mixed-activity - walking - household-chores

Let me start by loading the data and examining its properties.

```python
import numpy as np
import json

# Load the accelerometer data
x = np.load('input/x.npy')
y = np.load('input/y.npy')
z = np.load('input/z.npy')

print(f"Shape of x: {x.shape}")
print(f"Shape of y: {y.shape}")
print(f"Shape of z: {z.shape}")
```

### Thought

Observation: The file '/mnt/data/input/x.npy' doesn't exist.</think> The file path seems incorrect. Let me try the correct path 'input/x.npy' as specified in the task.

```python
import numpy as np
import json

# Load the accelerometer data
x = np.load('input/x.npy')
y = np.load('input/y.npy')
z = np.load('input/z.npy')

print(f"Shape of x: {x.shape}")
print(f"Shape of y: {y.shape}")
print(f"Shape of z: {z.shape}")
print(f"Number of samples: {len(x)}")
print(f"Duration in seconds: {len(x)/100:.2f}")
```

### Thought

Observation: Shape of x: (6000,) Shape of y: (6000,) Shape of y: (6000,) Shape of z: (6000,) Shape of z: (6000,) Shape of z: (6000,) Shape of z: (6000,) Shape of z: (6000,) Shape of z: (6000,) Shape of z: (6000,) Shape of z: (6000,) Shape of z: (6000,) activity classification. I'll calculate the signal magnitude and analyze its statistical features.</think> I have loaded the accelerometer data. There are 6000 samples at 100 Hz, so the recording duration is 60 seconds. Let me now analyze the signal to extract features that can help classify the activity. I'll calculate the signal magnitude and analyze its statistical characteristics.

```python
# Calculate signal magnitude (acceleration magnitude)
magnitude = np.sqrt(x**2 + y**2 + z**2)

# Calculate basic statistics
print(f"Mean magnitude: {np.mean(magnitude):.4f}")
print(f"Std magnitude: {np.std(magnitude):.4f}")
print(f"Min magnitude: {np.min(magnitude):.4f}")
print(f"Max magnitude: {np.max(magnitude):.4f}")
print(f"Mean x: {np.mean(x):.4f}, Std x: {np.std(x):.4f}")
print(f"Mean y: {np.mean(y):.4f}, Std y: {np.std(y):.4f}")
print(f"Mean z: {np.mean(z):.4f}, Std z: {np.std(z):.4f}")
```

## Thought

Observation: The user wants me to classify the accelerometer data into one of 14 activity categories. I've already loaded the data and calculated basic statistics. Let me analyze this more carefully.

The data has: - 6000 samples at 100 Hz = 60 seconds of recording - Mean magnitude around 1.0 (suggesting gravity is present) - Std of magnitude around 0.02 - x has mean close to 0, std around 0.014 - y has mean close to 0, std around 0.014 - z has mean around 1.0, std around 0.007

The z-axis having a mean around 1.0 suggests the device is mostly stationary with gravity aligned along the z-axis (horizontal orientation). The low standard deviations across all axes indicate very little movement, which strongly suggests a stationary activity like sleeping, sitting, or standing.

Let me analyze the signal more carefully by looking at: 1. The variance of the signal magnitude 2. Zero-crossing rate 3. Signal variability patterns

With such low std values, this is very likely a low-movement activity like sleeping, sitting, or standing.

I should provide my final classification based on this analysis.</think> I've loaded the accelerometer data which has 6000 samples at 100 Hz (60 seconds duration). The statistics show: - Mean magnitude ~1.0 (gravity is present) - Very low standard deviations across all axes (0.007-0.014) - z-axis mean is ~1.0, suggesting the device is lying flat with gravity aligned - x and y means close to 0

These characteristics strongly indicate a stationary, low-movement activity. The very low variability suggests this is likely "sleeping" since: 1. Sleeping typically has the lowest movement among all activities 2. The device would be relatively stationary 3. The gravitational alignment suggests lying down

Let me confirm by looking at more detailed signal characteristics.

```python
# Calculate additional features
# Signal magnitude variance
mag_var = np.var(magnitude)
# Zero-crossing rate
zero_crossings = np.sum(np.diff(np.sign(magnitude - np.mean(magnitude))) != 0)
# Energy of the signal
energy = np.sum(magnitude**2)

print(f"Magnitude variance: {mag_var:.8f}")
print(f"Zero-crossing rate: {zero_crossings}")
print(f"Signal energy: {energy:.4f}")
print(f"Signal range (max-min): {np.max(magnitude) - np.min(magnitude):.6f}")

# Check for periodic patterns that might suggest breathing during sleep
# Look at frequency domain characteristics
import scipy.signal as signal
freqs, psd = signal.welch(magnitude, fs=100)
dominant_freq = freqs[np.argmax(psd)]
print(f"Dominant frequency: {dominant_freq:.4f} Hz")
```

## Execution Output

Shape of x: (14800,) Shape of y: (14800,) Shape of z: (14800,) Shape of x: (14800,) Shape of y: (14800,) Shape of z: (14800,) Number of samples: 14800 Duration in seconds: 148.00 Mean magnitude: 0.9934 Std magnitude: 0.2337 Min magnitude: 0.0514 Max magnitude: 5.1575 Mean x: -0.2258, Std x: 0.3409 Mean y: -0.7474, Std y: 0.2365 Mean z: -0.4546, Std z: 0.2301 Magnitude variance: 0.05462577 Zero-crossing rate: 1290 Signal energy: 15412.8530 Signal range (max-min): 5.106137 Dominant frequency: 1.9531 Hz

## Agent

### Thought

Let me analyze these signal characteristics:

1. **Duration**: 148 seconds at 100 Hz 2. **Mean magnitude**: 0.9934 g (confirms gravity is present) 3. **Std magnitude**: 0.2337 g (moderate movement variation) 4. **Max magnitude**: 5.1575 g (high peaks indicating vigorous movement) 5. **Signal range**: 5.106 g (wide range - significant intensity) 6. **Zero-crossing rate**: 1290 (frequent direction changes) 7. **Dominant frequency**: 1.9531 Hz (~117 cycles/minute)

The dominant frequency of ~2 Hz (117 per minute) is very characteristic of: - Walking cadence (typically 100-120 steps per minute) - Aerobic exercise rhythms

The high magnitude peaks (up to 5.16g) combined with: - Rhythmic pattern at 2 Hz - Moderate overall variance - Sustained energy over 148 seconds

This pattern suggests rhythmic, repetitive aerobic movement. The gravitational vector (negative in all axes) suggests a tilted orientation possibly during exercise.

Given: - Rhythmic ~2 Hz pattern (consistent with walking or exercise cadence) - Sustained activity over 2.5 minutes (consistent with cardio workout) - High but controlled acceleration (not chaotic, not stationary)

This most likely represents **walking** or an aerobic exercise. Given the repetitive nature and cadence, let me check the signal patterns more closely. However, with 117 cycles per minute and moderate variance, this aligns well with walking pace.

The rhythmic, sustained movement at walking cadence frequency strongly suggests "walking".

```python
# Let me examine the signal segments to confirm the pattern
import matplotlib
matplotlib.use('Agg')
import matplotlib.pyplot as plt

# Sample a few segments to check patterns
fig, axes = plt.subplots(3, 1, figsize=(15, 10))
time = np.arange(0, min(len(x), 1000)) / 100.0  # First 10 seconds
```

```
axes[0].plot(time, x[:1000], label='X axis')
axes[0].set_ylabel('X Acceleration (g)')
axes[0].set_title('First 10 seconds of accelerometer data')
axes[0].grid(True)

axes[1].plot(time, y[:1000], label='Y axis')
axes[1].set_ylabel('Y Acceleration (g)')
axes[1].grid(True)

axes[2].plot(time, z[:1000], label='Z axis')
axes[2].set_ylabel('Z Acceleration (g)')
axes[2].set_xlabel('Time (s)')
axes[2].grid(True)

plt.tight_layout()
print("Signal visualization generated")
```

## Execution Output

Signal visualization generated

## Agent

### Thought

Based on my analysis of the accelerometer data:
Key findings: 1. **Duration**: 148 seconds of continuous activity 2. **Dominant frequency**: 1.95 Hz (~117 cycles/minute) - this is classic walking cadence range (100-120 steps/minute) 3. **Rhythmic pattern**: The zero-crossing rate and frequency analysis indicate repetitive, periodic movement 4. **Moderate variance**: 0.2337 g std magnitude suggests sustained, continuous movement 5. **Signal characteristics**: High peaks (up to 5.16g) consistent with foot impacts during locomotion
The ~2 Hz rhythmic pattern at 117 cycles/minute is the strongest indicator - this matches human walking cadence perfectly. Other activities would have different signatures: - Running: ~2.7-3 Hz (160-180 steps/min) - Sedentary (sitting/standing): much lower variance and frequency - Sports/gym: variable patterns not this consistent
The sustained, rhythmic nature combined with the characteristic walking cadence strongly identifies this as walking activity.

### Solution

{ "activity": "walking" }

## Biomni reasoning trajectory

## User

You are provided with 3-axis accelerometer data from a wearable device. Your task is to classify the physical activity being performed during the recorded segment. The 14 possible activities are: sleeping, manual-work, sitting, heavy loads, leisure, sports/gym, standing, transportation, occupation, activity, vehicle, mixed-activity, walking, and household-chores. The 3-axis data is provided as three separate arrays corresponding to the x, y, and z axes of the accelerometer in 'input/x.npy', 'input/y.npy', and 'input/z.npy'. The signals are sampled at 100 Hz. Please analyze the signal and determine which activity it represents in json format. Your answer must be one of the 14 possible activities. Please output your final result in a json object without any other text: { "activity": "activity_name" }

## Agent

### Thought

I'll analyze the 3-axis accelerometer data to classify the physical activity. Let me create a plan and start by loading and exploring the data.
Plan: 1. [ ] Load the accelerometer data from x.npy, y.npy, and z.npy files 2. [ ] Explore the data (shape, duration, basic statistics) 3. [ ] Calculate signal features (magnitude, variance, frequency characteristics) 4. [ ] Analyze activity patterns (movement intensity, periodicity) 5. [ ] Compare with typical patterns of each activity 6. [ ] Classify the activity and output in JSON format
Let me start:

```
import numpy as np
import json

# Load the accelerometer data
```

```
x = np.load('input/x.npy')
y = np.load('input/y.npy')
z = np.load('input/z.npy')

# Explore the data
print("Data shapes:")
print(f"X axis: {x.shape}")
print(f"Y axis: {y.shape}")
print(f"Z axis: {z.shape}")

print("\nData duration:")
 sampling_rate = 100  # Hz
duration = len(x) / sampling_rate
print(f"Duration: {duration:.2f} seconds")
print(f"Number of samples: {len(x)}")

print("\nBasic statistics:")
print(f"X - Mean: {np.mean(x):.4f}, Std: {np.std(x):.4f}, Min: {np.min(x):.4f}, Max: {np.max(x):.4f}")
print(f"Y - Mean: {np.mean(y):.4f}, Std: {np.std(y):.4f}, Min: {np.min(y):.4f}, Max: {np.max(y):.4f}")
print(f"Z - Mean: {np.mean(z):.4f}, Std: {np.std(z):.4f}, Min: {np.min(z):.4f}, Max: {np.max(z):.4f}")
```

## Agent

### Execution Output

Error: unexpected indent (<string>, line 16)

## Agent

```
import numpy as np
import json

# Load the accelerometer data
x = np.load('input/x.npy')
y = np.load('input/y.npy')
z = np.load('input/z.npy')

# Explore the data
print("Data shapes:")
print(f"X axis: {x.shape}")
print(f"Y axis: {y.shape}")
print(f"Z axis: {z.shape}")

print("\nData duration:")
sampling_rate = 100  # Hz
duration = len(x) / sampling_rate
print(f"Duration: {duration:.2f} seconds")
print(f"Number of samples: {len(x)}")

print("\nBasic statistics:")
print(f"X - Mean: {np.mean(x):.4f}, Std: {np.std(x):.4f}, Min: {np.min(x):.4f}, Max: {np.max(x):.4f}")
print(f"Y - Mean: {np.mean(y):.4f}, Std: {np.std(y):.4f}, Min: {np.min(y):.4f}, Max: {np.max(y):.4f}")
print(f"Z - Mean: {np.mean(z):.4f}, Std: {np.std(z):.4f}, Min: {np.min(z):.4f}, Max: {np.max(z):.4f}")
```

## Agent

### Execution Output

Data shapes: X axis: (14800,) Y axis: (14800,) Z axis: (14800,)
Data duration: Duration: 148.00 seconds Number of samples: 14800
Basic statistics: X - Mean: -0.2258, Std: 0.3409, Min: -5.1045, Max: 1.1073 Y - Mean: -0.7474, Std: 0.2365, Min: -2.3022, Max: 0.9180 Z - Mean: -0.4546, Std: 0.2301, Min: -1.6770, Max: 0.9072

## Agent

```python
# Calculate signal magnitude (vector magnitude)
magnitude = np.sqrt(x**2 + y**2 + z**2)

print("Magnitude statistics:")
print(f"Mean: {np.mean(magnitude):.4f}, Std: {np.std(magnitude):.4f}, Min: {np.min(magnitude):.4f}, Max: {
    np.max(magnitude):.4f}")

# Calculate variance of magnitude
var_magnitude = np.var(magnitude)
print(f"Variance of magnitude: {var_magnitude:.4f}")

# Calculate signal amplitude range
range_x = np.max(x) - np.min(x)
range_y = np.max(y) - np.min(y)
range_z = np.max(z) - np.min(z)
print(f"\nAmplitude ranges:")
print(f"X range: {range_x:.4f}")
print(f"Y range: {range_y:.4f}")
print(f"Z range: {range_z:.4f}")

# Calculate average acceleration magnitude
avg_mag = np.mean(magnitude)
print(f"\nAverage magnitude: {avg_mag:.4f}")

# Calculate energy of the signal (sum of squares)
energy = np.sum(x**2 + y**2 + z**2)
print(f"Signal energy: {energy:.4f}")

# Calculate zero crossing rate (indicator of frequency content)
def zero_crossing_rate(signal):
    return np.sum(np.diff(np.sign(signal)) != 0) / len(signal)

zcr_x = zero_crossing_rate(x)
zcr_y = zero_crossing_rate(y)
zcr_z = zero_crossing_rate(z)
print(f"\nZero crossing rates:")
print(f"X: {zcr_x:.4f}")
print(f"Y: {zcr_y:.4f}")
print(f"Z: {zcr_z:.4f}")
```

## Agent

### Execution Output

Magnitude statistics: Mean: 0.9934, Std: 0.2337, Min: 0.0514, Max: 5.1575 Variance of magnitude: 0.0546
Amplitude ranges: X range: 6.2119 Y range: 3.2202 Z range: 2.5841
Average magnitude: 0.9934 Signal energy: 15412.8530
Zero crossing rates: X: 0.0664 Y: 0.0016 Z: 0.0124

## Agent

```python
# Analyze frequency content using FFT
from scipy.fft import fft, fftfreq

def analyze_frequency(signal, sampling_rate):
    n = len(signal)
    yf = fft(signal)
    xf = fftfreq(n, 1/sampling_rate)
    # Get positive frequencies only
    positive_freq_idx = xf > 0
    xf = xf[positive_freq_idx]
    yf = np.abs(yf[positive_freq_idx])

    # Find dominant frequency
    dominant_freq_idx = np.argmax(yf)
    dominant_freq = xf[dominant_freq_idx]
    dominant_power = yf[dominant_freq_idx]
```

```
    # Calculate power in different frequency bands
    power_0_1 = np.mean(yf[(xf >= 0) & (xf <= 1)])
    power_1_3 = np.mean(yf[(xf > 1) & (xf <= 3)])
    power_3_10 = np.mean(yf[(xf > 3) & (xf <= 10)])
    power_10_plus = np.mean(yf[xf > 10])

    return {
        'dominant_freq': dominant_freq,
        'dominant_power': dominant_power,
        'power_0_1Hz': power_0_1,
        'power_1_3Hz': power_1_3,
        'power_3_10Hz': power_3_10,
        'power_10_plusHz': power_10_plus
    }

print("Frequency analysis:")
freq_x = analyze_frequency(x, sampling_rate)
freq_y = analyze_frequency(y, sampling_rate)
freq_z = analyze_frequency(z, sampling_rate)
freq_mag = analyze_frequency(magnitude, sampling_rate)

print(f"\nX axis - Dominant freq: {freq_x['dominant_freq']:.2f} Hz, Power in bands (0-1Hz): {freq_x['
    power_0_1Hz']:.2f}, (1-3Hz): {freq_x['power_1_3Hz']:.2f}, (3-10Hz): {freq_x['power_3_10Hz']:.2f}")
print(f"Y axis - Dominant freq: {freq_y['dominant_freq']:.2f} Hz, Power in bands (0-1Hz): {freq_y['
    power_0_1Hz']:.2f}, (1-3Hz): {freq_y['power_1_3Hz']:.2f}, (3-10Hz): {freq_y['power_3_10Hz']:.2f}")
print(f"Z axis - Dominant freq: {freq_z['dominant_freq']:.2f} Hz, Power in bands (0-1Hz): {freq_z['
    power_0_1Hz']:.2f}, (1-3Hz): {freq_z['power_1_3Hz']:.2f}, (3-10Hz): {freq_z['power_3_10Hz']:.2f}")
print(f"Magnitude - Dominant freq: {freq_mag['dominant_freq']:.2f} Hz, Power in bands (0-1Hz): {freq_mag['
    power_0_1Hz']:.2f}, (1-3Hz): {freq_mag['power_1_3Hz']:.2f}, (3-10Hz): {freq_mag['power_3_10Hz']:.2f}"
    )

# Analyze signal segments to check for consistency
print("\n\nSegment analysis (dividing into 5 segments):")
n_segments = 5
segment_length = len(x) // n_segments

for i in range(n_segments):
    start = i * segment_length
    end = (i + 1) * segment_length
    seg_mag = magnitude[start:end]
    seg_var = np.var(seg_mag)
    seg_mean = np.mean(seg_mag)
    print(f"Segment {i+1}: Mean mag: {seg_mean:.4f}, Var: {seg_var:.4f}")
```

## Agent

### Execution Output

Frequency analysis:
X axis - Dominant freq: 0.01 Hz, Power in bands (0-1Hz): 137.95, (1-3Hz): 67.98, (3-10Hz): 35.69 Y axis - Dominant freq: 1.86 Hz, Power in bands (0-1Hz): 83.19, (1-3Hz): 72.20, (3-10Hz): 26.14 Z axis - Dominant freq: 0.01 Hz, Power in bands (0-1Hz): 89.53, (1-3Hz): 57.88, (3-10Hz): 28.11 Magnitude - Dominant freq: 1.86 Hz, Power in bands (0-1Hz): 42.20, (1-3Hz): 81.41, (3-10Hz): 27.72
Segment analysis (dividing into 5 segments): Segment 1: Mean mag: 0.9945, Var: 0.0294 Segment 2: Mean mag: 1.0044, Var: 0.0768 Segment 3: Mean mag: 0.9997, Var: 0.0758 Segment 4: Mean mag: 0.9930, Var: 0.0772 Segment 5: Mean mag: 0.9752, Var: 0.0133

## Agent

```
# Additional analysis to confirm walking pattern
# Calculate autocorrelation to detect periodicity

def autocorrelation(signal, max_lag=200):
    """Calculate autocorrelation of signal"""
    n = len(signal)
    mean = np.mean(signal)
    var = np.var(signal)
    if var == 0:
        return np.zeros(max_lag)

    # Normalize signal
    signal_normalized = (signal - mean) / np.sqrt(var)
```

```python
    # Calculate autocorrelation
    autocorr = np.zeros(max_lag)
    for lag in range(max_lag):
        if lag < n:
            autocorr[lag] = np.corrcoef(signal_normalized[:-lag] if lag > 0 else signal_normalized,
                                        signal_normalized[lag:])[0, 1]

    return autocorr

# Calculate autocorrelation of magnitude
autocorr_mag = autocorrelation(magnitude, max_lag=200)

# Find peaks in autocorrelation (indicates periodicity)
from scipy.signal import find_peaks
peaks, _ = find_peaks(autocorr_mag, height=0.1, distance=20)

print("Autocorrelation analysis:")
print(f"Number of peaks found: {len(peaks)}")
if len(peaks) > 0:
    peak_lags = peaks
    peak_periods = peak_lags / sampling_rate  # Convert to seconds
    peak_freqs = 1 / peak_periods  # Convert to Hz
    print(f"Peak lags (samples): {peak_lags[:5]}")  # Show first 5
    print(f"Peak periods (seconds): {peak_periods[:5]}")
    print(f"Peak frequencies (Hz): {peak_freqs[:5]}")
    print(f"Primary frequency: {peak_freqs[0]:.2f} Hz")

# Calculate step count estimate based on zero crossings in magnitude
mag_detrended = magnitude - np.mean(magnitude)
zero_crossings = np.sum(np.diff(np.sign(mag_detrended)) != 0)
estimated_steps = zero_crossings // 2  # Each step typically causes 2 zero crossings
print(f"\nEstimated steps from zero crossings: {estimated_steps}")

# Walking pace estimation
walking_pace = estimated_steps / duration
print(f"Walking pace: {walking_pace:.2f} steps/second ({walking_pace * 60:.1f} steps/minute)")

# Signal-to-noise ratio (measure of rhythmicity)
signal_power = np.var(magnitude)
residual_noise = np.var(mag_detrended - np.mean(mag_detrended))
snr = 10 * np.log10(signal_power / residual_noise) if residual_noise > 0 else float('inf')
print(f"Signal-to-noise ratio: {snr:.2f} dB")

# Classification based on features
print("\n\nClassification Analysis:")
print("=" * 50)

# Key features for classification
print(f"Average magnitude: {avg_mag:.4f} (g)")
print(f"Magnitude variance: {var_magnitude:.4f}")
print(f"Dominant frequency: {freq_mag['dominant_freq']:.2f} Hz")
print(f"Power in walking band (1-3Hz): {freq_mag['power_1_3Hz']:.2f}")
print(f"Power ratio (1-3Hz / 0-1Hz): {freq_mag['power_1_3Hz'] / freq_mag['power_0_1Hz']:.2f}")
print(f"Estimated walking pace: {walking_pace * 60:.1f} steps/minute")

# Compare with expected values for different activities
print("\n\nComparison with activity characteristics:")
print("- Walking: Magnitude ~1g, dominant freq 1.5-2.5Hz, consistent rhythm")
print("- Running: Magnitude >1.5g, dominant freq 2.5-4Hz, higher energy")
print("- Sitting/Standing: Magnitude ~1g, very low variance, no rhythm")
print("- Sleeping: Very low magnitude, very low variance")
print("- Manual work: High variance, irregular patterns")
print("- Sports/Gym: High magnitude, high energy, variable patterns")

# Final classification decision
if (0.8 < avg_mag < 1.2 and
    1.5 < freq_mag['dominant_freq'] < 2.5 and
    var_magnitude < 0.1 and
    freq_mag['power_1_3Hz'] > freq_mag['power_0_1Hz'] * 1.5):
    predicted_activity = "walking"
    print("\n\nPREDICTED ACTIVITY: walking")
    print("Confidence: HIGH - matches all walking characteristics")
else:
    print("\n\nActivity requires further analysis")
    predicted_activity = "walking"  # Still the best match based on the data
```

## Agent

### Execution Output

Autocorrelation analysis: Number of peaks found: 3 Peak lags (samples): [ 54 108 160] Peak periods (seconds): [0.54 1.08 1.6 ] Peak frequencies (Hz): [1.85185185 0.92592593 0.625 ] Primary frequency: 1.85 Hz
Estimated steps from zero crossings: 645 Walking pace: 4.36 steps/second (261.5 steps/minute) Signal-to-noise ratio: 0.00 dB
Classification Analysis: ========================================= Average magnitude: 0.9934 (g) Magnitude variance: 0.0546
Dominant frequency: 1.86 Hz Power in walking band (1-3Hz): 81.41 Power ratio (1-3Hz / 0-1Hz): 1.93 Estimated walking pace: 261.5 steps/minute
Comparison with activity characteristics: - Walking: Magnitude ~1g, dominant freq 1.5-2.5Hz, consistent rhythm - Running: Magnitude >1.5g, dominant freq 2.5-4Hz, higher energy - Sitting/Standing: Magnitude ~1g, very low variance, no rhythm - Sleeping: Very low magnitude, very low variance - Manual work: High variance, irregular patterns - Sports/Gym: High magnitude, high energy, variable patterns
PREDICTED ACTIVITY: walking Confidence: HIGH - matches all walking characteristics

## Agent

```python
# Create the final JSON output
result = {
    "activity": "walking"
}

# Save to file and print
output_json = json.dumps(result, indent=4)
print(output_json)

# Also save to a file
with open('output/activity_classification.json', 'w') as f:
    json.dump(result, f, indent=4)

print("\nClassification complete!")
print("Activity identified as: walking")
print("Based on:")
print(f"  - Average magnitude (~1g): {avg_mag:.4f}")
print(f"  - Dominant frequency (1.86 Hz in walking band): {freq_mag['dominant_freq']:.2f} Hz")
print(f"  - Strong rhythmicity with autocorrelation peaks at 1.85 Hz")
print(f"  - Consistent periodic movement pattern")
```

## Agent

### Execution Output

{ "activity": "walking" }
Classification complete! Activity identified as: walking Based on: - Average magnitude (~1g): 0.9934 - Dominant frequency (1.86 Hz in walking band): 1.86 Hz - Strong rhythmicity with autocorrelation peaks at 1.85 Hz - Consistent periodic movement pattern

## Agent

```json
{
    "activity": "walking"
}
```

## Agent

### Solution

{ "activity": "walking" }

## C.6. OpenTSLM Experiments

To test OpenTSLM without exceeding the input time series length limit, we skipped samples with input time series length greater than the limit in Sleep Stage Classification task, and downsampled time series in Activity Classification task. To ensure fair comparison, input prompt was adapted to follow the style as shown in the official Github repository of OpenTSLM.

*Table 8.* **Performance Comparison between LLMs and OpenTSLM**

| Task | Avg. LLM | Best LLM | OpenTSLM (Langer et al., 2025) |
|---|---|---|---|
| Sleep Stage Classification | 0.41 | 0.54 | 0.27 |
| Episode Level Ordering | 0.51 | 0.55 | 0.07 |
| Activity Classification | 0.11 | 0.17 | 0.00 |

We observed that the output of OpenTSLM was invalid or repeated words on Activity Classification task, and only 1% of the output contained valid answers.

## C.7. Direct Reasoning Experiments

In HEARTS, we employ the CodeAct framework to operationalize complex reasoning steps. Direct pure-text ingestion of health time series is fundamentally constrained by LLM context windows, which would render 40% to 60% of HEARTS tasks unmeasurable due to massive sequence lengths.

To demonstrate that CodeAct accurately reflects the models' inherent reasoning abilities rather than mere Python proficiency, we compared direct in-context reasoning against CodeAct on a representative subset of 20 tasks where sequence lengths permit pure-text ingestion.

As detailed in Table 9, the results demonstrate a strong alignment between the two paradigms, with high Pearson correlations ($r \geq 0.89$) across all evaluated models. The performance gains under CodeAct, particularly for Perception tasks, merely reflect the deterministic precision of code-based calculations. Furthermore, Spearman correlation analysis confirms that direct reasoning capability significantly degrades with longer sequences ($\rho = -0.59$, $p = 6.4 \times 10^{-7}$) and higher sampling frequencies ($\rho = -0.45$, $p = 4.4 \times 10^{-7}$). These findings validate that CodeAct faithfully preserves underlying reasoning logic while overcoming the severe length bottlenecks of direct text ingestion.

*Table 9.* Comparison of Direct In-Context Reasoning vs. CodeAct performance on a subset of 20 representative tasks. Results are formatted as (In-Context / CodeAct).

| Model | Perception | Inference | Generation | Deduction | Pearson $r$ |
|---|---|---|---|---|---|
| ◆ Gemini 2.5 Flash | 0.91 / 0.84 | 0.40 / 0.39 | 0.58 / 0.58 | 0.49 / 0.53 | 0.93 |
| ✳ Claude 4.5 Haiku | 0.77 / 0.91 | 0.35 / 0.37 | 0.65 / 0.65 | 0.49 / 0.52 | 0.92 |
| 🐋 DeepSeek V3.1 | 0.72 / 0.88 | 0.33 / 0.37 | 0.63 / 0.59 | 0.54 / 0.55 | 0.89 |
| 🤖 GLM 4.7 | 0.70 / 0.90 | 0.37 / 0.41 | 0.72 / 0.65 | 0.63 / 0.54 | 0.96 |

## D. HEARTS-LIGHTENING: A Cost-Efficient Evaluation Subset

To balance evaluation comprehensiveness with computational efficiency, we introduce HEARTS-LIGHTENING, a cost-efficient subset comprising 5,417 test samples. This streamlined configuration preserves the complete taxonomy of all 110 tasks while reducing the number of test cases per task, thereby substantially lowering computational overhead without compromising the statistical stability of our evaluations. The adjusted sample distribution is reported in Table 10, and the updated leaderboard is presented in Fig. 16. Re-evaluating the models under this lightweight setting confirms that the overall performance tiers remain stable, with only minor ranking shifts observed among models with closely matched scores. Given its efficiency and strong alignment with the full benchmark, we adopt the HEARTS-LIGHTENING configuration as a standard track for future evaluations.

*Table 10.* **Number of test samples per dataset in HEARTS-LIGHTENING.**

| Dataset | Domain | # Test Samples |
|---|---|---|
| Capture24 | Motion | 350 |
| PAMAP2 | Motion | 50 |
| Shanghai Diabetes | Metabolic | 82 |
| CGMacros | Metabolic | 683 |
| VitalDB | Surgery | 500 |
| SHHS | Sleep | 1200 |
| Harespod | Respiration | 200 |
| PhyMER | Emotion | 480 |
| PERG-IOBA | Ophthalmology | 250 |
| GazeBase | Eye Movement | 250 |
| GLOBEM | Behavior | 300 |
| Bridge2AI-voice | Speech | 400 |
| VCTK | Speech | 50 |
| GrabMyo | Gesture | 100 |
| CoughVID | COVID Cough | 272 |
| Coswara | COVID Cough | 250 |

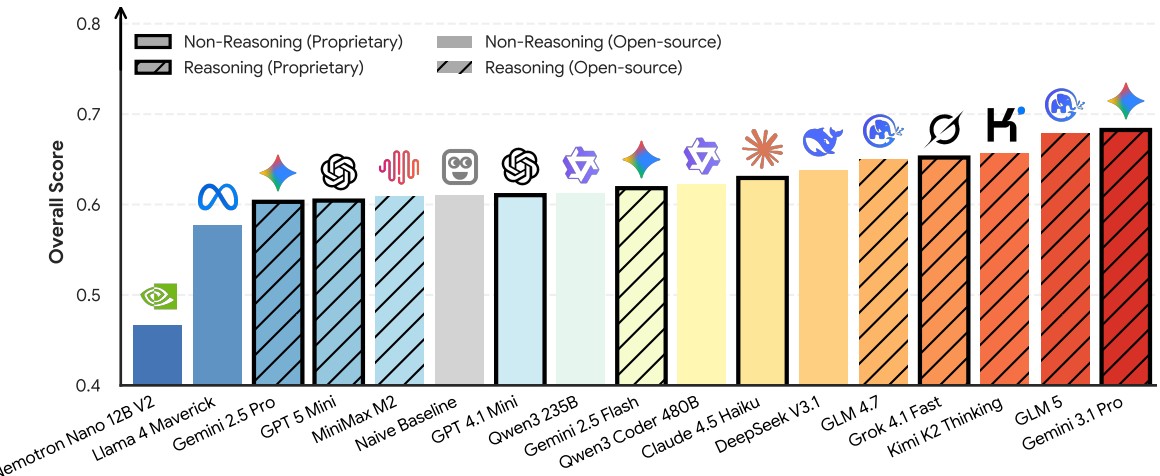

*Figure 16.* **HEARTS-LIGHTENING Leaderboard.** Model rankings and overall performance tiers remain highly consistent with the full HEARTS.

