# OpenReview forum: "HEARTS: Benchmarking LLM Reasoning on Health Time Series"
_ICML.cc/2026/Conference — ICML 2026 regular_

### Official Review · Reviewer_WpYK · 2026-03-03

**Soundness:** 3
**Presentation:** 4
**Significance:** 2
**Originality:** 2
**Overall Recommendation:** 4
**Confidence:** 5

**Summary:**

This paper introduces HEARTS, a large-scale benchmark for evaluating LLM reasoning over health time series. It integrates real-world datasets across 12 domains and 20 modalities, defining 110 tasks organized into four reasoning categories: Perception, Inference, Generation, and Deduction. The authors evaluate 14 LLMs and show that (1) LLMs lag behind specialized models, (2) performance degrades with temporal complexity, and (3) scaling does not significantly improve reasoning ability.
The article's contribution is a unified and comprehensive benchmark for health time-series reasoning.

**Compliance With Llm Reviewing Policy:**

Affirmed.

**Final Justification:**

This paper is sound, well presented, and the authors have addressed my concerns about the soundness of the methodology. Therefore, I keep my score of 4.

**Key Questions For Authors:**

### 1. Dependence on CodeAct and Tool Usage

The evaluation framework relies on CodeAct, where models generate and execute Python code to analyze time-series data. While this setup mitigates context window limitations and enables structured reasoning, it also introduces a tool-augmented setting. To what extent do the reported results reflect intrinsic LLM reasoning ability versus external tool-calling and coding proficiency? Would overall conclusions change significantly if evaluation were conducted in a pure in-context reasoning setting without external Python execution? A controlled comparison would help clarify whether HEARTS assesses reasoning or tool-enabled data-processing skills.


---
### 2. Do LLMs Use Multiple Modalities Effectively?

The benchmark includes tasks with multiple input modalities. An interesting question is whether LLMs can use these modalities in a smart way. For example, when given several channels (e.g., ECG + PPG + respiration), do models learn to focus more on the most informative signals?
- Does performance improve consistently when adding more modalities?
- Do models selectively rely on relevant channels?
- Or do additional modalities sometimes hurt performance?

Such analysis would help understand whether LLMs truly perform cross-modal reasoning, or whether they struggle to integrate heterogeneous physiological signals.

---

### 3. Dataset Contamination and Pretraining Overlap

HEARTS integrates multiple real-world, publicly available datasets. Given that many foundation models are trained on large-scale web data, there is a possibility of dataset exposure during pretraining. How do the authors address the risk of dataset overlap? Any planned measures to assess whether models may have seen parts of these datasets during pretraining?

---
### 4. Inconsistency in Comparison with SOTA ML Models

Some findings related to the comparison with SOTA ML models may be potentially misleading, because the preprocessing steps might be different across models. It is reasonable to treat SOTA ML models as an upper bound. However, while the model architectures are naturally different, the preprocessing pipeline (e.g., filtering, normalization, segmentation, feature extraction) could and should be aligned. If preprocessing differs significantly between LLM-based evaluation and specialized ML baselines, the performance gap may partly come from these differences rather than model capability alone.

---
### 5. Future directions?
The benchmark clearly shows that current LLMs struggle with health time-series reasoning. Given these findings, what do the authors see as the most promising future directions for time-series modeling, especially in the era of LLMs?

**Limitations:**

yes

**Strengths And Weaknesses:**

### Strengths
- Broad and diverse coverage (modalities, domains, temporal scales).
- Extensive empirical evaluation and behavioral analysis.
- Strong benchmarking rigor and comparison with specialized models.

### Weaknesses
- Heavy reliance on CodeAct framework may bias evaluation toward tool-usage rather than pure reasoning.
- Limited discussion on data leakage, dataset overlap, or memorization risks.

---

> ### Author Rebuttal · Authors · 2026-03-31
>
> Thank you for acknowledging our novel contributions and finding the paper comprehensive with strong empirical results. Below we address your concerns in detail, hoping our clarifications lead to a favorable increase of the score.
>
> &nbsp;
> > *W1&Q1: Dependence on CodeAct and tool usage*
>
> We appreciate the comment and would like to clarify our design choice:
>
> **- CodeAct in literature.** Following SOTA reasoning agents [1,2], we adopt CodeAct not as a coding test, but as an established mechanism to operationalize and execute complex, hierarchical reasoning steps.
>
> **- Long sequence handling.** Pure text ingestion cannot support massive sequences and would restrict HEARTS evaluation to only **40-60%** of its tasks.
>
> **- Restricted actions.** To assess inherent reasoning, we intentionally restrict the environment to minimal packages (e.g., `neurokit2`, `mne`), and explicitly exclude advanced tools.
>
> **- Direct reasoning results.** To further address the concerns, we provide direct in-context reasoning results without CodeAct. We kindly refer the reviewer to our reply to **Reviewer H4RB’s W1&Q1** for detailed results. The key takeaway: high correlations confirm CodeAct reliably mirrors pure reasoning and fully supports our claims in **Sec 4.3**.
>
> These results and discussions support our findings in **Sec 4** and will be added into the revision.
>
> &nbsp;
> > *Q2: Do LLMs use multiple modalities effectively?*
>
> We thank the reviewer for the insightful question. In fact, we have already explored the exact topic in **Sec. 4.6** and **Appendix C.5**, with a comprehensive breakdown in **Fig. 15**. Our primary finding is that providing more input modalities does not necessarily improve performance and can even degrade it due to 2 primary reasons:
>
> **- Contextual neglect.** Agents completely ignore auxiliary data and rely exclusively on the target channel.
>
> **- Informational distraction.** The added context is misinterpreted as noise or triggers redundant reasoning.
>
> We will make it clearer in the revision.
>
> &nbsp;
> > *W2&Q3: Dataset contamination and pretraining overlap?*
>
> We appreciate and agree with the comment. We actively mitigate contamination risks through:
>
> **- Task novelty. (Sec. 3.2)** While raw data can be public, our problem formulations, evaluation dimensions, and answer formats are newly designed, which reduces the possibility of direct question-answer memorization.
>
> **- Input transformations.** We conceal dataset origins by providing only basic signal type & frequency, and evaluating strictly on filtered snippets rather than full recordings.
>
> **- Living updates. (Sec. 3.3)** As a **living benchmark**, HEARTS will continuously integrate new datasets alongside formal contamination checks (e.g., option masking and paraphrasing) to assess memorization and performance variation.
>
> We will add the above discussions to the revision, and hope it will serve as a sound basis and inspire future work. We certainly welcome any further specific suggestions regarding possible extensions.
>
> &nbsp;
> > *Q4: Inconsistency in comparison with SOTA ML models?*
>
> Thank you for the thoughtful comment. As explicitly acknowledged in **Appendix B.3**, SOTA models are “*not interpreted as a strictly controlled competitive evaluation*” and solely serve as an “**upper-bound reference**”. They quantify the performance gap and and demonstrate that LLMs remain substantially below specialized models (**Sec. 4.1**, **Fig. 4**, **Table 5**). We intentionally leave preprocessing undefined since autonomous data preparation is integral to the reasoning itself. We will add clarifications to emphasize these points clearer in the revision.
>
> &nbsp;
> > *Q5: Future directions?*
>
> We appreciate the chance to discuss broader impacts. Our results suggest that scaling general-purpose models alone is insufficient for complex health reasoning. A promising path is hybrid systems where large models coordinate specialized temporal solvers, offloading numerical computation while focusing on deduction and overall planning. This elegantly bypasses context limits while maximizing their core strengths in logical deduction and clinical planning.
>
> We will add the above directions to the revised Sec. 5.
> ***
> References
> 1. Biomni: A general-purpose ... 2025
> 2. S1-NexusAgent: A Self-Evolving ... 2026
>
> &nbsp;
> ***
> We hope our responses adequately address all your concerns. Meanwhile, we’d like to update that we have extended our benchmark evaluation to Gemini 3 Pro and Claude Sonnet 4.6. We will expand upon these results in the revision. As stated, we commit to maintaining HEARTS as a **living** ecosystem, **fully open source** the codebase, and welcome contributions from the community to track and add new models / datasets timely. We believe HEARTS will advance the community to help the progress in the field.
>
> Once again, thanks for your time and constructive feedback. We would really appreciate it if the reviewer could consider raising their scores and championing the paper.

---

> > ### Author Rebuttal · Reviewer_WpYK · 2026-04-01
> >
> > Thank the authors for their responses. My concerns are resolved, so I will maintain my original rating.

---

> > > ### Author Response · Authors · 2026-04-05
> > >
> > > Thank you for your encouraging feedback and for acknowledging our rebuttal! We are thrilled that our responses have fully resolved your concerns, and we remain more than happy to address any additional questions that may arise. Given that all initial concerns have now been clarified, if you find it satisfactory, we would be deeply grateful if you might consider raising your final score.
> > >
> > > Once again, we truly appreciate your time, effort, and the insightful suggestions that have significantly strengthened our manuscript.

---

### Official Review · Reviewer_FZ3Q · 2026-03-06

**Soundness:** 3
**Presentation:** 3
**Significance:** 3
**Originality:** 3
**Overall Recommendation:** 4
**Confidence:** 3

**Summary:**

This paper introduces HEARTS, a comprehensive benchmark for evaluating LLM capabilities on generating responses for health time-series data. The benchmark integrates datasets across 12 healthcare domains and multiple signal modalities, and defines a large set of tasks grouped into four capability categories: perception, inference, generation, and deduction, aimed at assessing LLM reasoning (This is a concrete design to evaluate abstract concept of "reasoning"). The authors evaluate 14 LLMs and agent frameworks on large test samples. Results show interesting insights that while LLMs perform well on basic signal perception, they struggle with higher-level temporal reasoning tasks and still lag significantly behind specialized machine learning models on many tasks.

**Compliance With Llm Reviewing Policy:**

Affirmed.

**Final Justification:**

The authors have addressed my main concerns in the rebuttal. Given the additional experimental results, I have increased my score for soundness.

**Key Questions For Authors:**

Q1 (W1). Analysis of model failures.
Could the authors provide more detailed analysis of failure modes, such as reasoning traces, or error breakdowns across task categories? This could help clarify why models struggle on higher-level reasoning tasks.

Q2 (W2). Reasoning process vs. final outcomes.
The benchmark evaluates final task performance. Did the authors analyze intermediate reasoning steps to assess whether models perform meaningful reasoning versus heuristic pattern matching? Such analysis could strengthen the interpretation of the results.

Q3. In ablation study, besides auxiliary signals, would it be interesting to study other forms of contextual information, such as providing domain metadata (e.g., signal type or sampling frequency), task instructions describing how to analyze the signal, or clinical background information?

Q4. The results suggest that input format has only a moderate effect, with visual ingestion generally underperforming text-based formats. This appears inconsistent with prior benchmark work (e.g., BEDTime), where vision–language models often perform competitively or even outperform language-only models on time-series tasks. Could the authors clarify why this difference is likely occuring in this benchmark?

**Limitations:**

yes

**Strengths And Weaknesses:**

Strengths

S1. Good visualizations. The paper includes clear and intuitive figures that help explain the benchmark design, task taxonomy, and data coverage. The visualizations make it easier to understand the scope of the benchmark and how different types of health signals and tasks are organized.

S2. Attempt to concretely define "reasoning" as tasks in four levels of capabilities. The paper organizes tasks into four levels—perception, inference, generation, and deduction—to operationalize different levels of reasoning over time-series signals.

S3. Comprehensive benchmark design.
The benchmark integrates datasets across multiple healthcare domains and signal modalities and defines a large number of tasks. This broad coverage allows evaluation of models across diverse types of health time-series data and reasoning settings.

S4. This is good comparisons against not only LLM but also domain specific ML models.

****
Weaknesses

W1. The paper mainly reports aggregate performance metrics. More detailed analysis, such as qualitative examples, reasoning traces, or error breakdowns, would help readers better understand where models fail and why.

W2. A main concern is that many tasks seem closer to temporal pattern recognition than to reasoning with verifiable intermediate steps. The paper does not clearly investigate whether the generated reasoning process is correct, or whether it helps or hurts the final answer.

W3. In the deduction category, it is not fully clear whether tasks such as Temporal Ordering and Trajectory Analysis truly require deduction, rather than pattern recognition or simple interpretation of temporal trends.

---

> ### Author Rebuttal · Authors · 2026-03-31
>
> Thank you for your thoughtful feedback. Below, we address your concerns in detail. We hope that these will further clarify our work and lead to a favorable increase of the score.
>
> &nbsp;
> > *W1&Q1: Analysis of model failures.*
>
> Thank you for the constructive comment. We agree with the reviewer and provide more evidence below to support our claims.
>
> **- Qualitative Analysis.** Due to strict space limits, we cannot include full traces. However, as discussed in **Sec 4.2** and **Appendix C.2**, LLMs exhibit consistent behavioral patterns and reasoning bottlenecks across tasks. We kindly refer the reviewer to our reply to **Reviewer Qwts’s W3** for detailed summaries.
>
> **- Error Breakdown.** We agree that LLMs often produce "invalid outputs" rather than simply wrong answers. We categorize failures into 3 modes: incorrect content, format non-compliance, and safety guardrail refusals.
>
> Following your suggestion, we will synthesize the above discussions and additional trajectories into the revision.
>
> &nbsp;
> > *W2&Q2: Reasoning process vs final outcomes.*
>
> We appreciate and agree with your comment. In fact, as we explicitly pointed out in **Sec. 5**, evaluating trajectory faithfulness remains a difficult open challenge.
>
> To examine this directly, we conducted a new **LLM-as-judge analysis** [1] using GLM-4.7 to assess GPT-4.1-mini’s reasoning traces on 4 tasks. Using a rubric with 5 task-specific error types we found **judged error patterns do not reliably predict success**. Most failures stem from numerical inaccuracy rather than identifiable heuristics, showing weak correlations with final scores (maximum $r$=-0.18, $p$=0.09). Due to the rebuttal length limit we cannot show the full results here but will detail them in the revision.
>
> &nbsp;
> > *W3: In deduction … simple interpretation of temporal trends*
>
> Thank you for raising this. In fact, Deduction tasks strictly require logical deduction rather than simple pattern recognition. For example, in **Reading Sequence Recognition** task, timestamps are disconnected, so the agent must deduce the timeline from spatial features. A trace excerpt shows a clear deductive chain: (1) formulating a rule ("*readers start at the top and work downward*"), (2) deriving a premise ("*FIRST segment = top, LAST = bottom*"), and (3) computing gaze positions to deduce chronological order. We will add such traces in the revision.
>
> &nbsp;
> > *Q3: In ablation study ... other forms of contextual … information?*
>
> We thank the reviewer for this constructive suggestion. Following your point, we conduct new experiments using GLM 4.7 and DeepSeek V3.1 across:
>
> **- Domain Metadata.** Every HEARTS prompt explicitly defines signal types and sampling frequencies.
>
> **- Task Instructions.** We use SHHS arousal detection task to evaluate the impact of including specific AASM Manual scoring instructions.
>
> **- Clinical Background.** In the SHHS sleep stage classification task, we provide reference bandpowers for each stage as clinical context.
>
> **- Other Information.** HEARTS already compares different forms of auxiliary information. On Coswara, we further examine how explicit symptom descriptions affect COVID diagnosis accuracy.
>
> ||Task Instructions|Clinical Background|Other  Info|
> |-|-|-|-|
> |GLM 4.7 w/o|0.27|0.23|0.56|
> |GLM 4.7 w|0.22|0.42|0.73|
> |Deepseek V3.1 w/o|0.19|0.32|0.53|
> |Deepseek V3.1 w|0.08|0.45|0.74|
>
> Relevant context generally improves performance. Conversely arousal detection instructions introduce complex feature calculations that LLMs struggle to handle causing performance drops. We will detail these findings in the revision.
>
> &nbsp;
> > *Q4: input format has only a moderate effect … inconsistent … occuring?*
>
> Thanks for the question. The performance gap reflects different objectives: BEDTime evaluates **time series description**, where success largely depends on visual properties (as noted in their Sec 4.2.1), while HEARTS evaluates complex **reasoning**. As visual input loses critical quantitative features (e.g. EEG bandpower), visual formats naturally underperform in our setting.
>
> Yet, we recognize multimodal reasoning as a promising future direction and hope HEARTS will serve as a rigorous VLM testbed across diverse settings. We will expand this discussion and appropriately cite these references in the revision.
> ***
> References
> 1. Curing Miracle Steps in LLM ... 2025
>
> &nbsp;
> ***
> We hope our responses adequately address all your concerns. For the revised manuscript we have expanded our evaluation to include Gemini 3 Pro and Claude Sonnet 4.6. As stated, we commit to maintaining HEARTS as a **living** ecosystem, **fully open source** the codebase, and welcome contributions from the community to track and add new models / datasets timely. We believe HEARTS will advance the community to help the progress in the field.
>
> Once again, thanks for your time and constructive feedback. We would really appreciate it if the reviewer could consider raising their scores and championing the paper.

---

> > ### Author Rebuttal · Reviewer_FZ3Q · 2026-03-31
> >
> > Thank you for the response. I have no more questions. I will raise the score of soundness.

---

> > > ### Author Response · Authors · 2026-04-05
> > >
> > > Thank you for the positive feedback and for acknowledging our rebuttal! We are thrilled that your concerns have been fully resolved and are more than happy to answer any additional questions that may arise. Given all concerns have been clarified, if the reviewer feels satisfactory, we would be grateful if the reviewer could consider raising your final score to reflect them.
> > >
> > > We truly appreciate your time, effort, and the insightful suggestions that have significantly improved the quality of our manuscript.

---

### Official Review · Reviewer_H4RB · 2026-03-11

**Soundness:** 1
**Presentation:** 4
**Significance:** 3
**Originality:** 3
**Overall Recommendation:** 4
**Confidence:** 5

**Summary:**

This paper introduces the Health Reasoning over Time Series (HEARTS) benchmark, a large extensible benchmark to evaluate hierarchical reasoning capabilities of LLMS on health time series. The benchmark includes 16 datasets across 12 domains, 20 signal modalities and 110 tasks. They evaluate 14 reasoning and non reasoning LLMs, compare them to naive baselines and SOTA ML models and present a suite of findings including that LLMs underperform specialized models and struggle with increasing temporal complexity.

**Compliance With Llm Reviewing Policy:**

Affirmed.

**Final Justification:**

This paper has strong significance, originality and presentation and the authors adequately addressed my concerns around soundness of the methodology. Therefore I raised my score from a 3 to a 4.

**Key Questions For Authors:**

1.	Given the use of the CodeAct framework, I’m wondering whether this benchmark truly reflects LLMs’ ability to reason directly over health time series data, or whether it primarily captures how well the models can generate Python code for time series analysis. Could the authors clarify how much the CodeAct framework is used in the analyses, and how much direct reasoning the LLMs conduct over the health time series themselves?

2.	Is the LLM model performance comparison with SOTA ML models truly fair since the ML models were trained for those specific tasks while the LLMs were not?

3.	Does the removal of long time series in the Open TSLM experiments introduce bias to the results?

**Limitations:**

yes

**Strengths And Weaknesses:**

Significance and Originality: This paper addresses the important and timely problem of evaluating health time series reasoning capabilities in LLMs. As time series data continues to grow in importance across healthcare fields, there remains a notable lack of large, standardized benchmarks in this space, making this work potentially high‑impact and highly relevant to the community. The breadth of the benchmark is also commendable as the paper evaluates models across 16 datasets, 12 diverse domains, 20 signal modalities, and 110 task types, which strengthens the benchmark’s potential utility and generalizability. In addition, the extensible design of the framework is nice, supporting the integration of new datasets, tasks, or modalities as they emerge in the community.

Presentation: The paper is presented very nicely. The structure is easy to follow, the writing is clear, and the figures are insightful and well‑designed. The analyses are thorough and thoughtfully laid out, making the findings accessible.

Soundness: My main concerns lie in the technical soundness of the paper.
Weakness 1: As detailed at the start of Section 4, the paper states that the LLMs use the CodeAct framework to ingest and read the time series traces. I question the validity of such an approach because this setup appears to evaluate how effectively LLMs can write and execute python code for reading, processing and analyzing time series, instead of directly assessing LLMs innate hierarchical reasoning capabilities for time series as originally stated in the paper. I recognize there are some fundamental limits here (e.g., context window sizes) and that this tool‑augmented evaluation might still be valuable, for instance, as a benchmark for assessing how well LLMs can use external tools to perform time‑series analysis. However, this represents a meaningfully different set of evaluations than what the paper originally positioned as its benchmark contribution (i.e., a measure of LLM capabilities for time series reasoning and inherent understanding of health time series structure.) As a result, I have concerns about the validity of the findings reported in Section 4 (e.g., findings about the LLM’s ability to understand fine-grained temporal distinctions or perform deep temporal reasoning) as these results may reflect limitations in how the Python code was generated or how the data was ingested, rather than the models’ inherent time‑series reasoning capabilities. See question 1.

Weakness 2: I question the comparison with the SOTA ML models as this feels like an apples-to-oranges comparison since the SOTA models were trained and optimized for each specific task whereas the LLMs were not. See question 2.

Weakness 3: OpenTSML experiments- As stated in Appendix C.6, samples with long input time series are skipped for the sleep classification task. This appears to be an unfair comparison which may introduce bias into the results as a large set of the test data is ignored. See question 3.

Minor comment: The naïve baselines for the Perception tasks do not offer much insight since they always report a perfect result (1.00). Would recommend use of an alternative method.

---

> ### Author Rebuttal · Authors · 2026-03-31
>
> Thank you very much for acknowledging the novelty and contributions of our work! We are glad you found the benchmark highly relevant to the community. Below we address your concerns in detail, hoping our clarifications lead to a favorable increase of the score.
>
> &nbsp;
> > *W1&Q1: Given use of CodeAct … direct reasoning LLMs conduct over health time series?*
>
> We appreciate the comment and would like to clarify our design choice:
>
> **- CodeAct in literature.** Following SOTA reasoning agents [1,2], we adopt CodeAct not as a coding test, but as an established mechanism to operationalize and execute complex, hierarchical reasoning steps.
>
> **- Long sequence handling.** Pure text ingestion cannot support massive sequences and would restrict HEARTS evaluation to only **40-60%** of its tasks.
>
> **- Restricted actions.** To assess inherent reasoning, we intentionally restrict the environment to minimal packages (e.g., `neurokit2`, `mne`), and explicitly exclude advanced tools.
>
> **- Direct reasoning results.** To further address the reviewer’s concerns, we provide direct in-context reasoning results via pure text ingestion without CodeAct. For direct comparisons, results below are formatted as (In-Context / CodeAct), alongside their Pearson correlation.
>
> ||Per.|Inf.|Gen.|Ded.|Pearson r|
> |-|-|-|-|-|-|
> |Gemini 2.5 Flash|0.91/0.84|0.40/0.39|0.58/0.58|0.49/0.53|0.93|
> |Claude Haiku 4.5|0.77/0.91|0.35/0.37|0.65/0.65|0.49/0.52|0.92|
> |Deepseek V3.1|0.72/0.88|0.33/0.37|0.63/0.59|0.54/0.55|0.89|
> |GLM 4.7|0.70/0.90|0.37/0.41|0.72/0.65|0.63/0.54|0.96|
>
> Consistent scores and high correlations (**r>0.89**) confirm CodeAct faithfully reflects pure reasoning, with expected Perception gains merely reflecting precise code calculations. Following **Sec 4.3**, Spearman correlation indicates that direct reasoning significantly degrades with longer sequences ($\rho$=-0.59, p=6.4e-7) and higher sampling frequencies ($\rho$=-0.45, p =4.4e-7).
>
> These results and discussions support our findings in **Sec 4** and will be added into the revision.
>
> &nbsp;
> > *W2&Q2: Is LLM … SOTA ML models truly fair?*
>
> Thank you for raising this. We believe there is a misunderstanding here by the reviewer. As explicitly acknowledged in **Appendix B.3**, SOTA models are “*not interpreted as a strictly controlled competitive evaluation*” and solely serve as an “**upper-bound reference**”. Since naive baselines establish lower-bound, SOTA models help quantify the performance gap and approximate room for improvements. This comparison intentionally exposes current LLM limitations, demonstrating their performance remains substantially below specialized models even under best case selection (**Sec. 4.1**, **Fig. 4**, **Table 5**). We will clarify this intent in the revision.
>
> &nbsp;
> > *W3&Q3: Does the removal … OpenTSLM introduce bias?*
>
> Thanks for pointing this out, and we apologize if it wasn’t clear. In fact, excluding long sequences was a **strict technical necessity due to OpenTSLM limitations** rather than a biased design choice
>
> While OpenTSLM claims to “*reason over multiple time series of any length*”, we empirically found its allowed sequence length is restricted to ~10K. As this limit affects **45%** (90/200) of the sleep stage test cases, we explicitly excluded them to ensure a *fair comparison* with other LLMs:
>
> ||Seq len ≤ 10K|Seq len > 10K|
> |-|-|-|
> |OpenTSLM|0.27|-|
> |Avg. LLM|0.42|0.41|
>
> We will explicitly clarify these technical limits and comparison settings in Appendix C.6.
>
> &nbsp;
> > *Minor comment: The naïve baselines … an alternative method*
>
> Thank you for this observation. We acknowledge that a 1.00 baseline might seem uninformative. This reflects the **clear, deterministic analytical solutions** inherent to Perception tasks. For example, calculating sleep efficiency relies on a strict mathematical definition (total sleep time / time in bed), naturally yielding a perfect score when executed correctly.
>
> Yet, we agree with the reviewer’s point, and we will mark (*) those numbers and explain our rationale in a footnote explicitly. We also welcome any concrete suggestions from the reviewer on how to better state it to be more informative.
> ***
> References
> 1. Biomni: A general-purpose ... 2025
> 2. S1-NexusAgent: A Self-Evolving ... 2026
>
> &nbsp;
> ***
> We hope these responses adequately address all your concerns. We would like to also give an update that we have extended our benchmark to include two recently released models: Gemini 3 Pro and Claude Sonnet 4.6.
>
> We will continue expanding upon these results in the revision. As stated, we commit to maintaining HEARTS as a **living** ecosystem, **fully open source** the codebase, and welcome community contributions to track and add new models / datasets timely. We believe HEARTS will advance the community to help the progress in the field.
>
> Once again, thanks for your time and constructive feedback. We would really appreciate it if the reviewer could consider raising their scores and championing the paper.

---

> > ### Author Rebuttal · Reviewer_H4RB · 2026-03-31
> >
> > Thank you for your detailed response. You have sufficiently addressed my concerns so I will raise my score.

---

> > > ### Author Response · Authors · 2026-04-05
> > >
> > > We sincerely appreciate your encouraging feedback and strong support for our paper! We are thrilled that our rebuttal has fully resolved your initial concerns, and we remain entirely at your disposal for any further questions. Once again, thank you for the time and insightful comments that have significantly improved our manuscript.

---

### Official Review · Reviewer_Qwts · 2026-03-12

**Soundness:** 3
**Presentation:** 4
**Significance:** 3
**Originality:** 2
**Overall Recommendation:** 4
**Confidence:** 3

**Summary:**

This paper proposes a benchmark to evaluate how LLM-based agents perform for health time series analysis tasks. It integrates 16 real-world datasets across 12 domains and 20 modalities. The tasks are categorized into Perception, Inference, Generation, and Deduction types. The benchmark is developed based on the CodeAct agent framework, which generates executable Python code to analyze time-series data and output results.

**Compliance With Llm Reviewing Policy:**

Affirmed.

**Final Justification:**

My concerns are addressed in the rebuttal, so I will keep my original positive recommendation unchanged.

**Key Questions For Authors:**

Please respond to the weaknesses above.

**Limitations:**

yes

**Strengths And Weaknesses:**

**Strengths:**
1. The paper is well written and easy to follow. The figures and visualizations are clear.
2. The dataset constructed is comprehensive. It covers diverse health domains, modalities, signal frequencies, and time ranges. This dataset could allow a much more comprehensive evaluation for future research efforts on health time series analysis.
3. The benchmark covers many open-source and proprietary LLM models, which offer comprehensive comparisons among them.
4. Both the dataset and the benchmark are open-sourced to allow further research along this direction.

**Weaknesses:**
1. The benchmark only covers general-purpose LLMs. It would be much more interesting to investigate how time-series foundation models or LLMs (e.g., Time-LLM [1]) compare with the general-purpose LLMs.
2. The action space is not described in detail. What are the allowed actions? Are all Python packages allowed to be used? Are there any models/tools specific to time-series analysis used as possible tools?
3. The benchmark only provides overall performance metrics, while it would be more interesting to look into the reasoning trajectory to pinpoint the reasoning bottleneck for future improvements. Appendix C.5.2 provides one example for the classification task. More examples for complex tasks could provide more insights.

**Minor issues:**
- Line 240, Page 5: "The sMAPE-based tasks, we first apply ..." should be "For the sMAPE-based tasks, we first apply ..."
- In the reference list, a few references have missing fields (e.g., (Wang et al., 2024a) is missing the conference name of ICML).




[1] Jin, Ming, et al. Time-LLM: Time series forecasting by reprogramming large language models. ICLR. 2024.

---

> ### Author Rebuttal · Authors · 2026-03-31
>
> Thank you for your supportive remarks! We are delighted you found the benchmark comprehensive and experiments extensive. In the following, we address your concerns in detail, hoping this will lead to a favorable increase of the score.
>
> &nbsp;
> > *W1: The benchmark only covers … time-series foundation models or LLMs (e.g., Time-LLM)*
>
> Thank you for your constructive comment. We agree that comparing general purpose LLMs with specialized time series (TS) models is crucial.
>
> First, we would like to clarify that we already included a comparison with a general TS reasoning model OpenTSLM. As detailed in **Sec. 4.1**, **Appendix C.6**, and **Table 8**, we evaluated OpenTSLM on 3 HEARTS tasks fitting its pretraining and context limits.
>
> Moreover, to further address your concern, we evaluate two available TS FMs, **TimesFM** [1] and **Chronos-2** [2]. Time-LLM [3] does not provide public checkpoints, so we omit now and leave it for future evaluations. As both models are forecasting centric, we evaluated them on 4 specific forecasting tasks that fall within their context limits:
>
> ||Step Count (GLOBEM)|CGM (CGMacros)|HR (PhyMER)|MBP (VitalDB)|
> |-|-|-|-|-|
> |Best LLM|0.78|0.56|0.62|0.58|
> |Chronos-2|0.78|0.50|0.67|0.64|
> |TimesFM|0.79|0.55|0.66|0.63|
>
>
> The findings support our discussion in **Sec. 4.1** that general purpose LLMs still lag behind specialized models on pure forecasting tasks. We will incorporate these additional results and discussions in the revision.
>
> &nbsp;
> > *W2: The action space … any models/tools specific to time-series analysis used as possible tools?*
>
> Thank you for raising this. We provide further clarification as follows:
>
> **(1) Action Space.** As stated in Sec. 3.2, we formalize the reasoning process as a Markov Decision Process $\tau = \{(s_i, a_i, r_i)\}_{i=0}^n$, where actions $a_i \in A$ denote intermediate decisions. Within CodeAct framework, they are strictly limited to *thinking* and *generating & executing Python code*.
>
> **(2) Allowed Tools.** We intentionally restrict the environment to minimal Python packages necessary for health TS processing (e.g., `neurokit2`, `mne`, and `statsmodels`). Our intention is to **evaluate the model's intrinsic reasoning capabilities**, rather than its ability to call advanced tools. Consequently, high-level, cutting-edge ML models and specialized TS solvers are intentionally excluded.
>
> We hope this clarifies our design choices. All aforementioned details, including the full available package list, will be incorporated into the revision to ensure clarity.
>
> &nbsp;
> > *W3: The benchmark … examples for complex tasks could provide more insights.*
>
> Thank you for the comment. We agree that analyzing reasoning trajectories is crucial.
>
> Due to strict space limits, we cannot include full trajectories. However, as discussed in **Sec. 4.2** and **Appendix C.2**, LLMs exhibit consistent behavioral patterns and reasoning bottlenecks across tasks. We summarize these commonalities below to further support our conclusions:
>
> |Task Category|Typical Behaviors|
> |-|-|
> |Perception|Near-perfect (>0.9) w/ known package; ~0.7 w/out (**Appendix C.2.1**)|
> |Inference|Succeeds using clear quantitative thresholds and common sense. Struggles heavily with fine grained distinctions or specialized domain features.|
> |Generation|Copy-paste, interpolation, simple averages, formulaic signal processing (**Appendix C.2.2**)|
> |Deduction|Lacks a fundamental grasp of temporal continuity. Depends entirely on general knowledge proxies.|
>
> Following your feedback, we will include the table alongside a set of detailed reasoning traces for more complex tasks in the revision.
>
> &nbsp;
> > *Issues (typos, references)*
>
> Thank you for the constructive feedback. We really appreciate them! We will definitely correct them per your detailed suggestions.
> ***
> References
> 1. A decoder-only foundation model for time ... 2024
> 2. Chronos-2: From univariate ... 2025
> 3. Time-llm: Time series forecasting ... 2025
>
> &nbsp;
> ***
> We hope these responses adequately address all your concerns. Meanwhile, we would like to also give an update that we have extended our benchmark to include 2 recently released models: Gemini 3 Pro and Claude Sonnet 4.6. We briefly summarize the results below, which align and support our claims in the paper (**Sec 4.4**):
>
> ||Perception|Inference|Generation|Deduction|
> |-|-|-|-|-|
> |Gemini 3 Pro|0.92|0.53|0.78|0.89|
> |Claude Sonnet 4.6|0.96|0.49|0.78|0.89|
>
> We will continue expanding upon these results in the revision. As stated, we commit to maintaining HEARTS as a **living** ecosystem, **fully open source** the codebase, and welcome community contributions to track and add new models / datasets timely. We believe HEARTS will advance the community to help the progress in the field.
>
> Once again, thanks for your time and constructive feedback. We would really appreciate it if the reviewer could consider raising their scores and championing the paper.

---

> > ### Author Rebuttal · Reviewer_Qwts · 2026-04-01
> >
> > My concerns are resolved, so I will maintain my original rating.

---

> > > ### Author Response · Authors · 2026-04-05
> > >
> > > Thank you for your encouraging remark! We are glad that our responses have addressed all your concerns. We are happy to address any further questions. Given all concerns have been clarified, if the reviewer feels appropriate, we would be grateful if the reviewer could consider raising your final score to reflect them.
> > >
> > > Once again, thank you for your time and the constructive feedback that has helped strengthen our paper.

---

### Decision · Program_Chairs · 2026-04-30

**Decision:**

Accept (regular)

**Comment:**

This paper presents HEARTS, a large and well-designed benchmark for evaluating LLM reasoning on health time series. Its main strength is breadth: it covers many domains, modalities, and task types, and the resulting empirical analysis yields useful insights into where current LLMs succeed and fail. Several reviewers found the resource likely to be valuable to the community.

The main concerns were about whether the benchmark measures intrinsic reasoning versus tool-augmented reasoning, the fairness of comparison to specialized ML models, and the need for more detailed failure analysis. The rebuttal addresses these points reasonably well by clarifying the intended role of CodeAct, adding direct in-context comparisons, and sharpening the interpretation of specialized models as upper-bound references. While some methodological caveats remain, I do not think they outweigh the contribution of the benchmark itself. I therefore recommend Weak Accept.